# Efficient Algorithms for Robust and Partial Semi-Discrete Optimal Transport

**Pankaj K. Agarwal**
Department of Computer Science
Duke University

**Sharath Raghvendra**
Department of Computer Science
North Carolina State University

**Pouyan Shirzadian**
Department of Computer Science
Virginia Tech

**Keegan Yao**
Department of Computer Science
Duke University

## Abstract

The sensitivity of optimal transport (OT) to noise has motivated the study of robust variants. In this paper, we study two such formulations of semi-discrete OT in $\mathbb{R}^d$: (i) the $\alpha$-optimal partial transport, which minimizes the cost of transporting a mass of $\alpha$; and (ii) the $\lambda$-robust optimal transport, which regularizes the OT problem using the total variation (TV) distance. First, we provide a novel characterization of the optimal solutions in these settings, showing they can be represented as a restricted Laguerre diagram. Second, we exploit this characterization to establish a strong algorithmic connection between the two problems, showing that any solver for one can be adapted to solve the other with comparable precision. Third, we overcome key challenges posed in extending the cost-scaling paradigm to compute these variants of OT and present an algorithm that computes the exact solution up to $\log(1/\varepsilon)$ bits of precision in $n^{O(d)}\log(1/\varepsilon)$ time, where $n$ is the support size of the discrete distribution. Finally, we present an $n^{1+o(1)}\varepsilon^{-O(d)}$ time approximation algorithm for the above variants of OT.

## 1 Introduction

Given two distributions $\mu$ and $\nu$ supported on sets $A$ and $B$ in $\mathbb{R}^d$ and a parameter $p \geq 1$, the cost of transporting $m$ units of mass from $a \in A$ to $b \in B$ is $m\|a - b\|^p$. The *optimal transport cost* (also called the *p-Wasserstein distance*) quantifies the minimum total effort required to move mass from $\mu$ to $\nu$. In many machine learning applications, one wishes to compute the OT cost between model and target distributions that are continuous. Due to lack of efficient algorithms that work directly on continuous distributions, practitioners often approximate the OT cost by computing it between two discrete empirical distributions formed by $n$ samples [5, 17, 18, 39]. This OT between two discrete distributions can be computed by using a minimum-cost flow algorithm [38].

A more computationally feasible variant is the *semi-discrete optimal transport* (SDOT) problem, where one distribution is continuous and the other is discrete. In this case, the OT cost and plan can be characterized by a set of weights $y\colon B \to \mathbb{R}_{\geq 0}$ such that, for any point $b \in B$, the additively weighted Voronoi cell $V_y(b)$ of $b$ (also called the Laguerre cell) has the same probability mass as $b$. The OT plan assigns all mass in $V_y(b)$ to $b$ [8]. The semi-discrete OT has found numerous applications in machine learning [6, 20, 27, 31, 41].

Despite its usefulness, OT cost may be unduly influenced by outliers in either distribution. Two robust variants help reduce the impact of noise. The first variant, called the *$\alpha$-optimal partial transport* ($\alpha$-OPT), minimizes the cost of transporting a mass of $\alpha \in [0, 1]$ from $\mu$ to $\nu$, potentially discarding

39th Conference on Neural Information Processing Systems (NeurIPS 2025).

outlier mass [14]. The second variant, called the $\lambda$-*robust OT* ($\lambda$-ROT), introduces a regularization via total variation distance. Given a parameter $\lambda > 0$, $m$ units of mass can either be transported from $a \in A$ to $b \in B$ incurring a cost of $m\|a - b\|^p$ or be burnt incurring a cost of $\lambda m$ [34]. The $\lambda$-ROT minimumizes the cost under this burn-or-transport trade-off. In this paper, we study both $\alpha$-OPT and $\lambda$-ROT in the semi-discrete setting. We provide a characterization of them and design both exact and approximation algorithms for them.

**Related Work.** Several algorithms have been presented for additively approximating the semi-discrete OT using numerical solvers [8, 9, 15, 20, 29, 30, 32, 37] and entropic regularization [4, 10, 24]. These algorithms take $n^{O(d)}\text{poly}(1/\varepsilon)$ time to compute this approximation.

Recently, Agarwal *et al.* [1] adapted the cost-scaling paradigm that is used for the network-flow algorithm to the semi-discrete settings, obtaining the exact solution with $\log(1/\varepsilon)$ bits of precision in $n^{O(d)}\text{poly}\log 1/\varepsilon$ time. Thus, this algorithm achieves a polynomial time exact solution with $\text{poly}(n)$ bits of precision, while prior algorithms required exponential (in $n$) time to achieve the same accuracy. Agarwal *et al.* [3] also developed a *near-linear-time* Monte Carlo algorithm that provides a relative approximation. It is worth noting that *weighted Voronoi diagrams* under $\ell_p$ metrics can have complexity as high as $n^{\Omega(d)}$ in the worst case. This inherent geometric complexity suggests that substantially improving upon the current $n^{O(d)}$-time bounds for exact solutions may be difficult.

The $\alpha$-optimal partial transport ($\alpha$-OPT) problem for discrete distributions was introduced by Chapel *et al.* [14] as a means to detect and remove outliers, particularly when the fraction of outliers is known in advance; see also [21, 35] for useful properties of $\alpha$-OPT. Phatak *et al.* [40] later proposed computing the $\alpha$-OPT cost as a function of $\alpha \in [0, 1]$, allowing automatic outlier detection and removal without prior knowledge of the proportion of outliers. Although the $\alpha$-OPT distance is not a metric, Raghvendra *et al.* [42] introduced an $\alpha$-OPT based metric, called the RPW distance, that enjoys improved statistical robustness compared to the $p$-Wasserstein distance. In the semi-discrete settings, it is known that the $\alpha$-OPT routes mass from each discrete point to a subset of its weighted Voronoi cell [11], and the support of a partial transport plan satisfies the interior ball condition [12]— that is, the transported continuous mass lies within the union of balls of a fixed radius centered at the support of the discrete distribution.

Mukherjee *et al.* [34] introduced the $\lambda$-ROT (originally referred to by ROBOT distance). There is substantial work in establishing mathematical properties of $\lambda$-ROT distance as well as designing algorithms for it [33, 36]. Most of the existing algorithms for both $\alpha$-OPT and $\lambda$-ROT distances are restricted to discrete settings and no prior work has extended these robust formulations to the semi-discrete settings. We address the gap in this paper.

**Our Results.** We introduce the semi-discrete versions of $\alpha$-OPT and $\lambda$-ROT and make four novel contributions related to them.

First, we show that the optimal solution for the semi-discrete $\alpha$-OPT ($\alpha$-SDOPT) can be characterized using a *restricted Voronoi diagram* (see Lemma 2.1). The restricted Voronoi diagram combines the weighted Voronoi diagram structure [11] with the interior ball condition [12]. We further establish a stronger characterization than was previously known: in the optimal solution, for each point $b \in B$, all continuous mass within its *restricted Voronoi cell* is transported from $b$ and either (i) all mass of $b$ is transported, or (ii) $b$ attains the maximum weight among all points of $B$. We also establish a similar characterization for the semi-discrete $\lambda$-ROT ($\lambda$-SDROT) problem; see Lemma 2.2 in Section 2. To our knowledge, this is the first paper to give such a characterization of partial or robust OT in the semi-discrete setting.

Second, we demonstrate that any *approximate* solver for $\lambda$-SDROT can be used to obtain an approximate solution to the $\alpha$-SDOPT problem. Until now, these two problems have been studied separately, each with its own family of exact and approximation algorithms. The standard reduction used to establish duality between them [12] does not, however, preserve approximation guarantees, as we highlight in Section 3. We provide a refined reduction showing that an approximate solution to one problem can still be transformed into an approximate solution to the other with nearly matching accuracy.

Third, using our characterization of partial and robust OT in the semi-discrete setting, we design a cost-scaling algorithm for the SDROT problem. Extending cost-scaling approaches to solve optimal partial transport is challenging, even in the fully discrete case [43], and these challenges only increase

in the semi-discrete setting. We detail this challenge as well as our approach for resolving them in Section 4. Roughly speaking, the cost-scaling algorithm maintains an approximate restricted Voronoi diagram, which is successively refined at each scale. The algorithm maintains the invariants that the untransported mass of the continuous distribution lies outside the approximate Voronoi cells. Ensuring this invariant as we move from one scale to the next is challenging and requires new ideas. Our algorithm runs in $n^{O(d)} \log(\Delta/\varepsilon)$ time and computes a solution to the SDROT problem with $\log(1/\varepsilon)$ bits of precision; here $n$ is the support size of the discrete distribution and $\Delta$ is the diameter of the two distributions. Through our reduction, we can also compute a solution for the $\alpha$-SDOPT problem with a similar performance. We develop a preliminary implementation of this algorithm and experimentally demonstrate that (i) the algorithm executes within the theoretically predicted number of iterations, and (ii) both SDOPT and SDROT predominantly transport inlier mass. See Section 4.

Finally, we describe an $\varepsilon$-approximation algorithm for the SDROT problem running in near-linear time in $n$. Our algorithm first reduces SDROT to the discrete robust OT problem, then reduces the problem to computing a transport plan on a sparse graph, and finally either uses the recent minimum-cost flow algorithm [16] or uses the multiplicative-weight update method as in [3, 22, 28]. Although the algorithm builds on the one in [3], new ideas are needed at both reduction steps to make it work for partial OT. For $p$-Wasserstein distance, i.e., the cost is $\|a - b\|^p$, the algorithm takes $n^{1+o(1)}\varepsilon^{-4d} \log \Delta \log \varepsilon^{-1}$ time for any constant value of $p$. For $p = 1$, we present a faster Monte Carlo algorithm that runs in $O(n\varepsilon^{-4d-4} \log^3 \Delta \log n \log \varepsilon^{-1})$ time. Because of the lack of space, this algorithm is presented in Appendix A. See Theorem A.1.

## 2 Characterizing Partial and Robust Semi-Discrete Optimal Transport

In this section, we provide a characterization of partial and robust semi-discrete OT using weighted Voronoi diagrams. We begin by giving a formal definition of the variants of OT.

**Partial and Robust OT.** Suppose $\mu$ is a continuous probability distribution defined over compact support $A \subset \mathbb{R}^d$ and $\nu$ is a discrete distribution with support set $B$ of $n$ points in $\mathbb{R}^d$. For a parameter $p \geq 1$, define the distance of each pair of points $(a, b) \in A \times B$ as $\mathrm{d}(a, b) := \|a - b\|^p$. A *transport plan* $\tau : A \times B \to \mathbb{R}_{\geq 0}$ is a function whose marginals are dominated by $\mu$ and $\nu$, i.e., $\sum_{b \in B} \tau(a, b) \leq \mu(a)$ for all $a \subseteq A$ and $\int_A \tau(a, b)\, da \leq \nu(b)$ for all $b \in B$. The *cost* of a transport plan $\tau$ between $\mu$ and $\nu$ is defined as $\mathcal{c}(\tau) := \int_A \sum_{b \in B} \mathrm{d}(a, b)\tau(a, b)\, da$. Define $\mathrm{M}(\tau)$ as the total amount of mass transported by a transport plan $\tau$, i.e., $\mathrm{M}(\tau) := \int_A \sum_{b \in B} \tau(a, b)da$.

For a parameter $\alpha \in [0, 1]$, $\tau$ is an $\alpha$-*partial transport plan* if $\mathrm{M}(\tau) = \alpha$. An $\alpha$-*optimal partial transport ($\alpha$-OPT) plan* is a minimum-cost $\alpha$-partial transport plan.

For a parameter $\lambda > 0$, define the $\lambda$-*robust cost* of a transport plan $\tau$ as $w_\lambda(\tau) := \mathcal{c}(\tau) + (1 - \mathrm{M}(\tau))\lambda$. A $\lambda$-*robust optimal transport ($\lambda$-ROT) plan* is a transport plan with the minimum $\lambda$-robust cost.

**Voronoi cells.** For a set of weights $y(\cdot)$ for the points in $B$, the *weighted distance* between each point $a \in A$ and $b \in B$ is defined as $\mathrm{d}_y(a, b) := \mathrm{d}(a, b) - y(b)$. For any point $a \in A$, define its *(weighted) nearest neighbor* as the point $b \in B$ with the smallest weighted distance to $a$. For any point $b \in B$, the *Voronoi cell* of $b$ is the set of points of $A$ with $b$ as their weighted nearest neighbor, i.e.,

$$V_y(b) := \{a \in A \mid \mathrm{d}_y(a, b) \leq \mathrm{d}_y(a, b') \,\forall b' \in B\}.$$

See Figure 1(left). It is well known that for any set of weights $y(\cdot)$ for $B$ such that $\mu(V_y(b)) = \nu(b)$ for all points $b \in B$, the transport plan that transports the mass of each point $b \in B$ to the mass of $\mu$ inside $V_y(b)$ is an OT plan. Furthermore, one can prove the existence of such weights using the LP formulation of semi-discrete OT and the strong LP duality; see [8, 19, 23].

**Restricted Voronoi cells.** Given a set of non-negative weights $y(\cdot)$ for points in $B$, for any point $b \in B$, the *restricted Voronoi cell* of $b$ captures the points of the Voronoi cell of $b$ that are within a distance of $y(b)$ from $b$, i.e.,

$$RV_y(b) := \{a \in V_y(b) \mid \mathrm{d}(a, b) \leq y(b)\}.$$

See Figure 1(right). We refer to any point $b \in B$ as *balanced* (resp. *surplus*, *deficit*) if $\mu(RV_y(b)) = \nu(b)$ (resp. $\mu(RV_y(b)) < \nu(b)$, $\mu(RV_y(b)) > \nu(b)$).

**Valid weights, cover, and cap.** A weight function $y : B \to \mathbb{R}_{\geq 0}$ is called *valid* if all points $b \in B$ are either balanced or surplus and the surplus points have the maximum weight, i.e., for all surplus

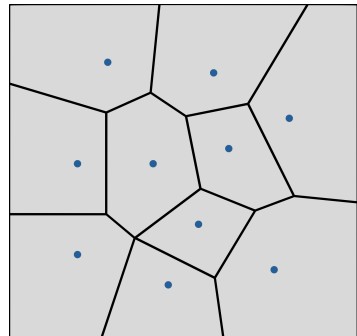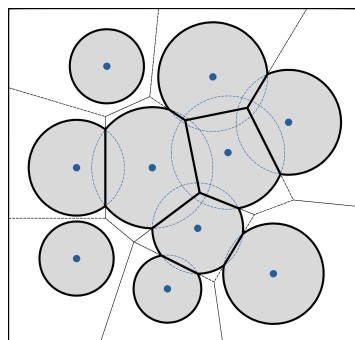

Figure 1: (left) Voronoi cells, and (right) restricted Voronoi cells

points $b \in B$, $y(b) = \max_{b' \in B} y(b')$. For a valid weight function $y$, we define the transport plan *induced by* $y$, denoted by $\tau_y : A \times B \to \mathbb{R}_{\geq 0}$, as the one that for each point $b \in B$, transports the mass of $b$ to the continuous mass of $\mu$ inside the restricted Voronoi cell $RV_y(b)$. Since each point $b \in B$ is surplus or balanced, there is enough mass at $b$ to transport to $RV_y(b)$. We define the *cover* of $y(\cdot)$ to be $\mathrm{Cover}(y) := \sum_{b \in B} \mu(RV_y(b))$, and its *cap* to be $\mathrm{Cap}(y) := \max_{b \in B} y(b)$.

**Characterization of partial and robust OT.** The following lemma characterizes an $\alpha$-OPT plan using a set of valid weights with a cover of $\alpha$.

**Lemma 2.1** ($\alpha$-OPT Characterization). *Let $\mu$ be a continuous distribution over compact support $A \subset \mathbb{R}^d$, $\nu$ a discrete distribution over a set $B$ of $n$ points in $\mathbb{R}^d$, and $\alpha \in [0,1]$ a parameter. There exists a valid weight function $y : B \to \mathbb{R}_{\geq 0}$ with $\mathrm{Cover}(y) = \alpha$ such that the transport plan $\tau_y$ induced by $y$ is an $\alpha$-OPT plan. Furthermore, this property holds for any valid weight function $y$ with $\mathrm{Cover}(y) = \alpha$.*

*Proof Sketch.* We provide a brief sketch of the proof below. The complete proof is provided in Appendix C. The $\alpha$-partial transport problem is a linear optimization problem with the linear primal objective function $\phi(\tau) := \int_A \sum_{b \in B} \tau(a,b) \mathrm{d}(a,b) da$ and mass constraints $\int_A \tau(a,b) da \leq \nu(b)$, $\sum_{b \in B} \tau(a,b) \leq \mu(a)$ and $\int_A \sum_{b \in B} \tau(a,b) da = \alpha$. We consider its corresponding dual optimization problem, which can be written as:

$$\max_{\lambda, \phi, \psi} \quad \alpha\lambda - \sum_{b \in B} \phi(b)\nu(b) - \int_A \psi(a)\mu(a)\, da$$
$$\text{subject to} \quad \lambda - \phi(b) - \psi(a) \leq \mathrm{d}(a,b), \qquad \forall (a,b) \in A \times B,$$
$$\phi, \psi \geq 0. \tag{1}$$

Set $y(b) = \lambda - \phi(b)$. By complementary slackness conditions on the optimal solutions for the primal and dual problems, one can argue that if primal constraint $\int_A \tau(a,b) da \leq \nu(b)$ is not tight then $y(b) = \lambda$, and that if primal variable $\tau(a,b) > 0$ then $y(b) - \psi(a) = \mathrm{d}(a,b)$. By feasibility of the optimal dual solution, both of these conditions imply that (i) $y(b) = \max_{b' \in B} y(b')$ for every surplus point $b \in B$, and (ii) $\tau(a,b) > 0$ for pairs $(a,b)$ only if $a \in RV_y(b)$.

Next, we differentiate the dual objective with respect to the unconstrained variable $\lambda$, argue that the dual objective is maximized when $\mathrm{Cover}(y) = \alpha$, and use strong duality and conclude the existence of optimal weights $y$. We finally use the feasibility constraints of the dual problem to argue that any such weights $y$ satisfying these conditions induce an optimal transport plan. $\square$

Next, we characterize a $\lambda$-robust optimal transport plan using a set of valid weights with a cap of $\lambda$.

**Lemma 2.2** ($\lambda$-ROT Characterization). *Let $\mu$ be a continuous distribution over compact support $A \subset \mathbb{R}^d$, $\nu$ a discrete distribution over a set $B$ in $n$ points in $\mathbb{R}^d$, and $\lambda > 0$ a parameter. There exists a valid weight function $y : B \to \mathbb{R}_{\geq 0}$ with $\mathrm{Cap}(y) = \lambda$ such that the transport plan induced by $y$ is a $\lambda$-ROT plan. Furthermore, this property holds for any valid weight function $y$ with $\mathrm{Cap}(y) = \lambda$.*

Suppose $y(\cdot)$ is a set of valid weights with a cap of $\lambda$ and cover of $\alpha$. A transport plan $\tau$ transporting the mass of each point $b \in B$ to the mass of $\mu$ inside $RV_y(b)$ would be both an $\alpha$-OPT plan as well

as a $\lambda$-ROT plan. In the next section, we show a strong relationship between the two robust variants of the semi-discrete OT problem.

**Relating the two characterizations.** For any value $\lambda \geq 0$, let $\mathrm{MaxCover}(\lambda)$ denote the largest value $\alpha \in [0, 1]$ such that there exists a set of valid weights $y(\cdot)$ for $B$ with a cover of $\alpha$ and a cap of $\lambda$. We note that $\mathrm{MaxCover}(\lambda)$ gives the maximum value of mass that can be transported if we restrict any mass to be transported at most $\lambda$ distance. Similarly, for any value $\alpha \geq 0$, let $\mathrm{MinCap}(\alpha)$ denote the smallest value $\lambda \geq 0$ such that there exists a set of valid weights $y(\cdot)$ for $B$ with $\mathrm{Cover}(y) = \alpha$ and $\mathrm{Cap}(y) = \lambda$. That is, the maximum distance the mass has to be transported in any $\alpha$-OPT plan has to be at least $\mathrm{MinCap}(\alpha)$. We show in the following lemma that both functions are monotonically non-decreasing.

**Lemma 2.3.** *The mappings* $\mathrm{MaxCover} \colon \mathbb{R}_{\geq 0} \to [0, 1]$ *and* $\mathrm{MinCap} \colon [0, 1] \to \mathbb{R}_{\geq 0}$ *are monotonically non-decreasing functions.*

From Lemmas 2.2 and 2.3, given an exact black-box solver $\mathcal{A}$ for the robust OT problem, one can compute a highly accurate $\alpha$-OPT plan using a simple binary search on the value of $\mathrm{MaxCover}(\alpha)$. Hence, we obtain the following:

**Proposition 2.4.** *Given a procedure* $\mathcal{A}$ *for computing semi-discrete robust OT, for any parameter* $\alpha \in [0, 1]$ *that requires* $m$ *bits of representation, a semi-discrete* $\alpha$-OPT *can be computed by making* $O(m)$ *calls to* $\mathcal{A}$.

## 3 Relating Approximate Partial Transport Models

In this section, we show that an approximate solver for $\lambda$-robust OT can be used to compute a near-optimal transport plan. We begin by formally defining near-optimal transport plans.

**Approximate transport plans.** We call an $\alpha$-partial transport plan $(\varepsilon_r, \varepsilon_a)$-*approximate* if $\cancel{c}(\tau) \leq (1 + \varepsilon_r)\cancel{c}(\tau^*) + \varepsilon_a$, i.e., both relative and additive error terms are allowed. Analogously, define a transport plan $\tau$ where $\mathrm{M}(\tau) = \alpha$ as an $(\varepsilon_r, \varepsilon_a)$-approximate $\lambda$-robust optimal transport plan if $\cancel{c}(\tau) + (1 - \alpha)\lambda \leq (1 + \varepsilon_r)\left[\cancel{c}(\tau^*) + (1 - \alpha^*)\lambda\right] + \varepsilon_a$, where $\tau^*$ is a $\lambda$-ROT and $\alpha^* = \mathrm{M}(\tau^*)$. When $\varepsilon_r = 0$, we refer to any $(0, \varepsilon_a)$-approximate $\alpha$-partial transport plan or any $(0, \varepsilon_a)$-approximate $\lambda$-robust transport plan as $\varepsilon_a$-*close*. We finally define approximate partial transport plans that also allow for some slack in the amount of mass transported. Define a partial transport plan $\tau$ as an $(\varepsilon_r, \varepsilon_a, \delta)$-*approximate* $\alpha$-*partial transport plan* if (1) $\mathrm{M}(\tau) \geq (1 - \delta)\alpha$, and (2) $\cancel{c}(\tau) \leq (1 + \varepsilon_r)\cancel{c}(\tau^*) + \varepsilon_a$, where $\tau^*$ is an optimal $\alpha$-partial transport plan. The first condition says that the partial transport plan $\tau$ routes at least $(1 - \delta)\alpha$ mass, while the second condition says that the partial transport plan is comparable in cost to the best possible $\alpha$-partial plan.

**Approximation algorithm.** We now describe an algorithm $\mathcal{A}_{PT}$ for partial OT using a $\lambda$-robust OT solver. Let $\alpha \in (0, 1)$ and $\varepsilon_r, \varepsilon_a \geq 0$ be parameters, and let $0 < \delta \leq \alpha \varepsilon_r^{-1}$ be some constant. Suppose we are given an $(\varepsilon_r, \varepsilon_a)$-approximation algorithm $\mathcal{A}_{RT}$ for the $\lambda$-SDROT problem that besides returning an $(\varepsilon_r, \varepsilon_a)$-approximate $\lambda$-robust transport plan $\tau_{\mathcal{A}}(\lambda)$ also returns the amount of mass $\tau_{\mathcal{A}}(\lambda)$ transports. We perform a binary search on $\lambda$ to compute an approximate $\alpha$-OPT plan.

First initialize values $\lambda_L = \frac{1}{3} \min_{b_1, b_2 \in B} \mathrm{d}(b_1, b_2)$ and $\lambda_R = \left(1 - \frac{\varepsilon_r}{\delta\alpha}\right)^{-1} \Delta + \frac{\varepsilon_a}{\delta\alpha}$, where $\Delta$ is the diameter of $A \cup B$ and is finite since it is assumed that $\mu$ has compact support. Run algorithm $\mathcal{A}_{RT}$ with inputs $\lambda_L$ and $\lambda_R$, obtaining approximate partial transport plans $\tau_L$ and $\tau_R$, respectively. If $\mathrm{M}(\tau_R) < \alpha$, we send the untrasported mass in a greedy manner and return the resulting transport plan (As we will see below, if $\alpha$ is close to 1 then $\mathcal{A}_{RT}$ might transport less than $\alpha$ mass even for $\lambda = \Delta$). If $\tau_L$ routes more than $\alpha$ mass, then one can greedily route all $\alpha$ mass within a tiny ball of radius at most $\lambda_L$ around each point of the discrete distribution. Otherwise, we perform a simple binary search on $\lambda$. Set $\lambda = \frac{1}{2}(\lambda_L + \lambda_R)$ and use algorithm $\mathcal{A}_{RT}$ to compute an approximate partial transport plan $\tau_\lambda$ given input $\lambda$. If $\mathrm{M}(\tau_\lambda) \geq \alpha$, then assign $\lambda_R \leftarrow \lambda$ and $\tau_R \leftarrow \tau_\lambda$. Otherwise, assign $\lambda_L \leftarrow \lambda$ and $\tau_L \leftarrow \tau_\lambda$. This maintains the invariant $\mathrm{M}(\tau_L) \leq \alpha \leq \mathrm{M}(\tau_R)$ while decreasing the gap between $\lambda_L$ and $\lambda_R$. The algorithm stops when $\lambda_R \leq (1 + \varepsilon_r)\lambda_L + \varepsilon_a$. If $\mathrm{M}(\tau_R) > \alpha$, we compute an $\alpha$-partial transport plan $\widetilde{\tau}_\alpha$ from $\tau_R$ by reducing the mass being transported in a greedy manner, i.e. not sending the mass to the farthest points of $A$. See the appendix for details.

**Pathological example.** Before we analyze the performance of this algorithm, we remark that one cannot guarantee Lemmas 2.1 and 2.2 to hold for an approximate partial transport plan. As a

consequence, one cannot immediately use the nice structure of Lemma 2.3 to guarantee that the above binary search returns a near-optimal $\alpha$-OPT. Roughly speaking, suppose $\lambda^* = \mathrm{MinCap}(\alpha)$. Even if we set $\lambda = (1 + \varepsilon)\lambda^*$, an $(\varepsilon, 0)$-approximate $\lambda$-ROT solver may not transport the mass at distance close to $\lambda^*$ and thus it may return a transport plan $\tau_{\mathcal{A},\lambda}$ with $\mathrm{M}(\tau_{\mathcal{A},\lambda}) < \alpha$. The problem becomes more acute as $\alpha$ approaches 1, as $\mathrm{M}(\tau_{\mathcal{A},\lambda}) < \alpha$ even for $\lambda > \Delta$. For instance, consider the following simple example where $A = \{a\}, B = \{b\}, \mathrm{d}(a, b) = 1$ and $\alpha = 1$. Then a transport plan $\tau_\lambda$ with $\mathrm{M}(\tau_\lambda) = 0$ is an $(\varepsilon, 0)$-approximate $\lambda$-robust OT plan for $\lambda = 1 + \varepsilon$. Furthermore, for $\lambda = 10$ a transport plan $\tau_\lambda$ with $\mathrm{M}(\tau_\lambda) = 1 - \frac{\varepsilon}{9}$ is an $(\varepsilon, 0)$-approximate $\lambda$-robust OT plan (as its cost is $10 \cdot \frac{\varepsilon}{9} + (1 - \frac{\varepsilon}{9}) = 1 + \varepsilon$). We, however, show that the binary search returns a transport plan that has an additive error in the cost or that guarantees to transport at least $(1 - \delta)\alpha$ mass in a near-optimal manner.

**Analysis.** The example above highlights the key phenomenon that allows us to provide a guarantee on the quality of the transport plan returned by the above algorithm: as $\lambda$ increases, the maximum amount of mass that is not transported by an approximate $\lambda$-robust transport plan $\tau$ decreases. We make this phenomenon precise in the following lemma.

**Lemma 3.1.** *Suppose $\tau$ is an arbitrary $(\varepsilon_r, \varepsilon_a)$-approximate $\lambda$-robust transport plan and $\delta \in (0, 1)$.*

*(i) For any $\alpha \in (0, 1)$, if $\mathrm{M}(\tau) \leq (1 - \delta)\alpha$, then $\lambda \leq (1 - \frac{\varepsilon_r}{\delta\alpha})^{-1}\mathrm{MinCap}(\alpha) + \frac{\varepsilon_a}{\delta\alpha}$.*

*(ii) Equivalently, for any $\alpha \in (0, 1 - \delta)$, if $\mathrm{M}(\tau) \leq \alpha$, then $\lambda \leq \left(1 - \frac{\varepsilon_r}{\delta\alpha}\right)^{-1} \mathrm{MinCap}\left(\frac{\alpha}{1-\delta}\right) + \frac{\varepsilon_a}{\delta\alpha}$.*

The following lemma proves that when the algorithm terminates with interval $[\lambda_L, \lambda_R]$, the most expensive mass of $\tau_R$ has a cost comparable to $\lambda_R$.

**Lemma 3.2.** *If $\lambda_R \leq (1 + \varepsilon_r)\lambda_L + \varepsilon_a$, then $\not{c}(\tau_R) - \not{c}(\tau_L) \geq [\mathrm{M}(\tau_R) - \mathrm{M}(\tau_L)]\lambda_R - (4\varepsilon_r\lambda_R + 4\varepsilon_a)$.*

Using Lemma 3.2, we prove that $\mathcal{A}_{PT}$ is also an approximation algorithm for the $\alpha$-OPT problem.

**Theorem 3.3.** *Let $\mu$ be a continuous distribution with compact support $A \subset \mathbb{R}^d$, and let $\nu$ be a discrete distribution with support $B \subseteq \mathbb{R}^d$ and minimum pairwise distance 1. Define $\Delta$ to be the diameter of $A \cup B$ and let $\alpha \in (0, 1)$ be a parameter. Suppose $\varepsilon_r, \varepsilon_a \geq 0$ and $\mathcal{A}_{RT}$ is an arbitrary $(\varepsilon_r, \varepsilon_a)$-approximation algorithm for the $\lambda$-robust OT problem. For any $\delta \in \left(0, \min\left\{\frac{4\varepsilon_r}{\alpha}, \frac{1-\alpha}{\alpha}, \frac{1}{2}\right\}\right)$, the algorithm $\mathcal{A}_{PT}$ is an $(\varepsilon_r, \eta)$-approximation algorithm for $\alpha$-partial transport, where $\eta = O\left(\frac{\varepsilon_r}{\delta\alpha}(\varepsilon_r\mathrm{MinCap}(\alpha(1 + \delta)) + \varepsilon_a)\right)$. For $\varepsilon_r = 0$, $\mathcal{A}_{PT}$ is an $(0, 5\varepsilon_a)$-approximation algorithm. The algorithm $\mathcal{A}_{PT}$ makes $O(\log \Delta)$ calls to $\mathcal{A}_{RT}$.*

The challenge for the approximate solver arises when some mass needs to be transported much farther than the cost of the transport plan (which is the expected distance of mass being transported), as in cases where $\mu$ has a long tail and $\alpha$ is such that some mass from this tail has to be transported. If $\alpha$ is in the range for which the cost of an $\alpha$-OPT plan $\tau_\alpha$ is comparable to the maximum distance the mass is transported in $\tau_{\alpha+\varepsilon_r}$, a stronger claim can be made on the quality of the plan returned by $\mathcal{A}_{PT}$.

**Corollary 3.4.** *Let $\mu$ be a continuous distribution with compact support $A \subset \mathbb{R}^d$, and let $\nu$ be a discrete distribution with support $B \subseteq \mathbb{R}^d$. Suppose $\varepsilon_r, \varepsilon_a \geq 0$, $\alpha \in (0, 1 - \varepsilon_r)$ is a parameter, and $\mathcal{A}_{RT}$ is an arbitrary $(\varepsilon_r, \varepsilon_a)$-approximation algorithm for the $\lambda$-robust OT problem. Additionally, suppose $\not{c}(\tau^*) \geq c \cdot \mathrm{MinCap}(\alpha + \varepsilon_r)$ for some constant $c \in (0, 1)$, where $\tau^*$ is an $\alpha$-OPT plan. Then, $\mathcal{A}_{PT}$ is an $\left(O(\frac{\varepsilon_r}{c}), O(\varepsilon_a)\right)$-approximation algorithm for $\alpha$-partial transport.*

Finally, if we allow some flexibility in the amount of mass transported by the returned partial transport plan, then one can still obtain a similar cost approximation guarantee for arbitrary distributions by running the above binary search but with input being $\alpha(1 - \delta)$.

**Theorem 3.5.** *Let $\mu$ be a continuous distribution with compact support $A \subset \mathbb{R}^d$, $\nu$ a discrete distribution with support $B \subseteq \mathbb{R}^d$, and $\alpha \in (0, 1)$ a parameter. Suppose $\varepsilon_r, \varepsilon_a \geq 0$, $\varepsilon_r < \frac{1}{5}\alpha$, and $\delta > \frac{5\varepsilon_r}{\alpha}$. Then algorithm $\mathcal{A}_{PT}$ is a $(\varepsilon_r, 5\varepsilon_a, \delta)$-approximation algorithm for $\alpha$-partial transport given any arbitrary $(\varepsilon_r, \varepsilon_a)$-approximation algorithm $\mathcal{A}_{RT}$ for the $\lambda$-robust OT problem.*

## 4 A Highly Accurate Algorithm for the Robust Optimal Transport Problem

In this section, we present an algorithm that computes an $\varepsilon$-close $\lambda$-robust transport plan, for $\lambda, \varepsilon > 0$, between a continuous distribution $\mu$ and a discrete distribution $\nu$ in $\tilde{O}(n^4 \log(\Delta/\varepsilon))$ time in $\mathbb{R}^2$ and in $n^{O(d)} \log(\Delta/\varepsilon)$ time in $\mathbb{R}^d$, where $\Delta$ is the diameter of $A \cup B$.

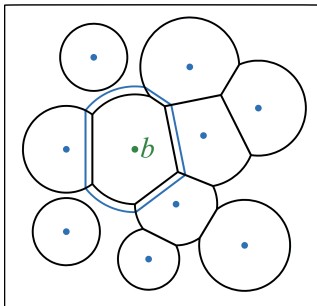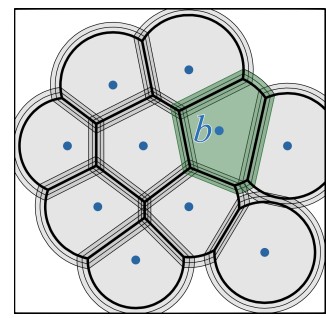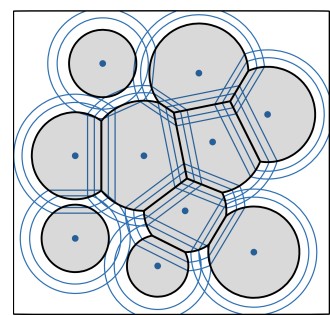

Figure 2: (left) The $\delta$-expanded restricted Voronoi cell of $RV_y^\delta(b)$, (middle) in a $\delta$-feasible transport plan, the point $b \in B$ can transport mass to its $2\delta$-expanded restricted Voronoi cell, (right) the decomposition $\mathcal{X}_\delta$.

## 4.1 Our Combinatorial Framework

We begin by introducing the notion of a $\delta$-feasible transport plan. Given a weight function $y(\cdot)$ on points of $B$ and a parameter $\delta \geq 0$, the $\delta$-*expanded restricted Voronoi cell* of $b$ is defined as

$$RV_y^\delta(b) := \{a \in A \mid \mathrm{d}(a,b) \leq y(b) + \delta \text{ and } \mathrm{d}_y(a,b) \leq \mathrm{d}_y(a,b') + \delta, \forall b' \in B\}.$$

See Figure 2(left). Any transport plan $\tau$ along with weights $y(\cdot)$ is $\delta$-*feasible* if, for any pair $(a,b) \in A \times B$ with $\tau(a,b) > 0$, we have $a \in RV_y^{2\delta}(b)$, i.e., each point $b \in B$ transports mass only to the regions within the $2\delta$-expansions of its restricted Voronoi cell (Figure 2(middle)). We refer to any point $b \in B$ (resp. $a \in A$) as *surplus* (resp. *deficit*) if $\int_A \tau(a,b)\,da < \nu(b)$ (resp. $\sum_{b \in B} \tau(a,b) < \mu(a)$) and as balanced if $\int_A \tau(a,b)\,da = \nu(b)$ (resp. $\sum_{b \in B} \tau(a,b) = \mu(a)$).

Given a weight function $y(\cdot)$ for the points in $B$, we derive a weight $\bar{y}(a)$ for each point $a \in A$ as

$$\bar{y}(a) := \max\{0, \max_{b \in B}\{y(b) - \mathrm{d}(a,b)\}\}. \tag{2}$$

Note that any $0$-feasible transport plan transports the mass of each point $b \in B$ to regions inside its restricted Voronoi cell. From Lemma 2.2, a $\lambda$-ROT plan is a $0$-feasible transport plan where $\mathrm{Cap}(y) = \lambda$, all surplus points $b \in B$ have $y(b) = \lambda$, and all deficit points $a \in A$ have $\bar{y}(a) = 0$. The following lemma shows that any $\delta$-feasible transport plan with similar restrictions on surplus and deficit points is a $2\delta$-close $\lambda$-robust transport plan.

**Lemma 4.1.** *For any parameter $\delta > 0$, suppose $\tau, y(\cdot)$ denotes a $\delta$-feasible transport plan, and let $\lambda := \mathrm{Cap}(y)$. Suppose the following two conditions hold: (F1) for every deficit point $a \in A$, $\bar{y}(a) = 0$, and (F2) for all surplus points $b \in B$, $y(b) = \lambda$. Then, $\tau$ is a $2\delta$-close $\lambda$-robust transport plan.*

**Challenges in extending cost-scaling technique.** Our algorithm builds on the cost-scaling framework of Agarwal *et al.* [1], which operates over multiple scales. Their algorithm begins at a coarse scale $\delta = \Delta$ with all weights $y(b) = 0$. In each scale $\delta$, weights are inherited from the previous scale $2\delta$. The algorithm discretizes the continuous support $A$ using an arrangement of $0$-, $\delta$-, and $2\delta$-expansions of all Voronoi cells and iteratively computes a $\delta$-feasible transport plan by adjusting the weights and increasing the amount of transported mass using augmenting paths. When the plan transports all the mass, the resulting weights are passed on to the next scale $\delta/2$. Their algorithm terminates when $\delta = \varepsilon/2$ and returns an $\varepsilon$-close transport plan.

Extending their approach to our setting creates new challenges. Unlike in the standard setting, our algorithm computes a $\delta$-feasible transport plan that may leave some mass untransported. We must, therefore, cap the weights assigned to each point $b \in B$ by $\lambda$ (Lemma 4.1 condition (F2)) and restrict the weights $\bar{y}(a)$ of all deficit regions in $A$ to $0$ (Lemma 4.1 condition (F1)). A naïve implementation of the scaling paradigm risks generating deficit regions within the restricted Voronoi cells – deficit points of $A$ with positive weights – violating Lemma 4.1. For example, in Figure 3, the algorithm creates a deficit region (green region) inside the restricted Voronoi cells, illustrating such a violation. To resolve this, we introduce "consolidating paths", which carefully restructure the transport plan to relocate deficit regions from inside the restricted Voronoi cells to their exterior (Figure 3(right)).

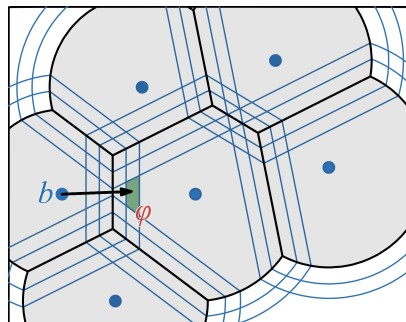 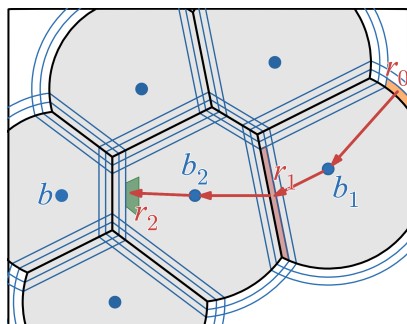

Figure 3: (left) In scale $2\delta$, the point $b$ transports mass to a region $\varphi$ inside its $4\delta$-expanded Voronoi cell, and (right) in scale $\delta$, the $\varphi$ becomes a violating deficit region. A consolidating path $\langle r_0, b_1, r_1, b_2, r_2 \rangle$ can be used to bring this deficit to the exterior of the restricted Voronoi cells.

Each scale of our algorithm consists of two steps. Step 1 updates the weights and the transport plan along a set of consolidating paths to restore condition (F1). Step 2 of our algorithm maintains $\delta$-feasibility and condition (F1) as invariants, and carefully raises the weight of the surplus points to $\lambda$, enforcing condition (F2). From Lemma 4.1, the resulting transport plan is $2\delta$-close. We provide a non-trivial proof that these adjustments do not impact the asymptotic execution time of our algorithm.

### 4.2 Our Algorithm

We begin by describing the basic terminology required to describe our algorithm.

**Decomposition of** $A$**.** Given a $\delta$-feasible transport plan $\tau, y(\cdot)$, let $\mathcal{X}_\delta$ denote the arrangement of the restricted Voronoi cell $RV_y(b)$ and the $\delta$- and $2\delta$-expanded restricted Voronoi cells $RV_y^\delta(b)$ and $RV_y^{2\delta}(b)$, for all points $b \in B$, namely, the subdivision of $\mathbb{R}^d$ induced by overlaying these cells. See Figure 2(right). For each region $\varphi \in \mathcal{X}_\delta$, pick an arbitrary *representative point* $r_\varphi$ inside $\varphi$ and assign it a mass of $\mu_{r_\varphi} := \mu(\varphi) = \int_\varphi \mu(a)\,da$. Let $A_\delta$ denote the set of representative points of all regions in $\mathcal{X}_\delta$. We refer to $r_\varphi \in A_\delta$ as a deficit point if the corresponding region $\varphi \in \mathcal{X}_\delta$ is deficit. The point $r_\varphi$ is *violating* if it is deficit and $\bar{y}(r_\varphi) > 0$.

In our algorithm, we store and represent a $\delta$-feasible semi-discrete transport plan $\tau, y(\cdot)$ using a (discrete) transport plan $\hat\tau$ over $A_\delta \times B$, where for any point $b \in B$ and any region $\varphi \in \mathcal{X}_\delta$, we set $\hat\tau(r_\varphi, b) := \tau(\varphi, b)$. Note that any $\delta$-feasible transport plan $\hat\tau, y(\cdot)$ over $A_\delta \times B$ can be converted back to a $\delta$-feasible semi-discrete transport plan $\tau$ that arbitrarily transports a mass of $\hat\tau(r_\varphi, b)$ from $b$ to $\varphi$ for any pair $(\varphi, b) \in \mathcal{X}_\delta \times B$. We refer to $\hat\tau$ as an implicit representation of $\tau$.

**Residual Graph.** Given a $\delta$-feasible transport plan $\tau, y(\cdot)$, define a residual graph $\mathcal{G}_\delta$ over the point set $A_\delta \cup B$, where for each pair $(r_\varphi, b) \in A_\delta \times B$, we add a *forward edge* directed from $b$ to $r_\varphi$ if $r_\varphi \in RV_y^{2\delta}(b)$. Additionally, if $\tau(\varphi, b) > 0$, we add a *backward edge* directed from $r_\varphi$ to $b$. For any triple $(b_1, r_\varphi, b_2)$ formed by a forward edge followed by a backward edge, the triple $(b_1, r_\varphi, b_2)$ is *admissible* if $d_y(r_\varphi, b_1) < d_y(r_\varphi, b_2)$. Any directed path $P$ in the residual graph is an *alternating path*. An alternating path $P$ is admissible if all triples $(b_i, r_i, b_{i+1}) \in P$ are admissible. We *update* $\hat\tau$ along $P$ with a mass of $\beta$ by increasing (resp. decreasing) the mass transported along each forward (resp. backward) edge of $P$ by $\beta$. When $P$ is a path from a surplus point $b \in B$ to a deficit point $r_\varphi \in A_\delta$, we refer to $P$ as an *augmenting path* and to updating $\hat\tau$ along $P$ as the *augment* process.

**Consolidating paths.** Any alternating path $P = \langle r_0, b_1, r_1, \ldots, b_k, r_k \rangle$ in the residual graph is called a *consolidating path* if $r_0$ is a point with $\bar{y}(r_0) = 0$ and $r_k$ is a deficit point with $\bar{y}(r_k) > 0$, i.e., $P$ is a path from a point outside of all $2\delta$-expanded cells to a violating deficit point. The consolidating path $P$ is admissible if all triples $(b_i, r_i, b_{i+1}) \in P$ are admissible. Note that updating $\hat\tau$ along $P$ with a mass of $\beta$ increases the total mass transported to the violating deficit point $r_k$ by $\beta$ (potentially making it balanced) and decreases the amount of mass transported to the zero-weight point $r_0$ by $\beta$.

**The scaling algorithm.** Our algorithm works in $O(\log(\Delta\varepsilon^{-1}))$ scales, where $\Delta$ is the diameter of $A \cup B$. Each scale is associated with a parameter $\delta > 0$ and computes a $\delta$-feasible $\lambda$-robust transport plan. Our algorithm maintains a set of weights $y(\cdot)$ for $B$ and uses a few subroutines that

are described in the appendix. At the beginning of the first scale, set $\delta = \Delta$ and $y(b) = 0$ for all $b \in B$. Execute the following steps while $\delta > \varepsilon/2$.

1. *Removing violating deficit points:* For any point $b \in B$, set $\tau_\delta(a,b) := \tau_{2\delta}(a,b)$ for each point $a \in RV_y^{2\delta}(b)$ and $\tau_\delta(a,b) = 0$ for any point $a \notin RV_y^{2\delta}(b)$. Compute $\mathcal{G}_\delta$ and $\hat{\tau}_\delta$ with respect to $\tau_\delta, y(\cdot)$. While there exists a violating deficit point $r \in A_\delta$:

   (i) Execute the SEARCHANDCONSOLIDATE procedure, which computes a set of admissible augmenting and consolidating paths in $\mathcal{G}_\delta$ and updates $\hat{\tau}_\delta$ along these paths. At the end of this step, there are no admissible augmenting and consolidating paths to the violating deficit points in the residual graph.

   (ii) Execute the REDUCEWEIGHTS procedure, which reduces the weights of all points of $B$ that have admissible paths to the violating deficit points in $A_\delta$ by $\delta$ and recomputes $\mathcal{G}_\delta$ and $\hat{\tau}_\delta$.

   (iii) Execute the ACYCLIFY procedure, which updates $\hat{\tau}_\delta$ and $\mathcal{G}_\delta$ so that the transport plan $\hat{\tau}_\delta$ is a forest and the residual graph does not have any admissible cycles.

2. *Raising surplus weights to $\lambda$:* Set all points $b \in B$ with $y(b) < \lambda$ as active. While there are active surplus points in $B$:

   (i) Execute the SEARCHANDAUGMENT procedure, which computes a set of admissible augmenting paths and admissible alternating paths from surplus active points to inactive points in $\mathcal{G}_\delta$ and updates $\hat{\tau}_\delta$ along these paths. At the end of this step, there are no admissible augmenting paths in the residual graph.

   (ii) Execute the INCREASEWEIGHTS procedure, which increases the weights of all active points of $B$ which have admissible paths from the active surplus points by $\delta$, marks any point $b \in B$ with $y(b) = \lambda$ as inactive, and recomputes $\mathcal{G}_\delta$ and $\hat{\tau}_\delta$.

   (iii) Execute the ACYCLIFY procedure, which updates $\hat{\tau}_\delta$ and $\mathcal{G}_\delta$ so that the transport plan $\hat{\tau}_\delta$ is a forest and the residual graph does not have any admissible cycles.

3. *Scale Update:* Set $\delta \leftarrow \delta/2$.

The SEARCHANDAUGMENT , INCREASEWEIGHTS , and ACYCLIFY procedures are straightforward adaptations of the procedures outlined in [1] and similarly maintain the $\delta$-feasibility conditions. The following lemma states the properties of the two new procedures.

**Lemma 4.2.** *During the execution of the* SEARCHANDCONSOLIDATE *and* REDUCEWEIGHTS *procedure, the transport plan remains $\delta$-feasible, each balanced point $b \in B$ remains balanced, the weight of each point $b \in B$ containing deficit regions inside $RV_y(b)$ decreases by $\delta$, and the weight of each surplus point remains unchanged.*

## 4.3 Analysis

**Correctness.** In each scale $\delta$, our algorithm begins with a $\delta$-feasible transport plan, restores the condition (F1) in the step 1, and restores (F2) while maintaining the $\delta$-feasibility condition and (F1) in step 2. Since both (F1) and (F2) are satisfied, from Lemma 4.1, the computed transport plan at the end of each scale $\delta$ is a $2\delta$-close $\lambda$-ROT plan.

**Efficiency.** As shown in the appendix, the residual graph contains $O(n^2)$ nodes and $O(n^3)$ edges in $\mathbb{R}^2$. Using this, we show that each iteration of steps 1 and 2 of our algorithm takes $O(n^2(\Phi + n \log n))$ time, where $\Phi$ denotes the query time of an oracle that computes the continuous mass of $\mu$ inside a query triangle. We also show that both steps run $O(n)$ iterations. Summing over all $O(\log(\Delta/\varepsilon))$ scales, we get a total running time of $O(n^3(\Phi + n \log n) \log(\Delta/\varepsilon))$ for our algorithm, as claimed. In the remainder of this section, we provide a sketch of the proof of the number of iterations.

Suppose $y_{2\delta}(\cdot)$ denotes the set of weights of $B$ maintained by our algorithm at the end of scale $2\delta$. For any iteration $i$ of step 1, let $\tau^i, y^i(\cdot)$ denote the $\delta$-feasible transport plan maintained by our algorithm after the $i$th iteration and define $\gamma^i(b) := y_{2\delta}(b) - y^i(b)$.

**Lemma 4.3.** *For any subset $S \subset B$, suppose in an iteration $i$ of step 1 of our algorithm, the reduction in the weight of any point in $S$ is more than $6\delta$ greater than the reduction in the weight of any point in $B \setminus S$, i.e., $\min_{b \in S} \gamma^i(b) > \max_{b' \in B \setminus S} \gamma^i(b') + 6\delta$. Then, there are no deficit regions inside the restricted Voronoi cells of the points in $S$.*

| $n$ | Error | Iterations | | path and cycle lengths | | Regions |
|---|---|---|---|---|---|---|
| | | Step 1 | Step 2 | Step 1 | Step 2 | |
| 10 | 0.009 | 1 | 4 | 41 | 361 | 198 |
| 20 | 0.018 | 2 | 4 | 320 | 991 | 529 |
| 30 | 0.021 | 3 | 5 | 688 | 1719 | 862 |
| 40 | 0.015 | 3 | 5 | 979 | 2385 | 1212 |
| 50 | 0.018 | 4 | 5 | 1603 | 3061 | 1544 |
| 60 | 0.015 | 5 | 6 | 1973 | 3673 | 1855 |
| 70 | 0.017 | 5 | 6 | 2512 | 4129 | 2130 |
| 80 | 0.016 | 6 | 7 | 2845 | 4785 | 2421 |

Table 1: The results of our experiments empirically verifying our theoretical claims.

Assume that for some iteration $i > 6n$, there still remains some violating deficit regions. Using Lemma 4.2, we show that for each surplus point $b_s \in B$, the weight of $b_s$ remains unchanged during step 1 (i.e., $\gamma^i(b_s) = 0$) and for each point $b \in B$ with deficit regions in inside $RV_y(b)$, the weight of $b$ decreases by $\delta$ in all iterations (i.e., $\gamma^i(b) = i\delta$). Therefore, for $i > 6n$, one can compute a set $S$ where $b \in S$ for all points $b \in B$ with deficit regions in $RV_y(b)$, $b_s \notin S$ for each surplus point $b_s \in B$, and that satisfies the conditions of Lemma 4.3. However, from Lemma 4.3, there are no deficit regions in the restricted Voronoi cells of points in $S$, which is a contradiction to our assumption. Hence, the number of iterations of step 1 is $O(n)$. Using a similar approach, we show that the number of iterations of step 2 of our algorithm is also $O(n)$. We conclude with the following theorem.

**Theorem 4.4.** *Let $\mu$ be a continuous distribution with compact support $A \subset \mathbb{R}^d$, $\nu$ a discrete distribution with support of $n$ points $B \subseteq \mathbb{R}^d$, and $\lambda > 0$ a parameter. Suppose there exists an oracle that returns the mass of $\mu$ inside any query triangle in $\Phi$ time. Then, there is an algorithm that computes an $\varepsilon$-close $\lambda$-robust optimal transport plan in $n^{O(d)}\Phi \log \Delta/\varepsilon$ time, where $\Delta$ is the diameter of $A \cup B$.*

**Experimental evaluation.** We implemented a prototype of our algorithm in Python to empirically verify its performance. The implementation omits two efficiency-critical components: (i) the computation of continuous mass within regions defined by restricted Voronoi cells, which we approximate using a fine uniform grid over the domain by summing the mass of grid points lying inside each region, and (ii) the dynamic tree data structure, which we simulate by naïvely iterating over all edges due to the lack of efficient Python implementations. We evaluated our implementation on a Gaussian distribution with mean $[0.5, 0.5]$ and covariance $0.15I$ bounded within the unit square, and added $10\%$ noise to the bottom-left corner by mixing with an exponential distribution with rate parameter 3. The discrete distribution consists of $n$ samples drawn from the same Gaussian with an additional $10\%$ noise sampled from an exponential distribution with rate parameter 3 in the top-right corner. We fixed $\lambda = 0.2$ and $\varepsilon = 0.02$, and varied $n$ from 10 to 80. For each value of $n$, we ran the program 10 times and reported the averaged statistics in Table 1. We observe that as $n$ increases, the additive error remains within the target threshold $\varepsilon = 0.02$. Moreover, the number of iterations in the two main steps of the algorithm consistently stays below the theoretical upper bounds ($6n$ and $12n$; see Lemma 4.3 and E.9). Finally, the total lengths of paths and cycles computed in these steps grow sub-quadratically in $n$, matching the asymptotic behavior predicted by our complexity analysis (Section 4.3). Additional experimental results are provided in Section B for the sake of space. The code is available at https://github.com/pouyansh/Efficient_Partial_and_Robust_SDOT.

## 5 Discussion

In this paper, we present new characterizations of the semi-discrete partial and robust OT problems, and design approximation algorithms based on these formulations. Our techniques are stated for $|\cdot|_p^p$ costs and extend to any cost function whose weighted Voronoi diagram has bounded complexity. The presented algorithms assume access to an oracle that, for a polyhedron $P$ of constant complexity, computes the continuous mass contained in $P$ in $\Phi$ time. For many natural distributions—such as those with sample access or simple closed-form densities—one can typically assume $\Phi = O(\text{poly} \log n)$. Achieving polynomial dependence on $d$ and $1/\varepsilon$ remains an interesting direction for future work.

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

## A  Fast Approximation Algorithms for Robust Optimal Transport

In this appendix, we describe algorithms that compute an $(\varepsilon_r, 0)$-approximate $\lambda$-robust transport plan between a discrete distribution $\nu$ and continuous distribution $\mu$ in near-linear time with respect to the support size of the discrete distribution $\nu$. We keep the description of the algorithm at a high level, and describe technical details in Appendix F. Our low-precision algorithms can be summarized with the following high-level sequence of steps.

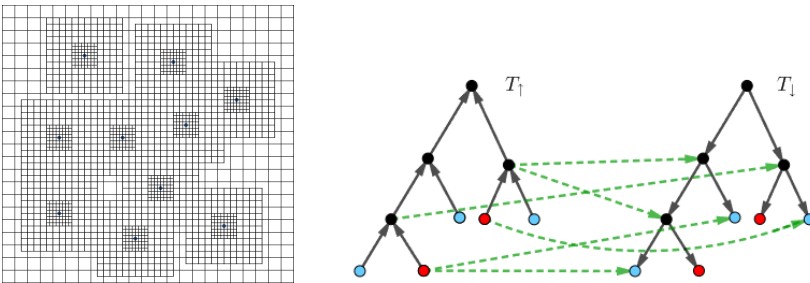

Figure 4: (left) Cover of $A$ by hypercubes, and (right) graph formed by two trees with cross edges

1. **Discretize continuous distribution.** First, we decompose the continuous domain $A$ into a near-linear size set $\mathcal{P}$ of hypercubes (see Figure 4). Each hypercube $\square$ is guaranteed to satisfy one of two properties: (1) $\square$ is within distance $\varepsilon \min_{b' \neq b} ||b' - b||$ to one of the input points $b \in B$ and has sidelength roughly $\varepsilon^2 \min_{b' \neq b} ||b' - b||$, or (2) the sidelength of $\square$ is at most roughly $\varepsilon$ times the minimum distance from its center to every point in $B$. We query the mass of each hypercube and contract this mass to a single point. If the hypercube is very close to a point of $B$, we contract its mass to $b$. Otherwise, we contract the mass to its center. This induces a discrete distribution $\hat{\mu}$ whose support $\hat{B}$ also has near linear size.

2. **Reduce to $\lambda$-capped transport.** Mukherjee et al. [34] showed that for the discrete optimal transport problem, computing a $\lambda$-robust OT plan is equivalent to computing a complete transport plan under a capped distance function $\hat{d}^\lambda(a,b) := \min\{d(a,b), \lambda\}$, which we denote as a $\lambda$-*capped transport plan*. We compute a $(\varepsilon, 0)$-approximate $\lambda$-capped transport plan $\hat{\tau}$ between $\hat{\mu}$ and $\nu$ using some discrete OT solver and discard all mass transported at cost more than $\lambda$. The best choice of near-linear time algorithm for $\lambda$-capped OT depends on the choice of cost function d; we provide efficient algorithms under squared Euclidean distance (Section F.3) and $\ell_p$ metrics (Section F.4).

   (a) **Discrete $\lambda$-capped $p$-Wasserstein.** We construct an $\varepsilon$-well separated pair decomposition ($\varepsilon$-WSPD) [13] to approximate the set of $n^2$ edges with a set of $O(n\varepsilon^{-d})$ pairs of regions in $\mathbb{R}^d$. We then construct two quadtrees, an up-tree $T_\uparrow$ and a down-tree $T_\downarrow$, and a set of cross edges connecting each WSPD region in the up-tree to each WSPD region in the down-tree (see Figure 4). This graph satisfies the property that every $a$ in the support of $\hat{\mu}$ has a unique directed path to each $b$ in $B$. We cap the edge costs in a natural manner and guarantee the path length from $a$ to $b$ is at most $(1 + \varepsilon)\hat{d}^\lambda(a,b)$, then run a minimum-cost flow algorithm [16] on the directed graph and shortcut the resulting flow using standard techniques.

   (b) **Faster $\lambda$-capped $1$-Wasserstein.** A modification to the graph construction above allows for faster, specialized machinery for the mininum-cost flow problem via the multiplicative weights method [47]. First, we employ a random shift on the quadtree to incur bounded distortion in expectation. Instead of making a cross edge between each WSPD pair, we take each WSPD pair and add a cross edge between the lowest ancestors whose sidelengths are at least $\frac{\varepsilon}{\log \Delta}$ times their tree distance. Then cap the edge costs as above. One can argue in standard fashion that the shortest path metric $d_G$ satisfies $\mathbb{E}[d_G(a,b)] \leq (1 + \varepsilon)\hat{d}^\lambda(a,b)$ for all $a, b$ in the supports of $\hat{\mu}$ and $\nu$. We make all edges undirected and run the multiplicative weights method.

   Unlike prior works on multiplicative weight update algorithms for the geometric transportation problem [3, 22, 28], our new approach uses a well-separated pair decomposition to inform what cross edges are good. This type of technique is necessary if one wants to simultaneously (1) bound the number of edges by $n\varepsilon^{O(d)} \log \Delta$, (2) approximate the capped Euclidean distance with a shortest path metric, and (3) argue the shortest path metric embeds into a tree metric with deterministic $O(\log \Delta)$ distortion.

3. **Extract semi-discrete plan.** Finally, we design a simple scheme to disperse the discrete transport plan from the contracted point throughout its corresponding region. The scheme disperses $\hat{\tau}(c_\square, b)$ uniformly throughout each hypercube $\square$ that is farther than distance $\varepsilon \min_{b' \neq b} ||b' - b||$ from $b$, where $c_\square$ denotes its center. By conditions (1) and (2) on the set of hypercubes, we argue that every point $a \in \square$ is approximately equidistant from $b$, and therefore any arbitrary dispersion gives comparable cost. Additionally for each hypercube $\square$ within distance $\varepsilon \min_{b' \neq b} ||b' - b||$ from $b$, we greedily assign the (approximate) closest $\hat{\tau}(b, b)$ mass from $\mu$ within a small ball of $b$ to $b$. The remainder is uniformly distributed among the remaining $b' \in B$.

We make the following conclusion about the algorithm.

**Theorem A.1.** *Let $\mu$ be a continuous distribution with compact support $A \subset \mathbb{R}^d$, $\nu$ be a discrete distribution with support $B \subseteq \mathbb{R}^d$, $\lambda > 0$ be a parameter and $d, p \geq 1$ be constants. Suppose $Q_1$ is the time complexity to compute $\int_\square \mu(a)da$ for any hypercube $\square \subseteq A$ and $Q_2$ is the time complexity to compute, given a point $b \in \mathbb{R}^d$ and constant $c \geq 0$, the radius $r \geq 0$ for which the Euclidean ball $\mathcal{B}(b, r)$ of radius $r$ centered at $b$ satisfies $\int_{\mathcal{B}(b,r)} \mu(a)da = c$. Then a $(\varepsilon, 0)$-approximate $\lambda$-robust semi-discrete transport plan can be computed in $n\varepsilon^{-2d} \log \varepsilon^{-1} \left(n^{o(1)}\varepsilon^{-2d} \log \Delta + Q_1 + Q_2\right)$ time if $\mathrm{d}(a, b) = ||a - b||^p$ and in $O\left(n\varepsilon^{-2d} \log \varepsilon^{-1} \left(\varepsilon^{-2d-4} \log^3 \Delta \log n + Q_1 + Q_2\right)\right)$ time with probability at least $\frac{1}{2}$ if $\mathrm{d}(a, b) = ||a - b||$.*

For simplicity of exposition in the introduction, we assumed $Q_1, Q_2 = O(\log \Delta)$. If one can represent the pdf function $\mu$ in a way that allows for fast integration over balls and hypercubes, then $Q_1, Q_2 = O(\log \Delta)$ amortized complexity holds. Some examples of such distributions include histograms and Guassian mixtures.

## A.1 Comparison with prior work

Before describing the algorithm in full detail, we make some remarks about our algorithm in comparison with existing work for semi-discrete OT and more generally the geometric transportation problem. The structure of our algorithm has many similarities with existing algorithms for full semi-discrete OT [3, 46]: we first discretize the Euclidean space by a dense sampling of the continuous distribution with hypercubes, then run a fast discrete OT solver using this geometric sampling, and finally disperse the mass from the discrete OT plan throughout each hypercube. There are a few major bottlenecks when using their algorithms for $\lambda$-capped OT that we overcome in this paper.

First, the algorithms of [3, 46] as described only work for $p = 1$. To reduce semi-discrete $\lambda$-robust $p$-Wasserstein to the discrete $\lambda$-robust $p$-Wasserstein problem for $p > 1$, one cannot simply perform a greedy routing of the mass within an $\varepsilon$ ball of each point $b \in B$ as done in [3, 46]. This is because the triangle inequality does not hold for $p > 1$, and therefore one cannot charge an $\varepsilon$ error to the greedy routing. We take a slightly different approach, where one greedily routes mass within the $\varepsilon$ ball of a point $b$ only if the discrete transport plan routes mass from $b$ to its $\varepsilon$ ball. This change is sufficient to construct an $(\varepsilon, 0)$-approximation for $\lambda$-capped $p$-Wasserstein distances when $p > 1$, as proved in Appendix F.5.

Second, existing algorithms for the discrete geometric transport problem (e.g. [2, 3, 22, 28]) do not consider $\lambda$-capped distances. Approximating this distance with a shortest path metric in a graph is nontrivial. If one chooses to cap edge costs to guarantee a particular path has length at most $\lambda$, then it is necessary to guarantee that other paths in the graph are not shortened by too much. If too many edges in the graph have their costs capped, then extraneous paths in the graph may develop undesirably low costs. To circumvent this challenge, we design geometric graphs whose shortest paths provably have a bounded number of edges with capped edge costs. This difference is highlighted in Appendices F.3 and F.4.

# B Additional Experimental Results

We first provide figures for our experiments described in Section 4.3, showing the shape of the transport plan as well as the untransported parts of the mass of the distributions. We then present results using the Beta distribution.

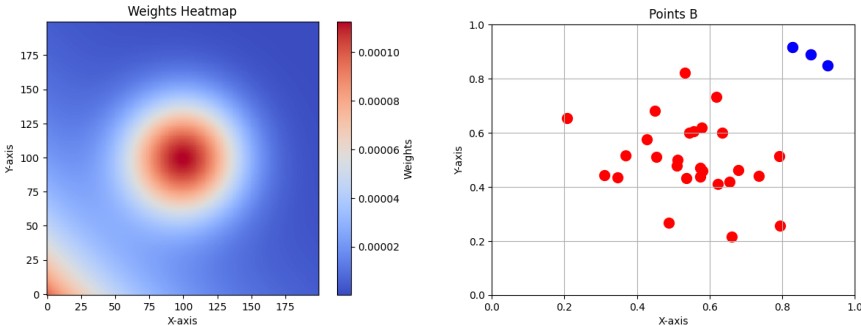

Figure 5: (left) The continuous distribution, and (right) the discrete distribution.

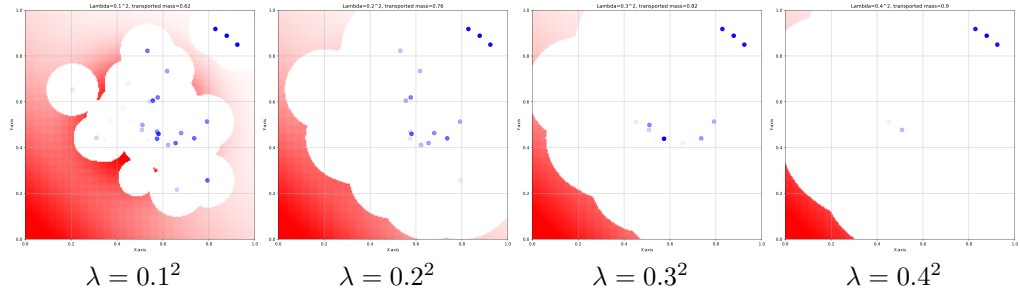

| $\lambda = 0.1^2$ | $\lambda = 0.2^2$ | $\lambda = 0.3^2$ | $\lambda = 0.4^2$ |

Figure 6: The parts of the mass of the distributions that are not transported by the $\lambda$-ROT plan computed by our algorithm.

## B.1 Experiments on Gaussian Distribution

As described in Section 4.3, in our experiments, we used a continuous distribution as a Gaussian distribution bounded inside the unit square that is contaminated with $10\%$ noise in the bottom left corner (See Figure 5 (left)). Our discrete support consists of $0.9n$ samples from the same Gaussian distribution (the red points in Figure 5 (right)) along with $0.1n$ samples from an exponential distribution centered at the top right corner of the square (the blue points in Figure 5 (right)), each assigned a mass of $1/n$.

Figure 6 shows the parts of the mass of the continuous and discrete distributions that are not transported by the $\lambda$-ROT plans computed by our algorithm, as the value of $\lambda$ increases from $0.1^2$ to $0.4^2$. Here, the ground distance is the squared Euclidean distance. We observe that by increasing the value of $\lambda$, the $\lambda$-ROT plan transports the majority of the inlier mass of the distributions and leaves the noise parts of the mass untransported.

We extended our experiments to 3 dimensions using a similar framework and constructed continuous and discrete distributions using the Gaussian distribution inside the unit cube with $10\%$ noise mass added to the different corners of the cube for continuous and discrete distributions. The average statistics of our experiments on different values of $n$ are shown in Table 2.

## B.2 Results using a Mixture of Beta Distributions

In this experiment, we use mixtures of Beta distributions that induce skewed and dispersed mass. Specifically, the discrete distribution consists of $n$ samples drawn from a Beta($\alpha = 2, \beta$) distribution, with $\beta$ varying in the range $[3, 10]$. As $\beta$ increases, the distribution becomes increasingly concentrated near $[0, 0]$. The continuous distribution is constructed by contaminating the same base distribution with $15\%$ noise drawn from a Beta($\alpha = 5, \beta = 2$) distribution (a distribution concentrated close to $[0.9, 0.9]$). For smaller values of $\beta$, the two components overlap significantly; for larger values, they become well-separated. Each experiment was repeated 10 times, and we report the averaged results.

Table 3 shows the costs and the percentage of inliers and outliers that are transported by a $0.15$-ROT plan computed by our algorithm. We observe that as $\beta$ increases (and the inlier and outlier parts

| $n$ | Error | Iterations | | path and cycle lengths | | Regions |
|---|---|---|---|---|---|---|
| | | Step 1 | Step 2 | Step 1 | Step 2 | |
| 10 | 0.002 | 2 | 5 | 25 | 454 | 389 |
| 20 | 0.016 | 5 | 6 | 303 | 2514 | 1776 |
| 30 | 0.013 | 5 | 6 | 500 | 4664 | 3292 |
| 40 | 0.016 | 11 | 8 | 1587 | 7057 | 4807 |
| 50 | 0.028 | 12 | 11 | 2820 | 10954 | 6681 |
| 60 | 0.024 | 16 | 12 | 3644 | 13141 | 8072 |
| 70 | 0.020 | 15 | 10 | 4735 | 15310 | 9008 |

Table 2: The results of our experiments in 3 dimensions.

| $\beta$ | Cost | Robust cost | % Inliers transported | % Outliers transported | Total mass transported |
|---|---|---|---|---|---|
| 3 | 0.022 | 0.041 | 0.980 | 0.493 | 0.907 |
| 4 | 0.020 | 0.041 | 0.992 | 0.350 | 0.896 |
| 5 | 0.019 | 0.042 | 0.995 | 0.245 | 0.883 |
| 6 | 0.017 | 0.041 | 0.998 | 0.191 | 0.877 |
| 7 | 0.015 | 0.042 | 0.998 | 0.091 | 0.862 |
| 8 | 0.013 | 0.041 | 0.999 | 0.085 | 0.862 |
| 9 | 0.012 | 0.040 | 0.999 | 0.053 | 0.857 |
| 10 | 0.011 | 0.040 | 1.000 | 0.048 | 0.857 |

Table 3: The results of our experiments on mixtures of Beta distribution.

become more distant from each other), the amount of outlier mass transported by the $0.15$-ROT plan computed by our algorithm decreases, while such transport plans transport almost all of the inlier mass for all values of $\beta$.

These experiments confirm that our algorithm remains reliable even when inliers and outliers overlap. In particular, the computed partial and robust transport plans consistently focus on the inlier mass while effectively disregarding much of the outlier mass, especially as the main and noise distributions become more dissimilar. Performance metrics (shown in Table 4) remain consistent with those observed in the Gaussian case.

## C   Missing Proofs and Details from Section 2

**Lemma 2.1** ($\alpha$-OPT Characterization). *Let $\mu$ be a continuous distribution over compact support $A \subset \mathbb{R}^d$, $\nu$ a discrete distribution over a set $B$ of $n$ points in $\mathbb{R}^d$, and $\alpha \in [0, 1]$ a parameter. There exists a valid weight function $y : B \to \mathbb{R}_{\geq 0}$ with $\mathrm{Cover}(y) = \alpha$ such that the transport plan $\tau_y$ induced by $y$ is an $\alpha$-OPT plan. Furthermore, this property holds for any valid weight function $y$ with $\mathrm{Cover}(y) = \alpha$.*

*Proof.* We rewrite the primal linear optimization problem for the $\alpha$-partial transport problem:

$$\min_{\tau} \sum_{b \in B} \int_A \tau(a, b) \mathrm{d}(a, b) da$$

$$\text{s.t. } \int_A \tau(a, b) da \leq \nu(b) \quad \forall b \in B,$$

$$\sum_{b \in B} \tau(a, b) \leq \mu(a) \quad \forall a \in A,$$

$$\sum_{b \in B} \int_A \tau(a, b) da = \alpha,$$

$$\tau \geq 0.$$

| $\beta$ | Error | Iterations | | path and cycle lengths | | Regions |
|---|---|---|---|---|---|---|
| | | Step 1 | Step 2 | Step 1 | Step 2 | |
| 3 | 0.012 | 4 | 6 | 420 | 993 | 495 |
| 4 | 0.016 | 7 | 8 | 332 | 1026 | 495 |
| 5 | 0.005 | 6 | 7 | 190 | 986 | 462 |
| 6 | 0.007 | 6 | 8 | 131 | 971 | 435 |
| 7 | 0.010 | 9 | 8 | 229 | 886 | 421 |
| 8 | 0.008 | 3 | 6 | 161 | 810 | 382 |
| 9 | 0.007 | 6 | 8 | 89 | 849 | 388 |
| 10 | 0.007 | 3 | 5 | 55 | 766 | 351 |

Table 4: The performance of our algorithm in our experiments on mixtures of Beta distribution.

Then its corresponding dual linear optimization problem is

$$\max_{\lambda,\phi,\psi} \ \alpha\lambda - \sum_{b\in B}\phi(b)\nu(b) - \int_A \psi(a)\mu(a)da \tag{3}$$
$$\text{s.t.} \ \lambda - \phi(b) - \psi(a) \le \mathrm{d}(a,b) \quad \forall a \in A, b \in B,$$
$$\phi,\psi \ge 0.$$

Note for any fixed value of $\lambda$ and $\phi$, the objective is maximized when $-\psi$ is as large as possible. The largest feasible $-\psi$ one can choose is $-\psi(a) = \min\{\min_{b\in B}(\mathrm{d}(a,b) - \lambda + \phi(b)), 0\}$ for all $a \in A$. After making this substitution, using the definition of the restricted Voronoi cells, we obtain the following optimization problem

$$\max_{\lambda,\phi} \ \alpha\lambda + \sum_{b\in B}\left[\int_{RV_{\lambda-\phi}(b)} \min\left\{\min_{b\in B}(\mathrm{d}(a,b) - \lambda + \phi(b)), 0\right\}\mu(a)da - \phi(b)\nu(b)\right]$$
$$\text{s.t.} \ \phi \ge 0.$$

This problem is concave and unconstrained on $\lambda$; therefore, it is maximized when $0 \in \partial D_\phi(\lambda)$, where $D_\phi \colon \mathbb{R} \to \mathbb{R}$ is the above dual objective for fixed values of $\phi$ and $\partial D_\phi$ denotes the superderivative of $D_\phi$. For values of $\phi$ where $D_\phi$ is smooth, we note

$$\frac{dD_\phi}{d\lambda}(\lambda) = \alpha - \sum_{b\in B}\mu(RV_{\lambda-\phi}(b)) = \alpha - \mathrm{Cover}(\lambda - \phi).$$

Therefore, $\frac{dD_\phi}{d\lambda}(\lambda) = 0$ if $\mathrm{Cover}(\lambda - \phi) = \alpha$. Similar observation extends this to the case when $D_\phi$ is not smooth, where one gets $0 \in \partial D_\phi(y)$ if $\mathrm{Cover}((\lambda - \delta) - \phi) \le \alpha \le \mathrm{Cover}((\lambda + \delta) - \phi)$ for all $\delta > 0$. For choice of weights $y(b) = \lambda - \phi(b)$ we conclude $\mathrm{Cover}(y) = \alpha$ as desired.

Now suppose we make the substitution $y(b) = \lambda - \phi(b)$ into the original dual linear optimization problem (3) to obtain the equivalent dual optimization problem

$$\max_{\lambda,y,\psi} \ \sum_{b\in B}y(b)\nu(b) - \int_A \psi(a)\mu(a)da - (1-\alpha)\lambda$$
$$\text{s.t.} \ y(b) - \psi(a) \le \mathrm{d}(a,b) \quad \forall a \in A, b \in B,$$
$$y(b) \le \lambda \quad \forall b \in B,$$
$$\psi \ge 0.$$

By complementary slackness, for any optimal primal solution $\tau$ and dual solution $y, \psi, \lambda$, if $\tau(a,b) > 0$ for any $a, b$, then $y(b) - \psi(a) = \mathrm{d}(a,b)$. But the objective is minimized when $-\psi(a) = \min\{\min_{b\in B}(\mathrm{d}(a,b) - y(b)), 0\}$, as observed above. Hence, we conclude that if $\tau(a,b) > 0$ then $a \in RV_y(b)$. Similarly, if $\int_A \tau(a,b)da < \nu(b)$ then $\phi(b) = 0$ or equivalently $y(b) := \lambda - \phi(b) = \lambda$. But by feasibility of the optimal dual solution, we note that $\lambda \ge y(b')$ for all $b' \in B$. Therefore, if $\int_A \tau(a,b)da < \nu(b)$ then $y(b) = \max_{b'\in B} y(b')$.

This proves that if $y$ exists, then $y$ satisfies the desired properties. We then note the optimal solutions to both primal and dual linear optimization problems exist by strong duality.

We finally show that for any $y$ where $\mathrm{Cover}(y) = \alpha$ and $y(b) = \max_{b' \in B} y(b')$ for all free $b \in B$, the transport plan $\tau$ induced by $y$ is an optimal $\alpha$-transport plan. Let $y$ be some arbitrary set of weights that satisfy these properties. Define a weight function $\psi : A \to \mathbb{R}$ that assigns $\psi(a) := \max_{b \in B} y(b) - \mathrm{d}(a, b)$ to each point $a \in A$. Additionally define $\nu_f^\tau(b) := \nu(b) - \int_A \tau(a, b) da$ and $\mu_f^\tau(a) := \mu(a) - \sum_{b \in B} \tau(a, b)$ as the free mass of $b$ and $a$ with respect to transport plan $\tau$, respectively. We can rewrite the cost of $\tau$ as:

$$\cent(\tau) = \sum_{b \in B} \int_A \mathrm{d}(a, b)\, \tau(a, b)\, da = \sum_{b \in B} \int_A (y(b) - \psi(a))\tau(a, b)\, da$$

$$= \sum_{b \in B} y(b)\nu(b) - \int_A \psi(a)\, d\mu(a) - \sum_{b \in B_f} y(b)\nu_f^\tau(b) + \int_{A_f} \psi(a)\, d\mu_f^\tau(a)$$

$$= \sum_{b \in B} y(b)\nu(b) - \int_A \psi(a)\, d\mu(a) - (1 - \alpha)y_{\max},$$

where $y_{\max} := \max_{b \in B} y(b)$. Let $\tau^*$ denote any optimal $\alpha$-partial transport plan between $\mu$ and $\nu$. Then, by the definition of dual weights for points in $A$,

$$\cent(\tau^*) = \sum_{b \in B} \int_A \mathrm{d}(a, b)\, \tau^*(a, b)\, da \geq \sum_{b \in B} \int_A (y(b) - \psi(a))\tau^*(a, b)\, da$$

$$= \sum_{b \in B} y(b)\nu(b) - \int_A \psi(a)\, d\mu(a) - \sum_{b \in B_f} y(b)\nu_f^{\tau^*}(b) + \int_{A_f} \psi(a)\, d\mu_f^{\tau^*}(a)$$

$$\geq \sum_{b \in B} y(b)\nu(b) - \int_A \psi(a)\, d\mu(a) - (1 - \alpha)y_{\max},$$

where the last inequality holds since $\psi(a) \geq 0$ and $y(b) \leq y_{\max}$. Combining the two bounds,

$$\cent(\tau) = \sum_{b \in B} y(b)\nu(b) - \int_A \psi(a)\, d\mu(a) - (1 - \alpha)y_{\max} \leq \cent(\tau^*).$$

Since $\tau^*$ is an $\alpha$-partial OT plan, we conclude that $\tau$ is also an $\alpha$-partial OT plan. $\qquad\square$

**Lemma 2.2** ($\lambda$-ROT Characterization)**.** *Let $\mu$ be a continuous distribution over compact support $A \subset \mathbb{R}^d$, $\nu$ a discrete distribution over a set $B$ in $n$ points in $\mathbb{R}^d$, and $\lambda > 0$ a parameter. There exists a valid weight function $y : B \to \mathbb{R}_{\geq 0}$ with $\mathrm{Cap}(y) = \lambda$ such that the transport plan induced by $y$ is a $\lambda$-ROT plan. Furthermore, this property holds for any valid weight function $y$ with $\mathrm{Cap}(y) = \lambda$.*

*Proof.* The primal linear optimization problem for $\lambda$-robust optimal transport is as follows.

$$\min_{\tau} \sum_{b \in B} \int_A \tau(a, b)\mathrm{d}(a, b)\, da + \lambda \left(1 - \sum_{b \in B} \int_A \tau(a, b)\, da\right)$$

$$\text{s.t.} \int_A \tau(a, b) da \leq \nu(b) \quad \forall b \in B,$$

$$\sum_{b \in B} \tau(a, b) \leq \mu(a) \quad \forall a \in A,$$

$$\tau \geq 0.$$

The corresponding dual linear optimization problem is

$$\max_{\phi, \psi} -\sum_{b \in B} \phi(b)\nu(b) - \int_A \psi(a)\mu(a) da$$

$$\text{s.t.} -\phi(b) - \psi(a) \leq \mathrm{d}(a, b) - \lambda \quad \forall a \in A, b \in B,$$

$$\phi, \psi \geq 0.$$

Define $y(b) := \lambda - \phi(b)$. Then, since $\lambda$ is a given parameter and the total mass at the points of $B$ is 1, the dual problem can be rewritten as

$$\max_{y,\psi} \quad \sum_{b \in B} y(b)\nu(b) - \int_A \psi(a)\mu(a)da - \lambda$$
$$\text{s.t.} \quad y(b) - \psi(a) \le \mathrm{d}(a,b) \quad \forall a \in A, b \in B,$$
$$y(b) \le \lambda, \quad \forall b \in B,$$
$$\psi \ge 0.$$

By complementary slackness, we observe that if $\int_A \tau(a,b)da < \nu(b)$ for some $b \in B$ then $y(b) = \lambda$. Then one would have $y(b) = \max_{b' \in B} y(b')$ and $\mathrm{Cap}(y) := \max_{b \in B} y(b) = \lambda$ as desired by dual feasibility constraint $y(b') \le \lambda$ for all $b' \in B$. Otherwise if $\int_A \tau(a,b)da = \nu(b)$ for all $b \in B$, then let $y_{\max} := \max_{b \in B} y(b)$ and define the new dual weights $y'(b) := y(b) + (\lambda - y_{\max})$, $\psi'(a) := \psi(a) + (\lambda - y_{\max})$. It follows that $y', \psi'$ are feasible and have an equal dual objective value since both $\mu$ and $\nu$ are probability distributions. Furthermore, $\mathrm{Cap}(y') := \max_{b \in B} y'(b) = \lambda$, as desired.

Additionally note that for any choice of weights $y(\cdot)$ for $B$, the optimal choice for the $\psi$ function would be $\psi(a) := \max(0, \max_{b \in B} y(b) - \mathrm{d}(a,b))$ for each point $a \in A$. By complementary slackness, for any pair $(a,b) \in A \times B$ with $\tau(a,b) > 0$, we have $y(b) - \psi(a) = \mathrm{d}(a,b)$. Therefore, the point $a$ has to be inside $RV_y(b)$.

We argue by strong duality that choice of $y$ optimizing this dual problem exists. Therefore, we conclude that there exists a valid weight function satisfying the desired properties that induces a $\lambda$-robust OT plan.

We finally show that for any $y$ where $\mathrm{Cap}(y) = \lambda$ and $y(b) = \max_{b' \in B} y(b')$ for all free $b \in B$, the transport plan $\tau$ induced by $y$ is a $\lambda$-robust optimal transport plan. Let $y$ be some arbitrary set of weights that satisfy these properties. Define a weight function $\psi : A \to \mathbb{R}$ that assigns $\psi(a) := \max\{0, \max_{b \in B} y(b) - \mathrm{d}(a,b)\}$ to each point $a \in A$. Let $\alpha = \sum_{b \in B} \int_A \tau(a,b)da$. Additionally define $\nu_f^\tau(b) := \nu(b) - \int_A \tau(a,b)da$ and $\mu_f^\tau(a) := \mu(a) - \sum_{b \in B} \tau(a,b)$ as the free mass of $b$ and $a$ with respect to transport plan $\tau$, respectively. Using the assumption that $\tau$ is induced by $y$, we can rewrite the robust cost of $\tau$ as:

$$w_\lambda(\tau) = \sum_{b \in B} \int_A \mathrm{d}(a,b)\, \tau(a,b)\, da + \lambda \left(1 - \sum_{b \in B} \int_A \tau(a,b)da\right)$$
$$= \sum_{b \in B} \int_A (y(b) - \psi(a))\tau(a,b)\, da + \lambda\,(1 - \alpha)$$
$$= \sum_{b \in B} y(b)\nu(b) - \int_A \psi(a)\, d\mu(a) - \sum_{b \in B_f} y(b)\nu_f^\tau(b) + \int_{A_f} \psi(a)\, d\mu_f^\tau(a) + \lambda(1 - \alpha)$$
$$= \sum_{b \in B} y(b)\nu(b) - \int_A \psi(a)\, d\mu(a) - \lambda(1 - \alpha) - \lambda(1 - \alpha)$$
$$= \sum_{b \in B} y(b)\nu(b) - \int_A \psi(a)\, d\mu(a).$$

Let $\tau^*$ denote any optimal $\lambda$-robust optimal transport plan between $\mu$ and $\nu$. Define $\alpha^* = \sum_{b \in B} \int_A \tau^*(a,b)da$. Then, by the definition of dual weights for points in $A$,

$$w_\lambda(\tau^*) = \sum_{b \in B} \int_A \mathrm{d}(a,b)\, \tau^*(a,b)\, da + \lambda \left(1 - \sum_{b \in B} \int_A \tau(a,b)da\right)$$
$$\ge \sum_{b \in B} \int_A (y(b) - \psi(a))\tau^*(a,b)\, da + \lambda(1 - \alpha^*)$$
$$= \sum_{b \in B} y(b)\nu(b) - \int_A \psi(a)\, d\mu(a) - \sum_{b \in B_f} y(b)\nu_f^{\tau^*}(b) + \int_{A_f} \psi(a)\, d\mu_f^{\tau^*}(a) + \lambda(1 - \alpha^*)$$

$$\geq \sum_{b \in B} y(b)\nu(b) - \int_A \psi(a)\, d\mu(a) - \lambda(1 - \alpha^*) + \lambda(1 - \alpha^*)$$

$$= \sum_{b \in B} y(b)\nu(b) - \int_A \psi(a)\, d\mu(a),$$

where the second to last line above holds since $\psi(a) \geq 0$ and $y(b) \leq y_{\max}$. Combining the two bounds,

$$w_\lambda(\tau) = \sum_{b \in B} y(b)\nu(b) - \int_A \psi(a)\, d\mu(a) \leq w_\lambda(\tau^*).$$

Since $\tau^*$ is a $\lambda$-robust OT plan, we conclude that $\tau$ is also a $\lambda$-robust OT plan. $\qquad\square$

**Lemma 2.3.** *The mappings* $\mathrm{MaxCover}\colon \mathbb{R}_{\geq 0} \to [0,1]$ *and* $\mathrm{MinCap}\colon [0,1] \to \mathbb{R}_{\geq 0}$ *are monotonically non-decreasing functions.*

*Proof.* Consider any two values $0 < \lambda_1 < \lambda_2$. Define $\alpha_1 := \mathrm{MaxCover}(\lambda_1)$ (resp. $\alpha_2 := \mathrm{MaxCover}(\lambda_2)$) and let $\tau_1$ (resp. $\tau_2$) denote the corresponding $\alpha_1$-robust (resp. $\alpha_2$-robust) OT plan. Since $\tau_1$ is an optimal $\lambda_1$-robust OT plan,

$$\mathcal{C}(\tau_1) + (1 - \alpha_1)\lambda_1 \leq \mathcal{C}(\tau_2) + (1 - \alpha_2)\lambda_1.$$

Consequently,

$$\mathcal{C}(\tau_1) - \mathcal{C}(\tau_2) \leq (\alpha_1 - \alpha_2)\lambda_1. \tag{4}$$

Similarly, since $\tau_2$ is an optimal $\lambda_2$-robust OT plan,

$$\mathcal{C}(\tau_2) + (1 - \alpha_2)\lambda_2 \leq \mathcal{C}(\tau_1) + (1 - \alpha_1)\lambda_2.$$

Therefore,

$$\mathcal{C}(\tau_1) - \mathcal{C}(\tau_2) \geq (\alpha_1 - \alpha_2)\lambda_2. \tag{5}$$

Combining Equations (4) and (5),

$$(\alpha_1 - \alpha_2)(\lambda_1 - \lambda_2) \geq 0.$$

In words, since $\lambda_1 < \lambda_2$, then $\alpha_1 \leq \alpha_2$. This proves $\mathrm{MaxCover}$ is an increasing function.

We now show $\mathrm{MinCap}$ is an increasing function. Let $0 \leq \alpha_1 < \alpha_2 \leq 1$ be two constants. Define $\lambda_1 := \mathrm{MinCap}(\alpha_1)$ and $\lambda_2 := \mathrm{MinCap}(\alpha_2)$. Let $y_i$ denote the corresponding $\lambda_i$-capped weight functions for each $\alpha_i$, $i \in \{1, 2\}$ known to exist by definition of $\mathrm{MinCap}$. Additionally let $\tau_i$ denote the corresponding $\lambda_i$-robust transport plans for $i \in \{1, 2\}$ induced by weights $y_i$ by Lemma 2.2. Then we analogously conclude by Lemma 2.2 that each $\tau_i$ is an optimal $\lambda_i$-robust transport plan, and therefore $\mathcal{C}(\tau_1) - \mathcal{C}(\tau_2) \leq (\alpha_1 - \alpha_2)\lambda_1$ and $\mathcal{C}(\tau_1) - \mathcal{C}(\tau_2) \geq (\alpha_1 - \alpha_2)\lambda_2$. Combining these inequalities again gives $(\alpha_1 - \alpha_2)(\lambda_1 - \lambda_2) \geq 0$, Therefore if $\alpha_1 < \alpha_2$, then $\mathrm{MinCap}(\alpha_1) \leq \mathrm{MinCap}(\alpha_2)$. $\qquad\square$

# D   Missing Proofs and Details from Section 3

We give the subroutine to return a transport plan with precisely $\alpha$ mass in this section, then allocate the rest of the section to proving various lemmas and theorems from the main text.

## D.1   Routing Exactly $\alpha$ Mass

In the binary search procedure as described from the main text, we obtain two transport plans $\tau_L$ and $\tau_R$, which satisfy $\mathrm{M}(\tau_L) \leq \alpha$ and $\mathrm{M}(\tau_R) \geq \alpha$. From these two transport plans, we require a simple procedure to return precisely $\alpha$ mass, as mentioned in the main text.

We assume it is possible to return a linear combination of two transport plans. Then return the linear interpolation $\tau_{\mathcal{A}'} = (1 - c)\tau_R + c\tau_L$ for the constant $c \geq 0$ which guarantees $\tau_{PT}$ transports $\alpha$ mass. The choice of $c := \frac{\mathrm{M}(\tau_R) - \alpha}{\mathrm{M}(\tau_R) - \mathrm{M}(\tau_L)}$ only depends on $\mathrm{M}(\tau_L)$, $\mathrm{M}(\tau_R)$ and $\alpha$, and assumes $\mathrm{M}(\tau_L) \leq \alpha$ and $\mathrm{M}(\tau_R) \geq \alpha$. This invariant is indeed maintained by construction of the binary search procedure.

We note that if one possesses stronger oracles on the distributions $\tau_{\lambda_R}$ and $\mu$, then it is possible to greedily remove mass from $\tau_R$ until $\tau_R$ transports $\alpha$ mass while guaranteeing comparable precision. Let $\hat{\tau}_r$ denote the transport plan defined by $\hat{\tau}_r(a,b) = \tau_R(a,b)$ for all $a,b$ such that $d(a,b) \le r$ and $\hat{\tau}_r(a,b) = 0$ otherwise. If one can compute the value of $r^* \ge 0$ where $\hat{\tau}_{r^*}$ routes precisely $\alpha$ mass, then we can return $\hat{\tau}_{r^*}$ and guarantee comparable approximation.

## D.2 Proofs of Lemmas

**Lemma 3.1.** *Suppose $\tau$ is an arbitrary $(\varepsilon_r, \varepsilon_a)$-approximate $\lambda$-robust transport plan and $\delta \in (0,1)$.*

*(i) For any $\alpha \in (0,1)$, if $M(\tau) \le (1-\delta)\alpha$, then $\lambda \le (1 - \frac{\varepsilon_r}{\delta\alpha})^{-1} \mathrm{MinCap}(\alpha) + \frac{\varepsilon_a}{\delta\alpha}$.*

*(ii) Equivalently, for any $\alpha \in (0, 1-\delta)$, if $M(\tau) \le \alpha$, then $\lambda \le \left(1 - \frac{\varepsilon_r}{\delta\alpha}\right)^{-1} \mathrm{MinCap}\left(\frac{\alpha}{1-\delta}\right) + \frac{\varepsilon_a}{\delta\alpha}$.*

*Proof.* We prove result (i). Then (ii) follows from substituting $\alpha' = \frac{\alpha}{1-\delta}$ into the bound from (i) and noting $\frac{1}{1-\delta} > 1$.

Let $\alpha_\tau = \int_A \sum_{b \in B} \tau(a,b) da$ be the total mass routed by the partial transport plan $\tau$. Then since $\tau$ is a $(\varepsilon_r, \varepsilon_a)$-approximate $\lambda$-robust transport plan, we note that

$$\cent(\tau) + (1 - \alpha_\tau)\lambda \le (1 + \varepsilon_r)\left[\cent(\tau_\alpha^*) + (1-\alpha)\lambda\right] + \varepsilon_a$$

where $\tau_\alpha^*$ is an optimal $\alpha$-partial transport plan. To get a bound on the cost of transforming $(1+\varepsilon_r)^{-1}\tau$ into $\tau_\alpha^*$, we apply some algebra on the above inequality and observe

$$\cent(\tau_\alpha^*) - \frac{\cent(\tau)}{1+\varepsilon_r} \ge \left(\alpha - \frac{\alpha_\tau}{1+\varepsilon_r} - \frac{\varepsilon_r}{1+\varepsilon_r}\right)\lambda - \frac{\varepsilon_a}{1+\varepsilon_r}.$$

Now let $\tau_{\alpha_\tau}^*$ denote an optimal $\alpha_\tau$-partial transport plan. By optimality of $\tau_{\alpha_\tau}^*$, we first note

$$\frac{\cent(\tau_{\alpha_\tau}^*)}{1+\varepsilon_r} - \frac{\cent(\tau)}{1+\varepsilon_r} \le 0.$$

Then by Lemma 2.3 as well as optimality of $\tau_\alpha^*(a,b)$ and $\tau_{\alpha_\tau}^*(a,b)$, we observe that

$$\frac{\cent(\tau_\alpha^*)}{1+\varepsilon_r} - \frac{\cent(\tau_{\alpha_\tau}^*)}{1+\varepsilon_r} \le \left(\frac{\alpha}{1+\varepsilon_r} - \frac{\alpha_\tau}{1+\varepsilon_r}\right) \mathrm{MinCap}(\alpha).$$

Finally, by Lemma 2.1 we observe

$$\cent(\tau_\alpha^*) - \frac{\cent(\tau_\alpha^*)}{1+\varepsilon_r} \le \left(\alpha - \frac{\alpha}{1+\varepsilon_r}\right) \mathrm{MinCap}(\alpha).$$

Adding the three above inequalities gives us

$$\cent(\tau_\alpha^*) - \frac{\cent(\tau)}{1+\varepsilon_r} \le \left(\alpha - \frac{\alpha_\tau}{1+\varepsilon_r}\right) \mathrm{MinCap}(\alpha).$$

Hence,

$$\left(\alpha - \frac{\alpha_\tau}{1+\varepsilon_r} - \frac{\varepsilon_r}{1+\varepsilon_r}\right)\lambda - \frac{\varepsilon_a}{1+\varepsilon_r} \le \left(\alpha - \frac{\alpha_\tau}{1+\varepsilon_r}\right) \mathrm{MinCap}(\alpha).$$

The result follows from dividing this inequality by $\left(\alpha - \frac{\alpha_\tau}{1+\varepsilon_r}\right)$, which is bounded above by $\delta\alpha$ by assumption of the Lemma statement, and isolating $\lambda$. $\square$

**Lemma 3.2.** *If $\lambda_R \le (1+\varepsilon_r)\lambda_L + \varepsilon_a$, then $\cent(\tau_R) - \cent(\tau_L) \ge [M(\tau_R) - M(\tau_L)]\lambda_R - (4\varepsilon_r\lambda_R + 4\varepsilon_a)$.*

*Proof.* Since $\tau_{\lambda_L}$ is a $(\varepsilon_r, \varepsilon_a)$-approximate $\lambda_L$-robust OT plan, we observe

$$\cent(\tau_{\lambda_L}) + (1 - M(\tau_{\lambda_L}))\lambda_L \le (1+\varepsilon_r)\left[\cent(\tau_{\lambda_R}) + (1 - M(\tau_{\lambda_R}))\lambda_L\right] + \varepsilon_a.$$

Additionally, since $\tau_{\lambda_R}$ is a $(\varepsilon_r, \varepsilon_a)$-approximate $\lambda_R$-robust OT plan, we observe

$$\cent(\tau_{\lambda_R}) + (1 - M(\tau_{\lambda_R}))\lambda_R \le (1+\varepsilon_r)\lambda_R + \varepsilon_a,$$

and therefore $\rlap{/}{c}(\tau_{\lambda_R}) \leq (1 + \varepsilon_r)\lambda_R + \varepsilon_a$ since $\mathrm{M}(\tau_{\lambda_R}) \leq 1$. Conclude

$$
\begin{aligned}
\rlap{/}{c}(\tau_{\lambda_R}) - \rlap{/}{c}(\tau_{\lambda_L}) &\geq [\mathrm{M}(\tau_{\lambda_R}) - \mathrm{M}(\tau_{\lambda_L}) - (1 - \mathrm{M}(\tau_{\lambda_R}))\varepsilon_r]\,\lambda_L - \varepsilon_r \rlap{/}{c}(\tau_{\lambda_R}) - \varepsilon_a \\
&\geq [\mathrm{M}(\tau_{\lambda_R}) - \mathrm{M}(\tau_{\lambda_L}) - \varepsilon_r]\,\lambda_L - \varepsilon_r\,[(1 + \varepsilon_r)\lambda_R + \varepsilon_a] - \varepsilon_a \\
&\geq [\mathrm{M}(\tau_{\lambda_R}) - \mathrm{M}(\tau_{\lambda_L}) - \varepsilon_r]\,\lambda_L - 2\varepsilon_r\lambda_R - 2\varepsilon_a
\end{aligned}
$$

Now we use the assumption $\lambda_R \leq (1 + \varepsilon_r)\lambda_L + \varepsilon_a$, observation $-1 \leq -(1 + \varepsilon_r)^{-1}$, and bound $0 \leq \mathrm{M}(\tau_{\lambda_R}) - \mathrm{M}(\tau_{\lambda_L}) \leq 1$ to obtain the inequality

$$
\begin{aligned}
[(\mathrm{M}(\tau_{\lambda_R}) - \mathrm{M}(\tau_{\lambda_L})) - \varepsilon_r]\,\lambda_L &\geq (1 + \varepsilon_r)^{-1}\,[(\mathrm{M}(\tau_{\lambda_R}) - \mathrm{M}(\tau_{\lambda_L})) - \varepsilon_r]\,\lambda_R - (1 + \varepsilon_r)\varepsilon_a \\
&\geq [(\mathrm{M}(\tau_{\lambda_R}) - \mathrm{M}(\tau_{\lambda_L})) - 2\varepsilon_r]\,\lambda_R - 2\varepsilon_a.
\end{aligned}
$$

Combining inequalities gives the desired result.

$\qquad\square$

**Theorem 3.3.** *Let $\mu$ be a continuous distribution with compact support $A \subset \mathbb{R}^d$, and let $\nu$ be a discrete distribution with support $B \subseteq \mathbb{R}^d$ and minimum pairwise distance 1. Define $\Delta$ to be the diameter of $A \cup B$ and let $\alpha \in (0, 1)$ be a parameter. Suppose $\varepsilon_r, \varepsilon_a \geq 0$ and $\mathcal{A}_{RT}$ is an arbitrary $(\varepsilon_r, \varepsilon_a)$-approximation algorithm for the $\lambda$-robust OT problem. For any $\delta \in \left(0, \min\left\{\frac{4\varepsilon_r}{\alpha}, \frac{1-\alpha}{\alpha}, \frac{1}{2}\right\}\right)$, the algorithm $\mathcal{A}_{PT}$ is an $(\varepsilon_r, \eta)$-approximation algorithm for $\alpha$-partial transport, where $\eta = O\left(\frac{\varepsilon_r}{\delta\alpha}(\varepsilon_r \mathrm{MinCap}(\alpha(1 + \delta)) + \varepsilon_a)\right)$. For $\varepsilon_r = 0$, $\mathcal{A}_{PT}$ is an $(0, 5\varepsilon_a)$-approximation algorithm. The algorithm $\mathcal{A}_{PT}$ makes $O(\log \Delta)$ calls to $\mathcal{A}_{RT}$.*

*Proof.* By the condition of $\frac{\lambda_R}{\lambda_L} = O(\varepsilon^{-1}\Delta)$ initially and end criteria $\lambda_R \leq (1 + \varepsilon_r)\lambda_L + \varepsilon_a$, we know that $\mathcal{A}_{PT}$ terminates within $O(\log \Delta)$ iterations. Additionally by construction of the plan $\tau_{\mathcal{A}_{PT}}$, $\mathrm{M}(\tau_{\mathcal{A}_{AT}}) := \int_A \sum_{b \in B} \tau_{\mathcal{A}_{PT}}(a, b)da = \alpha$ as desired. Therefore, it suffices to bound the cost of $\tau_{\mathcal{A}_{PT}}$.

Since $\tau_{\lambda_R}$ is an $(\varepsilon_r, \varepsilon_a)$-approximate $\lambda_R$-robust OT plan, we observe

$$
\rlap{/}{c}(\tau_{\lambda_R}) + (1 - \mathrm{M}(\tau_{\lambda_R}))\lambda_R \leq (1 + \varepsilon_r)\,[\rlap{/}{c}(\tau^*) + (1 - \alpha)\lambda_R] + \varepsilon_a. \tag{6}
$$

Now we note that the precise choice of constant $c$ in algorithm $\mathcal{A}_{PT}$ is $c := \frac{\mathrm{M}(\tau_{\lambda_R}) - \alpha}{\mathrm{M}(\tau_{\lambda_R}) - \mathrm{M}(\tau_{\lambda_L})}$. Using the inequality from Lemma 3.2 and multiplying by $c \leq 1$, we observe that

$$
c\,(\rlap{/}{c}(\tau_{\lambda_L}) - \rlap{/}{c}(\tau_{\lambda_R})) - (\alpha - \mathrm{M}(\tau_{\lambda_R}))\lambda_R \leq 4\varepsilon_r\lambda_R + 4\varepsilon_a. \tag{7}
$$

By definition, we note $\tau_{\mathcal{A}_{PT}} = \tau_{\lambda_R} + c(\tau_{\lambda_L} - \tau_{\lambda_R})$. Adding Inequalities (6) and (7) gives

$$
\begin{aligned}
\rlap{/}{c}(\tau_{\mathcal{A}_{PT}}) + (1 - \alpha)\lambda_R &= \rlap{/}{c}(\tau_{\lambda_R}) + c\,(\rlap{/}{c}(\tau_{\lambda_L}) - \rlap{/}{c}(\tau_{\lambda_R})) + (1 - \alpha)\lambda_R \\
&\leq (1 + \varepsilon_r)\rlap{/}{c}(\tau^*) + [(1 + \varepsilon_r)(1 - \alpha) + 4\varepsilon_r]\,\lambda_R + 5\varepsilon_a \\
&= (1 + \varepsilon_r)\rlap{/}{c}(\tau^*) + [1 - \alpha - \varepsilon_r\alpha + 5\varepsilon_r]\,\lambda_R + 5\varepsilon_a.
\end{aligned}
$$

Subtract $(1 - \alpha)\lambda_R$ from both sides of the inequality to get

$$
\begin{aligned}
\rlap{/}{c}(\tau_{\mathcal{A}_{PT}}) &\leq (1 + \varepsilon_r) \cdot \rlap{/}{c}(\tau^*) + (5\varepsilon_r - \varepsilon_r\alpha)\,\lambda_R + 3\varepsilon_a \\
&\leq (1 + \varepsilon_r) \cdot \rlap{/}{c}(\tau^*) + 5\varepsilon_r\lambda_R + 5\varepsilon_a. \tag{8}
\end{aligned}
$$

This immediately implies the result when $\varepsilon_r = 0$. For the remainder of the proof, we assume $\varepsilon_r > 0$. Then using Lemma 3.1, we note that $\lambda_L \leq \left(1 - \frac{\varepsilon_r}{\delta\alpha}\right)^{-1}\mathrm{MinCap}\left(\alpha(1 - \delta)^{-1}\right) + \frac{\varepsilon_a}{\delta\alpha}$ for any $0 < \delta < 1$. If $\delta \leq \frac{1}{2}$, then we can use $1 \leq (1 - \delta)^{-1} \leq 1 + 2\delta$ to simplify

$$
\lambda_L \leq \left(1 - \frac{\varepsilon_r}{\delta\alpha}\right)^{-1}\mathrm{MinCap}\left(\alpha(1 + 2\delta)\right) + \frac{\varepsilon_a}{\delta\alpha}.
$$

If additionally $\delta \leq \frac{2\varepsilon_r}{\alpha}$, then we use this same trick to obtain

$$
\lambda_L \leq \left(1 + \frac{2\varepsilon_r}{\delta\alpha}\right)\mathrm{MinCap}\left(\alpha(1 + 2\delta)\right) + \frac{\varepsilon_a}{\delta\alpha}.
$$

Substituting into the condition $\lambda_R \leq (1 + \varepsilon_r)\lambda_L + \varepsilon_a$ gives

$$\lambda_R \leq (1 + \varepsilon_r)\left[\left(1 + \frac{2\varepsilon_r}{\delta\alpha}\right)\text{MinCap}\left(\alpha(1 + 2\delta)\right) + \frac{\varepsilon_a}{\delta\alpha}\right] + \varepsilon_a$$

$$\leq \left(1 + \frac{4\varepsilon_r}{\delta\alpha}\right)\text{MinCap}\left(\alpha(1 + 2\delta)\right) + \frac{3\varepsilon_a}{\delta\alpha}$$

$$\leq \frac{6\varepsilon_r}{\delta\alpha} \cdot \text{MinCap}\left(\alpha(1 + 2\delta)\right) + \frac{3\varepsilon_a}{\delta\alpha}.$$

We conclude the Theorem statement after substituting this bound on $\lambda_R$ into Inequality (8). $\qquad\square$

**Corollary 3.4.** *Let $\mu$ be a continuous distribution with compact support $A \subset \mathbb{R}^d$, and let $\nu$ be a discrete distribution with support $B \subseteq \mathbb{R}^d$. Suppose $\varepsilon_r, \varepsilon_a \geq 0$, $\alpha \in (0, 1 - \varepsilon_r)$ is a parameter, and $\mathcal{A}_{RT}$ is an arbitrary $(\varepsilon_r, \varepsilon_a)$-approximation algorithm for the $\lambda$-robust OT problem. Additionally, suppose $\cent(\tau^*) \geq c \cdot \text{MinCap}(\alpha + \varepsilon_r)$ for some constant $c \in (0, 1)$, where $\tau^*$ is an $\alpha$-OPT plan. Then, $\mathcal{A}_{PT}$ is an $\left(O(\frac{\varepsilon_r}{c}), O(\varepsilon_a)\right)$-approximation algorithm for $\alpha$-partial transport.*

*Proof.* By the proof of Theorem 3.3, we note

$$\cent(\tau_{\mathcal{A}_{PT}}) \leq (1 + \varepsilon_r) \cdot \cent(\tau^*) + 5\varepsilon_r\lambda_R + 5\varepsilon_a.$$

and

$$\lambda_R \leq \frac{6\varepsilon_r}{\delta\alpha} \cdot \text{MinCap}\left(\alpha(1 + 2\delta)\right) + \frac{3\varepsilon_a}{\delta\alpha}.$$

Substituting this bound on $\lambda_R$ gives

$$\cent(\tau_{\mathcal{A}_{PT}}) \leq (1 + \varepsilon_r) \cdot \cent(\tau^*) + 5\varepsilon_r\left[\frac{6\varepsilon_r}{\delta\alpha} \cdot \text{MinCap}\left(\alpha(1 + 2\delta)\right) + \frac{3\varepsilon_a}{\delta\alpha}\right] + 5\varepsilon_a$$

$$= (1 + \varepsilon_r) \cdot \cent(\tau^*) + \frac{30\varepsilon_r^2}{\delta\alpha} \cdot \text{MinCap}\left(\alpha(1 + 2\delta)\right) + \frac{15\varepsilon_r\varepsilon_a}{\delta\alpha} + 5\varepsilon_a.$$

Now choose $\delta = \frac{\varepsilon_r}{2\alpha}$, which is indeed within the range $\left(0, \min\left\{\frac{4\varepsilon_r}{\alpha}, \frac{1-\alpha}{\alpha}, \frac{1}{2}\right\}\right)$ for $\varepsilon_r$ small enough. Substituting this precise choice of $\delta$ gives

$$\cent(\tau_{\mathcal{A}_{PT}}) \leq (1 + \varepsilon_r) \cdot \cent(\tau^*) + \frac{30\varepsilon_r^2}{\left(\frac{\varepsilon_r}{2\alpha}\right)\alpha} \cdot \text{MinCap}\left(\alpha\left(1 + 2\left(\frac{\varepsilon_r}{2\alpha}\right)\right)\right) + \frac{15\varepsilon_r\varepsilon_a}{\left(\frac{\varepsilon_r}{2\alpha}\right)\alpha} + 5\varepsilon_a$$

$$= (1 + \varepsilon_r) \cdot \cent(\tau^*) + 60\varepsilon_r \cdot \text{MinCap}\left(\alpha + \varepsilon_r\right) + 35\varepsilon_a.$$

Substituting the assumption of the Corollary gives the desired result:

$$\cent(\tau_{\mathcal{A}_{PT}}) \leq (1 + \varepsilon_r) \cdot \cent(\tau^*) + 60\varepsilon_r \cdot \text{MinCap}\left(\alpha + \varepsilon_r\right) + 35\varepsilon_a$$

$$\leq (1 + \varepsilon_r) \cdot \cent(\tau^*) + \frac{60\varepsilon_r}{c} \cdot \cent(\tau^*) + 35\varepsilon_a$$

$$= \left(1 + 60(1 + c) \cdot \frac{\varepsilon_r}{c}\right)\cent(\tau^*) + 35\varepsilon_a.$$

$\qquad\square$

**Theorem 3.5.** *Let $\mu$ be a continuous distribution with compact support $A \subset \mathbb{R}^d$, $\nu$ a discrete distribution with support $B \subseteq \mathbb{R}^d$, and $\alpha \in (0, 1)$ a parameter. Suppose $\varepsilon_r, \varepsilon_a \geq 0$, $\varepsilon_r < \frac{1}{5}\alpha$, and $\delta > \frac{5\varepsilon_r}{\alpha}$. Then algorithm $\mathcal{A}_{PT}$ is a $(\varepsilon_r, 5\varepsilon_a, \delta)$-approximation algorithm for $\alpha$-partial transport given any arbitrary $(\varepsilon_r, \varepsilon_a)$-approximation algorithm $\mathcal{A}_{RT}$ for the $\lambda$-robust OT problem.*

*Proof.* By the condition of $\frac{\lambda_R}{\lambda_L} = O(\varepsilon^{-1}\Delta)$ initially and stopping criteria $\lambda_R \leq (1 + \varepsilon_r)\lambda_L + \varepsilon_a$, we know that $\mathcal{A}_{PT}$ terminates within $O(\log \Delta)$ iterations. Additionally by construction of the plan $\tau_{\mathcal{A}_{PT}}$, $\text{M}(\tau_{\mathcal{A}_{PT}}) := \int_A \sum_{b \in B} \tau_{\mathcal{A}_{PT}}(a, b)da = (1 - \delta)\alpha$ as desired. Therefore, it suffices to bound the cost of $\tau_{\mathcal{A}_{PT}}$.

Since $\tau_{\lambda_R}$ is an $(\varepsilon_r, \varepsilon_a)$-approximate $\lambda_R$-robust OT plan, we observe

$$\cent(\tau_{\lambda_R}) + (1 - \text{M}(\tau_{\lambda_R}))\lambda_R \leq (1 + \varepsilon_r)\left[\cent(\tau^*) + (1 - \alpha)\lambda_R\right] + \varepsilon_a. \tag{9}$$

Now we note that the precise choice of constant $c$ in algorithm $\mathcal{A}_{PT}$ is $c := \frac{\mathrm{M}(\tau_{\lambda_R}) - \alpha(1-\delta)}{\mathrm{M}(\tau_{\lambda_R}) - \mathrm{M}(\tau_{\lambda_L})}$. Using the inequality from Lemma 3.2 and multiplying by $c \leq 1$, we observe that

$$c\left(\cent(\tau_{\lambda_L}) - \cent(\tau_{\lambda_R})\right) - (\alpha(1-\delta) - \mathrm{M}(\tau_{\lambda_R}))\lambda_R \leq 4\varepsilon_r\lambda_R + 4\varepsilon_a. \tag{10}$$

By definition, we note $\tau_{\mathcal{A}_{PT}} = \tau_{\lambda_R} + c(\tau_{\lambda_L} - \tau_{\lambda_R})$. Then adding Inequalities (9) and (10) gives

$$\begin{aligned}
\cent(\tau_{\mathcal{A}_{PT}}) + (1 - \alpha(1-\delta))\lambda_R &= \cent(\tau_{\lambda_R}) + c\left(\cent(\tau_{\lambda_L}) - \cent(\tau_{\lambda_R})\right) + (1 - \alpha(1-\delta))\lambda_R \\
&\leq (1 + \varepsilon_r)\cent(\tau^*) + \left[(1+\varepsilon_r)(1-\alpha) + 4\varepsilon_r\right]\lambda_R + 5\varepsilon_a \\
&= (1 + \varepsilon_r)\cent(\tau^*) + \left[1 - \alpha - \varepsilon_r\alpha + 5\varepsilon_r\right]\lambda_R + 5\varepsilon_a.
\end{aligned}$$

Subtract $(1 - \alpha(1-\delta))\lambda_R$ from both sides of the inequality to get

$$\cent(\tau_{\mathcal{A}_{PT}}) \leq (1 + \varepsilon_r) \cdot \cent(\tau^*) + \left(5\varepsilon_r - (\varepsilon_r + \delta)\alpha\right)\lambda_R + 5\varepsilon_a \leq (1 + \varepsilon_r) \cdot \cent(\tau^*) + 5\varepsilon_a,$$

where the second inequality above is true if $\delta \geq \frac{5\varepsilon_r}{\alpha}$. This concludes the Theorem statement. $\qquad\square$

# E   Missing Details and Proofs of Section 4

In this section, we present the missing details and proofs of the lemmas in Section 4. We begin by presenting missing proofs of our combinatorial framework in Section E.1. We then provide the details of the implementation of the procedures in Section E.2–E.6. Finally, we provide the proofs of correctness and efficiency of our algorithm in Section E.7.

## E.1   Missing proofs of the combinatorial framework

**Lemma 4.1.** *For any parameter $\delta > 0$, suppose $\tau, y(\cdot)$ denotes a $\delta$-feasible transport plan, and let $\lambda := \mathrm{Cap}(y)$. Suppose the following two conditions hold: (F1) for every deficit point $a \in A$, $\bar{y}(a) = 0$, and (F2) for all surplus points $b \in B$, $y(b) = \lambda$. Then, $\tau$ is a $2\delta$-close $\lambda$-robust transport plan.*

*Proof.* For any point $a \in A$, suppose $b_a \in B$ denotes the weighted nearest neighbor of $a$ with respect to weights $y(\cdot)$, i.e., $b_a := \arg\min_{b \in B} \mathrm{d}_y(a, b)$. For any transport plan $\tau'$, let $\mu^{\tau'}$ (resp. $\nu^{\tau'}$) denote the sub-measure of $\mu$ (resp. $\nu$) that is transported by $\tau'$, i.e., $\mu^{\tau'}$ (resp. $\nu^{\tau'}$) is the part of the mass of $\mu$ (resp. $\nu$) that is transported by $\tau'$. Let

$$\mu_f^{\tau'} := \mu - \mu^{\tau'} \qquad \text{and} \qquad \nu_f^{\tau'} := \nu - \nu^{\tau'}$$

denote the sub-measures of $\mu$ and $\nu$ that are not transported by $\tau'$, i.e., $\mu_f^{\tau'}$ and $\nu_f^{\tau'}$ are the untransported parts of the mass of $\mu$ and $\nu$, respectively. We can rewrite the $\lambda$-robust cost of the transport plan $\tau$ as

$$\begin{aligned}
w_\lambda(\tau) = \cent(\tau) + \lambda(1 - \mathrm{M}(\tau)) &= \sum_{b \in B} \int_A \mathrm{d}(a,b)\tau(a,b)\,da + \lambda(1 - \mathrm{M}(\tau)) \\
&= \sum_{b \in B} \int_A (\mathrm{d}(a,b) - y(b) + \bar{y}(a))\tau(a,b)\,da + \lambda(1 - \mathrm{M}(\tau)) \\
&\qquad + \sum_{b \in B} y(b)\nu^\tau(b) - \int_A \bar{y}(a)\mu^\tau(a)\,da.
\end{aligned} \tag{11}$$

Note that from property (F2), any point $b \in B$ with $\nu_f^\tau(b) > 0$ (i.e., any surplus point $b \in B$) has a weight $y(b) = \lambda$; therefore,

$$\sum_{b \in B} y(b)\nu^\tau(b) = \sum_{b \in B} y(b)\nu(b) - \sum_{b \in B} y(b)\nu_f^\tau(b) = \sum_{b \in B} y(b)\nu(b) - \lambda(1 - \mathrm{M}(\tau)). \tag{12}$$

Similarly, from property (F1), any point $a \in A$ with $\mu_f^\tau(a) > 0$ (i.e., any deficit point $a \in A$) has a zero weight. Hence,

$$\int_A \bar{y}(a)\mu^\tau(a)\,da = \int_A \bar{y}(a)\mu(a)\,da - \int_A \bar{y}(a)\mu_f^\tau(a)\,da = \int_A \bar{y}(a)\mu(a)\,da. \tag{13}$$

Finally, due to the $\delta$-feasibility of $\tau, y(\cdot)$, the point $a$ lies inside the $2\delta$-expanded restricted Voronoi cell of $b$, and therefore, by increasing the weights of $b$ by $2\delta$, the point $b$ becomes the weighted nearest neighbor of $a$ and also contains $a$ in its dual disc, i.e., $\bar{y}(a) \leq (y(b) + 2\delta) - \mathrm{d}(a,b)$. By rearranging this inequality,

$$\mathrm{d}(a,b) - y(b) + \bar{y}(a) \leq 2\delta. \tag{14}$$

Plugging Equations (12), (13), and (14) into Equation (11),

$$
\begin{aligned}
w_\lambda(\tau) &= \sum_{b\in B} \int_A (\mathrm{d}_y(a,b) + \bar{y}(a))\tau(a,b)\,da + \sum_{b\in B} y(b)\nu^\tau(b) - \int_A \bar{y}(a)\mu^\tau(a)\,da + \lambda(1 - \mathrm{M}(\tau)) \\
&= \sum_{b\in B} \int_A (\mathrm{d}_y(a,b) + \bar{y}(a))\tau(a,b)\,da + \sum_{b\in B} y(b)\nu(b) - \int_A \bar{y}(a)\mu(a)\,da \\
&\leq \sum_{b\in B} \int_A 2\delta\tau(a,b)\,da + \sum_{b\in B} y(b)\nu(b) - \int_A \bar{y}(a)\mu(a)\,da \\
&\leq 2\delta + \sum_{b\in B} y(b)\nu(b) - \int_A \bar{y}(a)\mu(a)\,da, \tag{15}
\end{aligned}
$$

where the last inequality holds since $\tau$ transports at most 1 unit of mass, i.e., $\mathrm{M}(\tau) \leq 1$. Next, let $\tau^*$ denote any $\lambda$-robust OT plan. We can rewrite the $\lambda$-robust cost of $\tau^*$ as

$$
\begin{aligned}
w_\lambda(\tau^*) &= \cancel{c}(\tau^*) + \lambda(1 - \mathrm{M}(\tau^*)) = \sum_{b\in B} \int_A \mathrm{d}(a,b)\tau^*(a,b)\,da + \lambda(1 - \mathrm{M}(\tau^*)) \\
&= \sum_{b\in B} \int_A (\mathrm{d}(a,b) - y(b) + \bar{y}(a))\tau^*(a,b)\,da + \lambda(1 - \mathrm{M}(\tau^*)) \\
&\quad + \sum_{b\in B} y(b)\nu^{\tau^*}(b) - \int_A \bar{y}(a)\mu^{\tau^*}(a)\,da. \tag{16}
\end{aligned}
$$

Since $\lambda = \mathrm{Cap}(y)$,

$$\sum_{b\in B} y(b)\nu^{\tau^*}(b) = \sum_{b\in B} y(b)\nu(b) - \sum_{b\in B} y(b)\nu_f^{\tau^*}(b) \geq \sum_{b\in B} y(b)\nu(b) - \lambda(1 - \mathrm{M}(\tau)). \tag{17}$$

Similarly, since all weights $\bar{y}(\cdot)$ are positive,

$$\int_A \bar{y}(a)\mu^{\tau^*}(a)\,da = \int_A \bar{y}(a)\mu(a)\,da - \int_A \bar{y}(a)\mu_f^{\tau^*}(a)\,da \leq \int_A \bar{y}(a)\mu(a)\,da. \tag{18}$$

Finally, by the definition of the derived weights, for any pair of points $(a,b) \in A \times B$,

$$\mathrm{d}(a,b) - y(b) + \bar{y}(a) \geq 0. \tag{19}$$

Plugging Equations (17), (18), and (19) into Equation (16),

$$
\begin{aligned}
w_\lambda(\tau^*) &= \sum_{b\in B} \int_A (\mathrm{d}_y(a,b) + \bar{y}(a))\tau^*(a,b)\,da + \lambda(1 - \mathrm{M}(\tau^*)) \\
&\quad + \sum_{b\in B} y(b)\nu^{\tau^*}(b) - \int_A \bar{y}(a)\mu^{\tau^*}(a)\,da \\
&\geq \sum_{b\in B} \int_A (\mathrm{d}_y(a,b) + \bar{y}(a))\tau^*(a,b)\,da + \sum_{b\in B} y(b)\nu(b) - \int_A \bar{y}(a)\mu(a)\,da \\
&\geq \sum_{b\in B} y(b)\nu(b) - \int_A \bar{y}(a)\mu(a)\,da, \tag{20}
\end{aligned}
$$

Combining Equations (15) and (20),

$$w_\lambda(\tau) \leq 2\delta + \sum_{b\in B} y(b)\nu(b) - \int_A \bar{y}(a)\mu(a)\,da \leq w_\lambda(\tau^*) + 2\delta,$$

and $\tau$ is a $2\delta$-close $\lambda$-ROT plan. $\qquad\square$

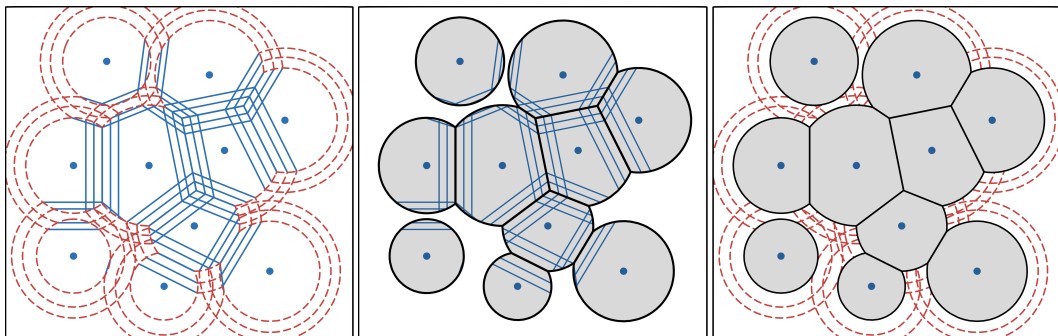

Figure 7: (left) the Voronoi-based boundaries (blue solid lines) and disk-based boundaries (red dashed lines), (middle) the subset of boundaries of the expansions that lie inside the restricted Voronoi cells, and (right) the subset of boundaries of the expansions that lies outside of all restricted Voronoi cells.

The following lemma shows that for any $\delta$-feasible transport plan $\tau, y(\cdot)$, one can interchangeably use the implicit representation $\hat{\tau}$ without violating the $\delta$-feasibility conditions.

**Lemma E.1.** *For any $\delta$-feasible semi-discrete transport plan $\tau, y(\cdot)$, the implicit representation $\hat{\tau}, y(\cdot)$ is also $\delta$-feasible. Similarly, for any set of weights $y(\cdot)$, any $\delta$-feasible transport plan $\hat{\tau}$ over $A_\delta \times B$ can be converted into a $\delta$-feasible semi-discrete transport plan $\tau, y(\cdot)$ over $A \times B$.*

*Proof.* Recall that for any semi-discrete transport plan $\tau$ between $\mu$ and $\nu$, the implicit representation of $\tau$ is defined as a transport plan where $\hat{\tau}(r_\varphi, b) := \int_\varphi \tau(a, b)\, da$ for each pair $(\varphi, b) \in \mathcal{X}_\delta \times B$. For each region $\varphi \in \mathcal{X}_\delta$, the representative point $r_\varphi$ is inside $\varphi$. Furthermore, since $RV_y^{2\delta}(b)$ is included in the set of cells used in the construction of $\mathcal{X}_\delta$, the region $\varphi$ is either completely inside $RV_y^{2\delta}(b)$ or is completely outside of it. Hence, due to the $\delta$-feasibility of $\tau, y(\cdot)$, for any pair $(a, b) \in A \times B$ such that $\tau(a, b) > 0$ (since $a \in RV_y^{2\delta}(b)$), the region $\varphi \in \mathcal{X}_\delta$ containing $a$ also completely lies inside $RV_y^{2\delta}(b)$ and $r_\varphi \in RV_y^{2\delta}(b)$. Consequently, for any pair $(r, b) \in A_\delta \times B$ with $\hat{\tau}(r, b) > 0$, the point $r$ lies inside the $2\delta$-expanded restricted Voronoi cell of $b$, and $\hat{\tau}, y(\cdot)$ is $\delta$-feasible.

Next, suppose $\hat{\tau}$ is a $\delta$-feasible transport plan over $A_\delta \times B$, i.e., for any pair $(r, b) \in A_\delta \times B$, if $\hat{\tau}(r, b) > 0$, then $r \in RV_y^{2\delta}(b)$. As discussed above, for any region $\varphi \in \mathcal{X}_\delta$, if $r_\varphi \in RV_y^{2\delta}(b)$, then $\varphi$ would be completely inside $RV_y^{2\delta}(b)$. Hence, if we define a transport plan $\tau$ such that $\tau(\varphi, b) = \hat{\tau}(r_\varphi, b)$ for each pair $(\varphi, b) \in \mathcal{X}_\delta \times B$, then for any pair $(a, b) \in A \times B$ such that $\tau(a, b) > 0$, then $a \in RV_y^{2\delta}(b)$, and $\tau, y(\cdot)$ is $\delta$-feasible. $\qquad\square$

**Size of residual graph.** Next, we show that the residual graph has $O(n^d)$ points and $O(n^{d+1})$ edges for $d$-dimensional distributions, which will be used in analyzing the running time of different steps of our algorithm.

Fix any set of weights $y(\cdot)$ for $B$. For any point $b \in B$, for simplicity in presentation, we omit the weights $y(\cdot)$ from notation and let $RV^0(b) := RV_y^\delta(b), RV^1(b) := RV_y^\delta(b)$, and $RV^2(b) := RV_y^{2\delta}(b)$ to denote the restricted Voronoi cell, $\delta$-expanded, and $2\delta$-expanded Voronoi cells of $b$, respectively. For any cell $RV^i(b)$ for $b \in B$ and $i \in \{0, 1, 2\}$, let $RV_V^i(b)$ (resp. $RV_D^i(b)$) denote the part of the boundary of $RV^i(b)$ that is formed by the Voronoi cell $V_y^{i\delta}(b)$ (resp. disk with radius $y(b) + i\delta$ centered at $b$). We refer to this subset of boundaries as *Voronoi-based* (resp. *disk-based*) boundaries. See Figure 7(left).

The following lemma provides an important property of these boundaries, which helps us in bounding the size of the residual graph.

**Lemma E.2.** *Suppose $\mathcal{R} := \bigcup_{b \in B} RV_y(b)$ denotes the union (of the interior) of all restricted Voronoi cells. For any point $b \in B$ and any $i \in \{1, 2\}$, all interior points of the Voronoi-based boundaries $RV_V^i(b)$ lies inside $\mathcal{R}$ and all interior points of the disk-based boundaries $RV_D^i(b)$ lies outside $\mathcal{R}$.*

*Proof.* Consider any interior point $p \in RV_V^i(b)$. Since this point is determined by the $i\delta$-expanded Voronoi cell of $b$, then $\mathrm{d}(p, b) < y(b) + i\delta$. Suppose $p$ lies on a weighted bisector between $b$ and another point $b' \in B$. In this case,

$$\mathrm{d}(p, b') - y(b') = \mathrm{d}(p, b) - (y(b) + i\delta) > 0.$$

Since $b'$ is a weighted nearest neighbor of $p$ and has a positive weighted distance to $p$, the point $p$ lies inside the restricted Voronoi cell of $b'$. See Figure 7(middle).

Similarly, consider any interior point $p \in RV_D^i(b)$. By construction, $\mathrm{d}(p, b) = y(b) + i\delta$. Consider a weight function $y_b^i(\cdot)$ that assigns $y_b^i(b) := y(b) + i\delta$ and $y_b^i(b') := y(b')$ for all $b' \neq b$. By the definition of restricted Voronoi cells, the weighted nearest neighbor of $p$ with weights $y_b^i(\cdot)$ is $b$, and any other point $b' \in B$ would have a non-positive weighted distance to $p$. Since the weighted distance of each point $b' \neq b$ to $p$ is the same with respect to $y(\cdot)$ and $y_b^i(\cdot)$, the point $p$ lies outside of the restricted Voronoi cell of $b'$. See Figure 7(right). $\square$

From Lemma E.2, any intersection point in the arrangement is formed using either $d$ Voronoi-based boundaries or $d$ disk-based boundaries. As shown in [1], there are $O(n^d)$ intersection points between the Voronoi-based boundaries. Furthermore, any subset of $d$ disks intersect in at most 2 points. Therefore, the total number of such intersection points is $O(\binom{n}{d}) = O(n^d)$. Hence, the number of intersection points, and consequently the number of regions in the arrangement is $O(n^d)$. Since the residual graph is a bipartite graph between set $A_\delta$ (with size $O(n^d)$) and set $B$ (with size $n$), the number of its edges would be $O(n^{d+1})$, leading to the following lemma.

**Lemma E.3.** *For $d$-dimensional distributions, for any $d \geq 2$, the residual graph has $O(n^d)$ nodes and $O(n^{d+1})$ edges.*

## E.2  SEARCHANDCONSOLIDATE Procedure

Define $\mathcal{V}$ as the set of all violating deficit points of $A_\delta$. Mark all points of $B$ and all backward edges as unvisited. Let $U := B$ denote the set of unvisited points, and at any point during the execution of the algorithm, let $Q$ denote the search path. The SEARCHANDCONSOLIDATE procedure initiates a partial DFS from each point in $\mathcal{V}$ in the graph $\overleftarrow{\mathcal{G}}_\delta$ to find admissible consolidating and augmenting paths. For any point $r \in \mathcal{V}$, set $Q = \langle r = r_1 \rangle$ and execute the following steps until $Q$ becomes empty.

1. If $Q = \langle r = r_1, b_1, \ldots, r_i, b_i \rangle$,
   (a) If $b_i$ is a surplus point, then the reverse path $P = \langle b_i, r_i, b_{i-1}, \ldots, r_1 \rangle$ is an admissible augmenting path in $\mathcal{G}_\delta$. Augment $\hat{\tau}_\delta$ along $P$. If $r$ remains a (violating) deficit point, then set $Q = \langle r = r_1 \rangle$ and continue. Otherwise, stop the search from $r$.
   (b) If $b_i$ is not a surplus point, for any unvisited backward edge $(b_i, r')$ in $\overleftarrow{\mathcal{G}}_\delta$, add $r'$ as $r_{i+1}$ to $Q$. If there are no unvisited backward edges from $b_i$ in $\overleftarrow{\mathcal{G}}_\delta$, then mark $b_i$ as visited and remove $b_i$ from $U$ and $Q$. If $Q$ becomes empty, stop the search from $b$.

2. If $Q = \langle r = r_1, b_1, \ldots, b_i, r_{i+1} \rangle$, let $b^* := \arg\min_{b' \in U} d_y(r_{i+1}, b')$ denote the unvisited point with the minimum weighted distance to $r_{i+1}$.
   (a) If $\bar{y}(r_{i+1}) = 0$, then the reverse path $P = \langle r_{i+1}, b_i, r_i, \ldots, r_1 \rangle$ is an admissible consolidating path in $\mathcal{G}_\delta$. Update $\hat{\tau}_\delta$ along $P$. If $r$ remains a (violating) deficit point, set $Q = \langle r = r_1 \rangle$ and continue. Otherwise, stop the search from $b$.
   (b) Otherwise, if $(b^*, r_{i+1}, b_i)$ is admissible, i.e., $d_y(r_{i+1}, b^*) < d_y(r_{i+1}, b_i)$, then add $b^*$ as $b_{i+1}$ to $Q$.
   (c) Otherwise, remove $r_{i+1}$ from $Q$ and mark the backward edge $(b_i, r_{i+1})$ as visited.

The procedure terminates when all partial search procedures from all points in $\mathcal{V}$ are terminated.

**Lemma E.4.** *Given a $\delta$-feasible transport plan $\tau_\delta, y(\cdot)$, after the execution of the* SEARCHANDCONSOLIDATE *procedure,*

(SC1)  *the transport plan remains $\delta$-feasible,*

(SC2)  *each balanced point $b \in B$ remains balanced, and*

*(SC3)* *there are no admissible consolidating paths and no admissible augmenting paths to violating deficit regions in the residual graph.*

*Proof.* Suppose $\tau_\delta$ (resp. $\tau'_\delta$) denotes the transport plan maintained by the algorithm after (resp. before) the execution of the SEARCHANDCONSOLIDATE procedure.

For any edge $(r, b) \in A_\delta \times B$ with $\hat{\tau}_\delta(r, b) > 0$, if $\hat{\tau}'_\delta(r, b) > 0$, then by the $\delta$-feasibility of $\hat{\tau}'_\delta, y(\cdot)$, the point $r$ lies inside $RV_y^{2\delta}(b)$. Otherwise, the edge $(r, b)$ was a forward edge on an augmenting or consolidating path $P$ so that after updating the transport plan along $P$, it has become a backward edge. In this case, there is a forward edge from $b$ to $r$ in the residual graph, which means that $r \in RV_y^{2\delta}(b)$ and (SC1) holds.

Next, note that any augmenting path $P$ is from a surplus point $b \in B$ to a point $r_\varphi \in A_\delta$. Furthermore, any consolidating path $P$ is from a zero-weight point $r' \in A_\delta$ to a violating deficit point $r_\varphi \in A_\delta$. For any point $b \in B$ other than the two endpoints of a path $P$, the amount of mass transported from $b$ remains unchanged after updating the transport plan along $P$. Therefore, updating a transport plan along any augmenting or consolidating path does not change the amount of mass transported from any balanced point, leading to (SC2).

Finally, to prove property (SC3), we begin by showing the following properties:

(SC4) there are no admissible cycles in the residual graph, and

(SC5) any point $b \in B$ and any backward edge that are marked as visited does not form any admissible augmenting paths to violating deficit points or consolidating paths in the same execution of the SEARCHANDCONSOLIDATE procedure.

Suppose $P^1, \ldots, P^k$ denotes the sequence of augmenting and consolidating paths computed by the SEARCHANDCONSOLIDATE procedure, and for any $i \in [0, k]$, let $\hat{\tau}_\delta^i$ and $\mathcal{G}_\delta^i$ denote the transport plan and residual graph maintained after updating the transport plan along $P^i$, respectively. Define $V^i$ (resp. $U^i$) to be the set of visited (resp. unvisited) points of $B$ with respect to $\hat{\tau}_\delta^i$.

We prove (SC4) using an induction on $i$. Since $\hat{\tau}_\delta^0$ is obtained from the transport plan computed by the ACYCLIFY procedure, there are no cycles of admissible triples in the residual graph, and (SC4) holds for $\mathcal{G}_\delta^0$. For any $i \in [1, k]$, assuming (SC4) on $\mathcal{G}_\delta^0, \ldots, \mathcal{G}_\delta^{i-1}$, we show that (SC4) also holds for $\mathcal{G}^i$. For any admissible triple $(b, r, b')$ in $\mathcal{G}^i$ formed after updating $\hat{\tau}_\delta^{i-1}$ along $P^i$, we show below that the triple $(b, r, b')$ does not form admissible cycles and conclude (SC4) on $\mathcal{G}_\delta^i$.

For any visited point $b \in V^{i-1} \setminus V^{i-2}$, the search from $b$ did not lead to the computation of an augmenting or consolidating path; therefore, any point $b'$ that is reachable from $b$ by an admissible path in the reverse of residual graph $\overleftarrow{\mathcal{G}^{i-1}}$ (and therefore, is reachable by our partial DFS procedure from $b$) would have also been marked as visited. In other words, any point $b' \in B$ that has an admissible path to the visited point $b$ in the residual graph $\mathcal{G}^{i-1}$ is also visited. By this observation, there are no admissible paths from any unvisited point in $U^{i-1}$ to any visited point in $V^{i-1}$.

Below, we show that for any newly formed admissible triple $(b, r, b')$, $b \in V^{i-1}$ and $b' \in U^{i-1}$. Assuming this, there are no admissible triples from any visited point to any unvisited point (in the reversed residual graph), and therefore, the newly formed admissible triples (which are from an unvisited point to a visited point in the reversed residual graph) do not form any admissible cycles and (SC4) holds for $\mathcal{G}_\delta^i$ as well. Consider any triple $(b, r, b')$ that is admissible in $\mathcal{G}_\delta^i$ but not in $\mathcal{G}_\delta^{i-1}$. Recall that the SEARCHANDCONSOLIDATE procedure does not change the weights of points, and therefore, since $(b, r, b')$ is not admissible in $\mathcal{G}_\delta^{i-1}$, $(r, b')$ is not a backward edge in $\mathcal{G}_\delta^{i-1}$, i.e., $(r, b') \in P^i$ as a forward edge. From step 2(b) of the SEARCHANDCONSOLIDATE procedure, $b'$ is the weighted nearest unvisited neighbor of $r$; therefore, since $d_y(r, b) > d_y(r, b')$ (from the admissibility of $(b, r, b')$) and the procedure added $b'$ to the search path (instead of $b$), the point $b$ was marked as visited by the procedure. Therefore, for any newly formed admissible triple $(b, r, b')$, $b \in V^{i-1}$ and $b' \in U^{i-1}$.

Next, we show that (SC5) holds. For any visited point $b \in V^i$, as discussed above, all vertices that are reachable from $b$ by an admissible path in the reversed residual graph are also visited. Since all surplus points $b_s \in B$ (resp. points $b_z \in B$ transporting mass to a zero-weight point $r \in A_\delta$) are

unvisited, $b_s$ (resp. $b_z$) is not reachable from any visited point $b \in B$ by our search algorithm and therefore, the visited points do not participate in an admissible augmenting or consolidating path. Furthermore, the procedure marks a backward edge $(r, b)$ as visited if, for each admissible triple $(b', r, b)$, the point $b'$ is visited. Since the point $b'$ cannot be included in an admissible augmenting path to a violating deficit point or a consolidating path, the visited backward edge $(r, b)$ also does not form an admissible augmenting path to a violating deficit point or a consolidating path.

For any violating deficit point $r_\varphi \in A_\delta$ with respect to $\hat{\tau}_\delta, y(\cdot)$, the search from $r_\varphi$ has been terminated since $Q$ was empty; hence, all forward edges incident on $r_\varphi$ are to visited points of $B$, which from (SC5) do not form admissible augmenting/consolidating paths; therefore, $r_\varphi$ will not form any admissible augmenting paths to violating deficit points or consolidating paths during the same execution of the SEARCHANDCONSOLIDATE procedure, leading to (SC3). □

## E.3 REDUCEWEIGHTS Procedure

The procedure reduces the weights of all points in $B$ that have admissible alternating paths to some deficit points $r \in A$ with $\bar{y}(r) > 0$, leading to the formation of new admissible augmenting and consolidating paths to the violating free points in the residual graph. The procedure performs partial DFS from all violating deficit points in the reverse of the residual graph to compute a set $\mathcal{K}$ of all points of $B$ that have admissible alternating paths to the violating deficit points. The procedure then decreases the weights of all points in $\mathcal{K}$ by $\delta$ and shrinks their restricted Voronoi cells. It finally recomputes $A_\delta, \mathcal{G}_\delta$, and $\hat{\tau}_\delta$. We describe the steps of the DFS below.

Define $\mathcal{V} \subset A_\delta$ as the set of violating deficit points. Given the reversed residual graph $\overleftarrow{\mathcal{G}}_\delta$, mark all points $b \in B$ and all backward edges $(b, r)$ as unvisited and set $\mathcal{K} = \emptyset$ and let $U = B$ denote the set of unvisited points of $B$. For any point $r \in \mathcal{V}$, start a partial DFS in the reverse residual graph $\overleftarrow{\mathcal{G}}_\delta$ by setting $Q := \langle r = r_1 \rangle$ as the search path that the algorithm grows. Execute the following steps.

1. If $Q = \langle r = r_1, \ldots, b_i \rangle$,
   (a) If there exists an unvisited backward edge $(b_i, r^*)$ in $\overleftarrow{\mathcal{G}}_\delta$, add $r^*$ to $Q$ as $r_{i+1}$.
   (b) Otherwise, mark $b_i$ as visited and remove $b_i$ from $U$ and $Q$. Add $b_i$ to $\mathcal{K}$.

2. If $Q = \langle r = r_1, \ldots, b_i, r_i \rangle$, let $b^* := \arg\min_{b' \in U} d_y(r_i, b')$.
   (a) If $d_y(r_i, b^*) < d_y(r_i, b_i)$, i.e., $(b^*, r_i, b_i)$ is admissible, add $b^*$ as $b_{i+1}$ to $Q$.
   (b) Otherwise, remove $r_i$ from $Q$ and mark $(b_i, r_i)$ as visited.

After all DFS procedures from all points in $\mathcal{V}$ terminate, for each point $b \in \mathcal{K}$, set $y(b) \leftarrow y(b) - \delta$.

**Lemma E.5.** *Given a $\delta$-feasible transport plan $\tau_\delta, y(\cdot)$ such that there are no admissible consolidating paths and no admissible augmenting path to a violating deficit region, after the execution of the* REDUCEWEIGHTS *procedure,*

*(R1) the transport plan remains $\delta$-feasible,*

*(R2) the weight of each point $b \in B$ containing violating deficit regions inside $RV_y(b)$ decreases by $\delta$, and*

*(R3) the weight of each surplus point $b \in B$ remains unchanged.*

*Proof.* Let $y(\cdot), \mathcal{X}_\delta, A_\delta$ (resp. $y'(\cdot), \mathcal{X}'_\delta, A'_\delta$) denote the set of weights of $B$, decomposition of $A$, and the set of representative points before (resp. after) the execution of the REDUCEWEIGHTS procedure.

To prove property (R1), first note that for each point $b \in \mathcal{K}$, there are no zero-weight points $r \in RV_y^{2\delta}$ such that $\tau_\delta(r, b) > 0$, since otherwise, there would have been an admissible consolidating path in the residual graph, which is in contrast with Lemma E.4 (SC3). Below, we show that for any pair of points $(a, b) \in A \times B$ such that $\tau_\delta(a, b) > 0$, we have $a \in RV_{y'}^{2\delta}(b)$, and hence (R1) holds. Let $b_a$ (resp. $b'_a$) denote the weighted nearest neighbor of $a$ with respect to weights $y(\cdot)$ (resp. $y'(\cdot)$). From the $\delta$-feasibility of $\tau_\delta, y(\cdot), d_y(a, b) \leq d_y(a, b_a) + 2\delta$. Let $r_a$ (resp. $r'_a$) denote the representative point of the region containing $a$. As discussed above, $\bar{y}(r_a) > 0$ and $r_a$ lies inside the restricted Voronoi cell of $b_a$.

- If $b \in \mathcal{K}$,
    - if $b'_a \in \mathcal{K}$, then
    $$\mathrm{d}_{y'}(a, b) = \mathrm{d}_y(a, b) + \delta \le \mathrm{d}_y(a, b'_a) + 3\delta = \mathrm{d}_{y'}(a, b'_a) + 2\delta.$$
    - Otherwise, the triple $(b'_a, r_a, b)$ would not be admissible (since otherwise $b'_a \in \mathcal{K}$), and $\mathrm{d}_y(r_a, b) \le \mathrm{d}_y(r_a, b'_a)$. Hence, $\mathrm{d}_y(a, b) \le \mathrm{d}_y(a, b'_a) + \delta$ and
    $$\mathrm{d}_{y'}(a, b) = \mathrm{d}_y(a, b) + \delta \le \mathrm{d}_y(a, b'_a) + 2\delta = \mathrm{d}_{y'}(a, b'_a) + 2\delta.$$
- Otherwise, $b \notin \mathcal{K}$. In this case,
$$\mathrm{d}_{y'}(a, b) = \mathrm{d}_y(a, b) \le \mathrm{d}_y(a, b'_a) + 2\delta \le \mathrm{d}_{y'}(a, b'_a) + 2\delta.$$

Therefore, for any pair $(a, b)$ with $\tau_\delta(a, b) > 0$, we have $a \in RV_y^{2\delta}(b)$, and $\tau_\delta, y'(\cdot)$ would be $\delta$-feasible, i.e., (R1) holds.

Since the REDUCEWEIGHTS procedure initiates a partial DFS from each violating deficit point $r \in \mathcal{V}$, all points of $B$ with violating deficit regions in their restricted Voronoi cell will be added to the search path and to the set $\mathcal{K}$ in step 1(b) of the procedure; hence, their weights will be reduced by $\delta$ (leading to (R2)). Furthermore, for any surplus point $b \in B$, since there are no admissible augmenting paths from the violating deficit regions (from Lemma E.4), the surplus point $b$ will not be added to the search path and hence, cannot be added to the set $\mathcal{K}$, leading to (R3). □

### E.4 SEARCHANDAUGMENT Procedure

The SEARCHANDAUGMENT procedure initiates a partial DFS from deficit points of $A_\delta$ in the reverse of the residual graph to find admissible augmenting paths and admissible alternating paths to inactive points. The procedure then updates the transport plan along those paths. The SEARCHANDAUGMENT procedure in this paper is similar to the SEARCHANDAUGMENT procedure introduced in [1], and the only difference is that when the search path reaches an inactive point of $B$, i.e., a point $b \in B$ with $y(b) = \lambda$, the procedure updates the transport plan along the alternating path and restarts the search.

Let $\mathcal{D} \subset A_\delta$ denote the subset of deficit points of $A_\delta$ with respect to $\tau_\delta$. Mark all points of $B$ and all backward edges as unvisited. Define $U := B$ as the set of unvisited points of $B$. For any point $r \in \mathcal{D}$, initiate a partial DFS $Q = \langle r = r_1 \rangle$ in the reverse residual graph $\overleftarrow{\mathcal{G}}_\delta$ as the search path that the procedure grows. Execute the following steps.

1. If $Q = \langle r = r_1, b_1, \ldots, r_i, b_i \rangle$,
    (a) If $b_i$ is surplus, then the reverse path $P = \langle b_i, r_i, \ldots, r_1 \rangle$ is an admissible augmenting path in $\mathcal{G}_\delta$. Augment $\hat{\tau}_\delta$ along $P$. If $r$ is still a deficit point, set $Q = \langle r = r_1 \rangle$. Otherwise, stop the current search.
    (b) Otherwise, if $b_i$ is an inactive point (i.e., $y(b_i) = \lambda$), then the reverse path $P = \langle b_i, r_i, \ldots, r_1 \rangle$ is an admissible alternating path to an inactive point in $\mathcal{G}_\delta$. Update $\hat{\tau}_\delta$ along $P$. If $r$ is still a deficit point, set $Q = \langle r = r_1 \rangle$. Otherwise, stop the current search.
    (c) Otherwise, if there is an unvisited backward edge $(b_i, r')$ in $\overleftarrow{\mathcal{G}}_\delta$, then add $r'$ as $r_{i+1}$ to $Q$. Otherwise, mark $b_i$ as visited and remove $b_i$ from $U$ and $Q$.
2. If $Q = \langle r = r_1, b_1, \ldots, b_i, r_{i+1} \rangle$, let $b := \arg\min_{b' \in U} \mathrm{d}_y(r_{i+1}, b')$.
    (a) If $\mathrm{d}_y(r_{i+1}, b) < \mathrm{d}_y(r_{i+1}, b_i)$, i.e., $(b, r_{i+1}, b_i)$ is admissible, add $b$ as $b_{i+1}$ to $Q$.
    (b) Otherwise, remove $r_{i+1}$ from $Q$ and mark the backward edge $(b_i, r_{i+1})$ as visited.

The procedure terminates when all partial DFS executions from all points in $\mathcal{D}$ terminate.

**Lemma E.6.** *Given a $\delta$-feasible transport plan $\tau_\delta, y(\cdot)$ satisfying condition (F1) in Lemma 4.1, after the execution of the SEARCHANDAUGMENT procedure,*

*(SA1) the transport plan $\hat{\tau}_\delta, y(\cdot)$ remains $\delta$-feasible and (F1) remains satisfies, and*

*(SA2) there are no admissible cycles, admissible augmenting paths, and admissible alternating paths from inactive points to deficit regions in the residual graph.*

*Proof.* Suppose $\tau_\delta$ (resp. $\tau'_\delta$) denotes the transport plan maintained by the algorithm after (resp. before) the execution of the SEARCHANDAUGMENT procedure.

For any edge $(r, b) \in A_\delta \times B$ with $\hat{\tau}_\delta(r, b) > 0$, if $\hat{\tau}'_\delta(r, b) > 0$, then by the $\delta$-feasibility of $\hat{\tau}'_\delta, y(\cdot)$, the point $r$ lies inside $RV_y^{2\delta}(b)$. Otherwise, the edge $(r, b)$ was a forward edge on an augmenting path $P$ (resp. alternating path $P'$ from an inactive point) so that after updating the transport plan along $P$ (resp. $P'$), it has become a backward edge. In this case, there is a forward edge from $b$ to $r$ in the residual graph, which means that $r \in RV_y^{2\delta}(b)$, i.e., $\hat{\tau}_\delta, y(\cdot)$ is also $\delta$-feasible. Furthermore, since the SEARCHANDAUGMENT procedure does not change the weights, (F1) continues to be true, and (SA1) holds.

Suppose $P^1, \ldots, P^k$ denotes the sequence of admissible augmenting paths and alternating paths to inactive points computed by the SEARCHANDAUGMENT procedure, and for any $i \in [0, k]$, let $\hat{\tau}_\delta^i$ and $\mathcal{G}_\delta^i$ denote the transport plan and residual graph maintained after updating the transport plan along $P^i$, respectively. Define $V^i$ (resp. $U^i$) to be the set of visited (resp. unvisited) points of $B$ with respect to $\hat{\tau}_\delta^i$.

We use an induction on $i$ to show that there are no admissible cycles in the residual graph. Since $\hat{\tau}_\delta^0$ is obtained from the transport plan computed by the ACYCLIFY procedure, there are no admissible cycles in the residual graph $\mathcal{G}_\delta^0$. For any $i \in [1, k]$, assuming there are no admissible cycles in $\mathcal{G}_\delta^0, \ldots, \mathcal{G}_\delta^{i-1}$, we show that the same holds for $\mathcal{G}^i$. For any admissible triple $(b, r, b')$ in $\mathcal{G}^i$ formed after updating $\hat{\tau}_\delta^{i-1}$ along $P^i$, we show below that the triple $(b, r, b')$ does not form admissible cycles and conclude that there are no admissible cycles in $\mathcal{G}_\delta^i$.

For any visited point $b \in V^{i-1} \setminus V^{i-2}$, the search from $b$ did not lead to the computation of an augmenting path or an alternating path to an inactive point; therefore, any point $b'$ that is reachable from $b$ by an admissible path in the reverse of residual graph $\overleftarrow{\mathcal{G}^{i-1}}$ (and therefore, is reachable by our partial DFS procedure from $b$) would have also been marked as visited. In other words, any point $b' \in B$ that has an admissible path to the visited point $b$ in the residual graph $\mathcal{G}^{i-1}$ is also visited. By this observation, there are no admissible paths from any unvisited point in $U^{i-1}$ to any visited point in $V^{i-1}$.

Below, we show that for any newly formed admissible triple $(b, r, b')$, $b \in V^{i-1}$ and $b' \in U^{i-1}$. Assuming this, there are no admissible triples from any visited point to any unvisited point (in the reversed residual graph), and therefore, the newly formed admissible triples (which are from an unvisited point to a visited point in the reversed residual graph) do not form any admissible cycles, as claimed. Consider any triple $(b, r, b')$ that is admissible in $\mathcal{G}_\delta^i$ but not in $\mathcal{G}_\delta^{i-1}$. Recall that the SEARCHANDAUGMENT procedure does not change the weights of points, and therefore, since $(b, r, b')$ is not admissible in $\mathcal{G}_\delta^{i-1}$, $(r, b')$ is not a backward edge in $\mathcal{G}_\delta^{i-1}$, i.e., $(r, b') \in P^i$ as a forward edge. From step 2 of the SEARCHANDAUGMENT procedure, $b'$ is the weighted nearest unvisited neighbor of $r$; therefore, since $\mathrm{d}_y(r, b) > \mathrm{d}_y(r, b')$ (from the admissibility of $(b, r, b')$) and the procedure added $b'$ to the search path (instead of $b$), the point $b$ was marked as visited by the procedure. Therefore, for any newly formed admissible triple $(b, r, b')$, $b \in V^{i-1}$ and $b' \in U^{i-1}$. To summarize, any newly formed admissible triple is from a visited point to an unvisited point (in the residual graph), while there are no admissible triples from an unvisited point to a visited points; hence, the newly formed admissible triples do not form admissible cycles, and using the inductive hypothesis, the residual graph remains free of any admissible cycles.

To prove property (SA3), we next the following properties:

(SA4) any point $b \in B$ and any backward edge that is marked as visited does not form any admissible augmenting paths or alternating paths to inactive points in the same execution of the SEARCHANDAUGMENT procedure.

For any visited point $b \in V^i$, as discussed above, all vertices that are reachable from $b$ by an admissible path in the reversed residual graph are also visited. Since all inactive points $b_s \in B$ (resp. points $b_z \in B$ with zero-weight points $r \in A_\delta$ in $RV_y^{2\delta}(b)$) are unvisited, $b_s$ (resp. $b_z$) is not reachable from any visited point $b \in B$ by our search algorithm and therefore, the visited points do not participate in an admissible augmenting path or alternating path to inactive points. Furthermore, the procedure marks a backward edge $(r, b)$ as visited if, for each admissible triple $(b', r, b)$, the point

$b'$ is visited. Since the point $b'$ cannot be included in an admissible augmenting path or alternating path to inactive points, the visited backward edge $(r, b)$ also does not form an admissible augmenting path or alternating path to inactive points.

For any deficit point $r \in \mathcal{D}$ with respect to $\hat{\tau}_\delta, y(\cdot)$, the search from $r$ has been terminated since $Q$ was empty (otherwise, $r$ would have been balanced); hence, all forward edges incident on $r$ are to visited points of $B$, which from (SA4) do not form admissible augmenting paths or alternating paths to inactive points; therefore, $r$ will not form any admissible augmenting paths or alternating paths to inactive points during the same execution of the SEARCHANDCONSOLIDATE procedure, leading to (SA2). □

## E.5 INCREASEWEIGHTS Procedure

The INCREASEWEIGHTS procedure is also similar to the INCREASEWEIGHTS procedure introduced in [1], with the slight difference that it inactivates the points whose weights reach $\lambda$ after increasing their weights. In particular, the INCREASEWEIGHTS procedure computes a set $\mathcal{K}$ of points that are reachable from the surplus points by admissible alternating paths and increases the weights of those points by $\delta$. It then marks all points of $B$ whose weights have reached $\lambda$ as inactive and recomputes $A_\delta, \mathcal{G}_\delta$, and $\hat{\tau}_\delta$. For completeness, we include the description of the INCREASEWEIGHTS procedure below.

For each point $r \in A_\delta$, let $\mathcal{N}(r) \subseteq B$ denote the set of points $b \in B$ with $\hat{\tau}_\delta(r, b) > 0$. Let $\mathcal{S} \subset B$ denote the set of active surplus points of $B$. Mark all points $b \in B$ and all forward edges $(b, r)$ in the residual graph as unvisited and set $\mathcal{K} = \emptyset$. Let $U = B$ denote the set of unvisited points of $B$. Initiate a partial DFS from any point $b \in \mathcal{S}$ in the residual graph $\mathcal{G}_\delta$ by setting $Q := \langle b = b_0 \rangle$ as the search path that the algorithm grows. Execute the following steps.

1. If $Q = \langle b = b_0, r_1, \ldots, b_i \rangle$,
   (a) If there exists an unvisited forward edge $(b_i, r)$ in $\mathcal{G}_\delta$, add $r$ to $Q$ as $r_{i+1}$.
   (b) Otherwise, mark $b_i$ as visited and remove $b_i$ from $U$ and $Q$. Add $b_i$ to $\mathcal{K}$.
2. If $Q = \langle b = b_0, r_1, \ldots, b_i, r_i \rangle$, let $b^* := \arg\max_{b' \in U \cap \mathcal{N}(r_i)} \mathrm{d}_y(r_i, b')$.
   (a) If $\mathrm{d}_y(r_i, b^*) > \mathrm{d}_y(r_i, b_i)$, i.e., $(b_i, r_i, b^*)$ is admissible, add $b^*$ as $b_{i+1}$ to $Q$.
   (b) Otherwise, remove $r_i$ from $Q$ and mark $(b_i, r_i)$ as visited.

After all DFS procedures from all points in $\mathcal{S}$ terminate, for each point $b \in \mathcal{K}$, set $y(b) \leftarrow y(b) + \delta$. We next describe how to recompute the residual graph and the compressed transport plan with respect to the updated weights.

**Lemma E.7.** *Given a $\delta$-feasible transport plan $\tau_\delta, y(\cdot)$ satisfying condition (F1) in Lemma 4.1, after the execution of the INCREASEWEIGHTS procedure,*

*(IW1) the transport plan $\hat{\tau}_\delta, y(\cdot)$ remain $\delta$-feasible and condition (F1) remains satisfies,*

*(IW2) the weight of each active surplus point $b \in B$ increases by $\delta$, and*

*(IW3) the weights of all points $b \in B$ with deficit regions inside $RV_y^{2\delta}(b)$ and all inactive points remain unchanged.*

*Proof.* Let $y(\cdot), \mathcal{X}_\delta, A_\delta$ (resp. $y'(\cdot), \mathcal{X}'_\delta, A'_\delta$) denote the set of weights of $B$, decomposition of $A$, and the set of representative points before (resp. after) the execution of the INCREASEWEIGHTS procedure.

To prove property (IW1), we show that for any pair of points $(a, b) \in A \times B$ such that $\tau_\delta(a, b) > 0$, we have $a \in RV_{y'}^{2\delta}(b)$. Let $b_a$ (resp. $b'_a$) denote the weighted nearest neighbor of $a$ with respect to weights $y(\cdot)$ (resp. $y'(\cdot)$). From the $\delta$-feasibility of $\tau_\delta, y(\cdot)$, we have $\mathrm{d}_y(a, b) \leq \mathrm{d}_y(a, b'_a) + 2\delta$. Let $r_a$ denote the representative point of the region containing $a$.

- If $b \in \mathcal{K}$, then $b_a$ also is included in $\mathcal{K}$, since either $b = b_a$ or the triple $(b_a, r_a, b)$ would be admissible. Hence, $b_a$ remains the weighted nearest neighbor of $a$ with respect to weights $b'_a$, and
$$\mathrm{d}_{y'}(a, b) = \mathrm{d}_y(a, b) - \delta \leq \mathrm{d}_y(a, b_a) + \delta = \mathrm{d}_{y'}(a, b_a) + 2\delta.$$

- Otherwise, $b \notin \mathcal{K}$. In this case,
  - if $b'_a \notin \mathcal{K}$, then
    $$d_{y'}(a, b) = d_y(a, b) \leq d_y(a, b'_a) + 2\delta = d_{y'}(a, b'_a) + 2\delta.$$
  - otherwise, we would have $d_y(r_a, b'_a) \geq d_y(r_a, b)$, since otherwise $(b'_a, r_a, b)$ would be admissible and since $b'_a \in \mathcal{K}$, then $b \in \mathcal{K}$ as well. In this case, since $b'_a$ became the weighted nearest neighbor of $r_a$, we should have $r_a \in RV_y^\delta(b'_a)$, and since $d_y(r_a, b'_a) \geq d_y(r_a, b)$, we should also have that $r_a \in RV_y^\delta(b)$ and consequently $a \in RV_y^\delta(b)$. Then,
    $$d_{y'}(a, b) = d_y(a, b) \leq d_y(a, b'_a) + \delta = d_y(a, b'_a) + 2\delta.$$

Therefore, for any pair $(a, b)$ with $\tau_\delta(a, b) > 0$, we have $a \in RV_y^{2\delta}(b)$, and $\tau_\delta, y'(\cdot)$ would be $\delta$-feasible, i.e., (IW1) holds.

Note that the INCREASEWEIGHTS procedure initiates a partial DFS from each active surplus point, which terminates when the search path becomes empty; hence, each active surplus point $b \in B$ will be added to the set $\mathcal{K}$ in step 1(a) of the procedure, and its weight will be increased by $\delta$, proving (IW2).

Finally, note that from Lemma E.7, there are no admissible augmenting paths or alternating paths from active surplus points to inactive points in the residual graph. Hence, the partial DFS procedure from the active surplus points cannot reach any inactive point or any point $b \in B$ with deficit regions inside $RV_y^{2\delta}(b)$; therefore, all such points would not be in the set $\mathcal{K}$ and their weights remain unchanged, leading to (IW3).

$\square$

## E.6    ACYCLIFY Procedure

Given a $\delta$-feasible transport plan $\tau_\delta, y(\cdot)$ obtained after the execution of REDUCEWEIGHTS or INCREASEWEIGHTS procedure, the ACYCLIFY procedure updates the transport plan such that, for the implicit representation $\hat{\tau}_\delta$ of $\tau_\delta$, the transport plan is a forest, i.e., the graph containing all edges $(r, b) \in A_\delta \times B$ does not contain any edges. Furthermore, the ACYCLIFY procedure makes sure that there are no admissible alternating cycles in the residual graph. This procedure is identical to the ACYCLIFY procedure presented in [1]. The procedure first uses a dynamic tree structure Sleator and Tarjan [45] to make the transport plan a forest (using phase 1). The procedure then uses partial DFS to follow the admissible triples and remove any admissible alternating cycles (using phase 2). The procedure finally makes the transport plan a forest again (using phase 1). We summarize the two main phases of the procedure next.

**Phase 1: Acyclifying the Transport Plan.**    To make the transport plan a forest, the procedure initializes an empty transport plan $\hat{\tau}$ and iteratively adds the edges transporting mass in $\hat{\tau}_\delta$ to $\hat{\tau}$. The procedure also maintains a dynamic tree data structure and for any new edge, if the edge creates a cycle of transportation, the ACYCLIFY procedure cancels the cycle right away. More precisely, after adding an edge $(r_1, b_1)$ to the transport plan, if a cycle $(b_1, r_1, b_2, \ldots, b_k, r_k)$ is formed such that $\hat{\tau}(r_i, b_i) > 0$ and $\hat{\tau}(b_i, r_{i-1}) > 0$ for all $i \in [1, k]$ (assuming $r_0 = r_k$), then the procedure increases (resp. decreases) the amount of mass transported on the even-indexed (resp. odd-indexed) edges of this cycle until at least one of the odd-indexed edges gets removed from the transport plan. Using the dynamic tree structure by Sleator and Tarjan [45], each of the above-mentioned operations takes $O(\log n)$ amortized time; therefore, using Lemma E.3, since the number of edges of the residual graph is $O(n^{d+1})$, this phase takes a total of $O(n^{d+1} \log n)$ time.

**Phase 2: Acyclifying the Admissible Triples.**    This step uses a partial DFS on the reverse of the residual graph $\overleftarrow{\mathcal{G}}_\delta$ to remove any admissible cycles from the residual graph. For any admissible cycle $C = \langle b_1, r_1, \ldots, b_i, r_i, b_{i+1} = b_1 \rangle$ in $\overleftarrow{\mathcal{G}}_\delta$, define the *bottleneck capacity* of $C$ as $\min_{j \in [1,i]} \hat{\tau}(r_j, b_j)$. The procedure *cancels* $C$ by increasing (resp. decreasing) the amount of mass transported along the forward (resp. backward) edges of $C$ by the bottleneck capacity. The update will remove at least one of the transporting edges, hence removing the admissible cycle from the residual graph.

Mark all points of $B$ and all backward edges as unvisited. Define $U := B$ as the set of unvisited points. For any unvisited point $b \in B$, initiate a partial DFS in the reversed residual graph $\overleftarrow{\mathcal{G}}_\delta$ by setting $Q = \langle b = b_1 \rangle$ and search as follows until $Q$ becomes empty.

1. If $Q = \langle b_1, r_1, \ldots, b_i \rangle$,
    (a) If there exists an unvisited backward edge $(b_i, r_\varphi)$, add $r_\varphi$ as $r_{i+1}$ to $Q$.
    (b) Otherwise, mark $b_i$ as visited and remove $b_i$ from $Q$ and $U$.
2. If $Q = \langle b_1, r_1, \ldots, b_i, r_i \rangle$, let $b := \arg\min_{b' \in U} \mathrm{d}_y(r_i, b')$.
    (a) If $\mathrm{d}_y(r_i, b) < \mathrm{d}_y(r_i, b_i)$, i.e., $(b, r_i, b_i)$ is admissible, if $b$ lies on the search path $Q$ as $b_j$, then the cycle $C = \langle b_j, r_j, \ldots, b_i, r_i, b_{i+1} = b = b_j \rangle$ is admissible. Cancel the cycle $C$ and set $Q = \langle b_1, r_1, \ldots, b_j \rangle$. Otherwise, add $b$ as $b_{i+1}$ to $Q$.
    (b) Otherwise, $(b, r_i, b_i)$ is not admissible. Remove $r_i$ from $Q$ and mark the backward edge $(b_i, r_i)$ in $\overleftarrow{\mathcal{G}}_\delta$ as visited.

**Lemma E.8.** *Given a $\delta$-feasible transport plan $\tau_\delta, y(\cdot)$ satisfying condition (F1) in Lemma 4.1, after the execution of the* ACYCLIFY *procedure,*

*(A1)  the transport plan $\hat{\tau}_\delta, y(\cdot)$ remains $\delta$-feasible and condition (F1) remains satisfies, and*

*(A2)  the transport plan $\hat{\tau}_\delta$ is a forest and there are no admissible cycles in the residual graph.*

*Proof.* We begin by proving (A1). The ACYCLIFY procedure first makes the transportation graph acyclic. Since the edges carrying mass in the new transport plan is a subset of the original transportation network prior to applying ACYCLIFY , for each edge $(r, b) \in A_\delta \times B$ that transports mass in the new transport plan, $r \in RV_y^{2\delta}(b)$, which implies that the transport plan produced after this phase remains $\delta$-feasible. In the second phase of the procedure, at any time, all triples in the search path are admissible. Thus, updating the transport plan along any one of such cycles does not violate the $\delta$-feasibility conditions.

Furthermore, since the amount of mass transported to each region remains the same at all steps of the procedure, any point $r \in A_\delta$ that is a deficit in the new transport plan is also a deficit with respect to the transport plan prior to this phase of ACYCLIFY procedure. Since the procedure does not change the weights of the points, if (F1) holds prior to the execution of the ACYCLIFY procedure, any deficit region would have a zero weight, and condition (F1) continues to be true, i.e., (A1) holds.

To argue that property (A2) also holds, we aim to demonstrate:

(A3)  any point $b \in B$ (and any backward edge $(r, b)$) that is marked as visited does not participate in any admissible cycle in the same execution of the ACYCLIFY procedure, and

(A4)  the portion of the residual graph consisting of visited points and their adjacent regions does not include any admissible cycles. Moreover, there are no admissible paths from an unvisited point to any visited point.

Assuming (A3) is true, we conclude that no visited point $b \in B$ becomes part of an admissible cycle. Since the ACYCLIFY algorithm terminates only when all elements of $B$ have been marked as visited, this implies that no admissible cycles exist in the residual graph at termination. Additionally, since the procedure runs phase 1 once more after the execution of phase 2, the transportation network becomes acyclic, leading to (A2).

Furthermore, as discussed next, property (A3) naturally follows from (A4): For any visited point $b \in B$, all points $b' \in B$ from which there is an admissible path to $b$ must also have already been visited (since the algorithm performs the search in the reversed residual graph). If (A4) holds, then no admissible cycle can be formed solely from visited nodes. Similarly, a backward edge $(r, b)$ is marked as visited only if, for every admissible triple $(b', r, b)$, the point $b'$ has already been marked as visited. If $b'$ does not participate in any admissible cycles, then neither can the triple $(b', r, b)$. Accordingly, we prove (A4) below, which proves (A3) and consequently, proves (A2).

We use induction to show that (A4) holds throughout the second phase. Let $C^i, i \in [1, k]$ denote the sequence of admissible cycles computed by the procedure, and let $\hat{\tau}_\delta^0, \ldots, \hat{\tau}_\delta^k$ denote the sequence

of transport plans, where $\hat{\tau}_\delta^0$ is the transport plan maintained at the beginning of phase 2, and $\hat{\tau}_\delta^i$ is derived by updating $\hat{\tau}_\delta^{i-1}$ along $C^i$. Initially, at the start of the second phase, property (A4) clearly holds.

A point $b \in B$ is marked as visited when a search from $b$ does not lead to the formation of an admissible cycle. This means that for all admissible triples $(b', r, b)$, the point $b'$ and backward edge $(r, b)$ must already be marked as visited. Consequently, all points that can reach $b$ via an admissible path (all points that are reachable from $b$ by our backward DFS) are also visited, and $b$ has no admissible paths to any visited points in the residual graph. Therefore, if (A4) holds prior to marking $b$ as visited, it will remain valid afterward. We next consider the case where a cycle $C^i$ is eliminated.

Any admissible cycle computed by the procedure is made up only of unvisited points of $B$. After canceling such a cycle, any newly created admissible triple $(b, r, b')$ will consist of a visited point $b$ and an unvisited point $b'$. This means that cycle cancellation can only produce new admissible paths from visited to unvisited points. Provided that (A4) is held before canceling $C^i$, canceling $C^i$ does not introduce any admissible cycles among visited points. Additionally, since cancellation does not generate any admissible triples directed from unvisited to visited points, there will be no admissible paths from any unvisited point to any visited point. Therefore, property (A4) is preserved after canceling any cycle $C^i$. □

### E.7 Missing Proofs of Efficiency

**Number of iterations of step 1.** We begin by showing that the number of iterations of step 1 of our algorithm is $O(n)$.

**Lemma 4.3.** *For any subset $S \subset B$, suppose in an iteration $i$ of step 1 of our algorithm, the reduction in the weight of any point in $S$ is more than $6\delta$ greater than the reduction in the weight of any point in $B \setminus S$, i.e., $\min_{b \in S} \gamma^i(b) > \max_{b' \in B \setminus S} \gamma^i(b') + 6\delta$. Then, there are no deficit regions inside the restricted Voronoi cells of the points in $S$.*

*Proof.* Let $\tau_{2\delta}, y_{2\delta}(\cdot)$ denote the $2\delta$-feasible transport plan computed by our algorithm at the end of scale $2\delta$. Recall that $\tau^i, y^i(\cdot)$ denotes the $\delta$-feasible transport plan maintained after iteration $i$ of the initialization step and $\gamma^i(b) = y_{2\delta}(b) - y^i(b)$ for each point $b \in B$. Define $\mathcal{R}_S^\delta := \bigcup_{b \in S} RV_{y^i}^{2\delta}(b)$ as the union of all $2\delta$-expanded restricted Voronoi cells of the points in $S$.

To prove this lemma, we show that all the continuous mass inside the set $\mathcal{R}_S^\delta$ is transported to the points in $S$ in the transport plan $\tau_{2\delta}$. Given that there are no deficit points with respect to $\tau_{2\delta}$, the total continuous mass inside the set $\mathcal{R}_S^\delta$ would be no more than the discrete mass at the points in $S$. Due to the $\delta$-feasibility conditions, $\tau^i$ transports the mass of the point in $S$ only to the regions in $\mathcal{R}_S^\delta$, i.e., the total mass transported from $S$ in $\tau^i$ is no more than $\nu(S) = \sum_{b \in S} \nu(b)$. Since the REDUCEWEIGHTS procedure does not reduce the weight of any surplus points, no points in $S$ can be surplus; hence, no points in $S$ can be deficit, and all points in $S$ are balanced. In the following, we show that the total continuous mass inside $\mathcal{R}_S^\delta$ is at most $\nu(S)$.

Consider any point $a \in \mathcal{R}_S^\delta$, and let $b_a \in B$ be any $4\delta$-weighted nearest neighbor of $a$ with respect to weights $y_{2\delta}(\cdot)$. Below, we show that $b_a \in S$. For the sake of contradiction, suppose $b_a \notin S$, and let $b' \in S$ denote any point in the set $S$. Note that $\mathrm{d}_{y_{2\delta}}(a, b_a) \leq \mathrm{d}_{y_{2\delta}}(a, b') + 4\delta$. By the lemma's assumption, $\gamma^i(b') < \gamma^i(b_a) + 6\delta$, since $b_a \notin S$. Therefore,

$$\mathrm{d}_{y^i}(a, b') = \mathrm{d}_{y_{2\delta}}(a, b') + y_{2\delta}(b') - y^i(b') \geq \mathrm{d}_{y_{2\delta}}(a, b_a) + \gamma^i(b') - 4\delta$$
$$= \mathrm{d}_{y^i}(a, b_a) - \gamma^i(b_a) + \gamma^i(b') - 4\delta > \mathrm{d}_{y^i}(a, b_a) + 2\delta.$$

In words, since $b_a \notin S$, with respect to weights $y^i(\cdot)$, the weighted distance of $b_a$ to $a$ would be more than $2\delta$ less than the weighted distance of $a$ to any point $b' \in S$; consequently, the point $a$ cannot lie inside the union of the $2\delta$ expanded restricted Voronoi cells of $S$ (i.e., $a \notin \mathcal{R}_S^\delta$), which is a contradiction. Hence, any $4\delta$-weighted nearest neighbor of $a$ with respect to weights $y_{2\delta}(\cdot)$ has to be in the set $S$.

Next, note that any point $a \in \mathcal{R}_S^\delta$ lies inside the $2\delta$-expanded Voronoi cell of at least one point in $S$ with respect to weights $y^i(\cdot)$. Since all points in $S$ have their weights reduced by at least $6\delta$ (from the lemma's assumption), the derived weight $\bar{y}_{2\delta}(a)$ with respect to weights $y_{2\delta}(\cdot)$ would be positive, and from the $2\delta$-feasibility of $\tau_{2\delta}, y_{2\delta}(\cdot)$, the point $a$ cannot be a free point.

Consequently, all continuous mass inside $\mathcal{R}_S^\delta$ is transported to the points in $S$ in $\tau_{2\delta}$, as claimed. $\qquad\square$

For any iteration $i$ of step 1, let $A_F^i \subset A_\delta$ (resp. $B_S^i \subset B$) denote the subset of violating deficit points (resp. surplus points) with respect to $\tau^i$. Note that step 1 terminates as soon as $A_F^i$ becomes empty. Since the total amount of continuous mass transported by $\tau^i$ is no more than the total discrete mass at the points in $B$, if $A_F^i \neq \emptyset$, then there exists at least one surplus point $b_s \in B_S^i$. Since step 1 does not create new surplus points (from Lemma E.4, balanced points remain balanced), the point $b_s$ was surplus in all previous iterations as well, and $\gamma^i(b_s) = 0$. Furthermore, from Lemma 4.2, the weight of the point $b_r \in B$ containing any deficit point $r \in A_F^i$ reduces by $\delta$ in each iteration of step 1, and therefore, $\gamma^i(b_r) = \delta i$. Consequently, if $i > 6n$, then $\gamma^i(b_r) - \gamma^i(b_s) > 6n\delta$, and there will be a subset $S \subset B$ such that $b_r \in S$ for all points $b_r \in B$ containing a violating deficit point $r \in A_F^i$, $b_s \notin S$ for all surplus points $b_s \in B_S^i$, and that satisfies the conditions of Lemma 4.3. In this case, from Lemma 4.3, there are no deficit regions inside the restricted Voronoi cells of the points in $S$, which is a contradiction. Hence, there will be no violating deficit points in $A_\delta$ after $6n$ iterations of step 1, i.e., step 1 will terminate after at most $6n$ iterations.

**Number of iterations of step 2.** Next, we show that step 2 of our algorithm in each scale runs $O(n)$ iterations. For any iteration $i$, support $\tau_{\text{step2}}^i, y_{\text{step2}}^i(\cdot)$ denote the $\delta$-feasible transport plan maintained after after iteration $i$ of step 2. Define $\gamma_{\text{step2}}^i(b) := y_{\text{step2}}^i(b) - y_{2\delta}(b)$.

**Lemma E.9.** *For any subset $S \subset B$, suppose in an iteration $i > 12n$ of step 2, the increase in the weight of any point $b \in S$ is more than $6\delta$ greater than the increase in the weight of any point $b' \in B \setminus S$, i.e., $\min_{b \in S} \gamma_{\text{step2}}^i(b) > \max_{b' \in B \setminus S} \gamma_{\text{step2}}^i(b') + 6\delta$. Then, all points in $S$ are balanced in $\tau_{\text{step2}}^i$.*

*Proof.* First, recall that step 2 terminates when there are no active surplus points. Hence, we assume that there exists an active surplus point $b_s \in B$ in iteration $i$. Since in each iteration, the weight of $b_s$ increases by $\delta$, $\gamma_{\text{step2}}^i(b_s) = i\delta$. Also, $b_s$ has the highest increase in weight and therefore, $b_s \in S$. Note that any point $b \in B$ that is surplus with respect to the transport plan $\tau_{\text{step2}}^i$ was also a surplus point in all previous iterations of steps 1 and 2. Hence, the weight of $b$ has not been reduced in step 1. Recall that if $b$ was a surplus point in $\tau_{2\delta}$, then $b$ has been deactivated by our algorithm, which means $y_{2\delta}(b) = \lambda$. However, step 1 has not reduced the weight of $b$, and therefore, the point $b$ would have been deactivated in step 2, i.e., the surplus point $b_s$ is not balanced in $\tau_{2\delta}$ as is balanced.

Second, note that any surplus point $b \in B$ with respect to $\tau_{2\delta}, y_{2\delta}(\cdot)$ has a weight $y_{2\delta}(b) = \lambda$, and since the number of iterations of step 1 is at most $6n$, $\gamma_{\text{step2}}^i(b_s) \geq \gamma_{\text{step2}}^i(b) + 6n\delta$, and $b$ cannot be in $S$. Therefore, all points $b \in S$ are balanced in $\tau_{2\delta}$.

Define $\mathcal{R}_S^\delta := \bigcup_{b \in S} RV_{y_{\text{step2}}^i}^{-2\delta}(b)$ as the union of all $2\delta$-shrunken restricted Voronoi cells of the points in $S$. From the $\delta$-feasibility of $\tau_{\text{step2}}^i, y_{\text{step2}}^i(\cdot)$, the continuous mass inside $\mathcal{R}_S^\delta$ is transported only to the point in $S$. To prove this lemma, we show that the amount of continuous mass inside the set $\mathcal{R}_S^\delta$ is no less than the discrete mass at the points in $S$. Assuming this, since there are no deficit points in $B$ in step 2, no points in $S$ can be surplus, and all points in $S$ are balanced. In the following, we show that the total continuous mass inside $\mathcal{R}_S^\delta$ is equal to $\nu(S)$.

Consider any point $b \in S$ and any point $a \in A$ such that $\tau_{2\delta}(a, b) > 0$. Below, we show that $a \in \mathcal{R}_S^\delta$. Let $b' \notin S$ be any other point. Since $\tau_{2\delta}(a, b) > 0$, then $b$ is a $4\delta$-weighted nearest neighbor of $a$ with weights $y_{2\delta}(\cdot)$ and therefore, $d_{y_{2\delta}}(a, b) \leq d_{y_{2\delta}}(a, b') + 4\delta$. Furthermore, since $b' \notin S$, $\gamma_{\text{step2}}(b) > \gamma_{\text{step2}}(b') + 6\delta$. Therefore,

$$d_{y_{\text{step2}}^i}(a, b) = d_{y_{2\delta}}(a, b) + y_{2\delta}(b) - y_{\text{step2}}^i(b) \leq d_{y_{2\delta}}(a, b') + 4\delta - \gamma_{\text{step2}}^i(b)$$

$$= d_{y_{\text{step2}}^i}(a, b') + \gamma_{\text{step2}}^i(b') + 4\delta - \gamma_{\text{step2}}^i(b) < d_{y_{\text{step2}}^i}(a, b') - 2\delta.$$

In words, with respect to weights $y_{\text{step2}}^i(\cdot)$, the weighted distance of $b$ to $a$ would be more than $2\delta$ less than the weighted distance of $a$ to any point $b' \notin S$, i.e., $a \in \mathcal{R}_S^\delta$.

Consequently, all continuous mass inside $\mathcal{R}_S^\delta$ is transported to the points in $S$ in $\tau_{2\delta}$, as claimed. Since all points in $S$ are balanced in $\tau_{2\delta}$, the total discrete mass at the points of $S$ are no more than

the continuous mass inside $\mathcal{R}_S^\delta$, and since there are no deficit points in $B$, there are also no surplus points in $S$ and all points in $S$ are balanced. $\qquad\square$

From Lemma E.7, the weight of each active surplus point $b \in B$ increases by $\delta$ in each iteration of step 2. Furthermore, note that for any point $b \in B$ that had a weight of $\lambda$ in the previous scale (e.g., the surplus points of $B$ in $\tau_{2\delta}$), the weight of $b$ has been reduced in step 1 by at most $6n\delta$, and in step 2, it can increase by at most $6n\delta$. Assume, for the sake of contradiction, that the number of iterations of step 2 has exceeded $12n$, and consider any iteration $i > 12n$. In this case, for any surplus point $b_s \in B$, since $b_s$ was a surplus point in all previous iterations, including the iterations of step 1, we have

$$\gamma_{\text{step2}}^i(b_s) = y_{\text{step2}}^i(b_s) - y_{2\delta}(b_s) = (y_{\text{step2}}^i(b_s) - y^i(b_s)) - (y_{2\delta}(b_s) - y^i(b_s)) = 12n\delta - 0 = 12n\delta.$$

Furthermore, for any inactive point $b' \in B$ with respect to $\tau_{2\delta}, y_{2\delta}(\cdot)$, after increasing the weight of $b'$ by at most $6n\delta$, it becomes inactive; hence, the difference in the increases in the weights of $b_s$ and $b'$ is more than $6n\delta$, and there exists a subset $S$ of $B$ where all active surplus points are in $S$ and all inactive points are in $B \setminus S$. However, from Lemma E.9, all points in $S$ have to be balanced, which is a contradiction. Therefore, the step 2 of our algorithm terminates in at most $12n$ iterations.

**Efficiency of the SEARCHANDCONSOLIDATE and SEARCHANDAUGMENT Procedures.** Each execution of the SEARCHANDCONSOLIDATE (resp. SEARCHANDAUGMENT ) procedure runs partial DFS procedures on the residual graph to find a set of admissible augmenting and consolidating (resp. alternating and augmenting) paths. The partial DFS procedure, upon backtracking from a point $b \in B$ (resp. $r \in A_\delta$), marks the point $b$ (resp. the backward edge $(b', r)$ used to reach $r$) as visited and does not add it to the search path again in the same execution. Upon finding an admissible path $P$, the procedure updates the transport plan along $P$ in $O(|P|)$ time. Let $\{P_1, \ldots, P_k\}$ denote the set of all paths found by the SEARCHANDCONSOLIDATE (resp. SEARCHANDAUGMENT ) procedure. Since the residual graph has $O(n^{d+1}$ edges (from Lemma E.3), the running time of the procedure would be $O(n^{d+1} + \sum_{i=1}^k |P_i|)$. Next, we bound the total length of all paths found by the procedures.

**Lemma E.10.** *The total length of augmenting and consolidating paths computed during the execution of the* SEARCHANDCONSOLIDATE *procedure is $O(n^{d+1})$ in $d$ dimensions.*

*Proof.* Let $\hat{\tau}_\delta^0$ denote the transport plan maintained by the algorithm at the beginning of the execution of the SEARCHANDCONSOLIDATE procedure. To prove this lemma, we categorize the augmenting and consolidating paths found by the procedure based on the source of their bottleneck capacity, namely (1) set $\mathcal{P}_v$ consisting of augmenting and consolidating paths whose bottleneck capacity is determined based on the residual capacity of its endpoints, and (2) set $\mathcal{P}_e$ consisting of paths whose bottleneck capacity is determined based on mass transportation over its backward edges. We begin by showing that $|\mathcal{P}_v| = O(n^d)$ and then show $|\mathcal{P}_e| = O(n^d)$. Since each path has a length of at most $2n$ (since the number of points in $B$ is $n$), we then conclude that the total length of all paths computed by the procedure is $O(n^3)$.

Let $P$ be a consolidating path in $\mathcal{P}_v$. If the bottleneck capacity of $P$ is determined by its free endpoint $r \in A_\delta$, then the mass of $r$ will be fully transported after updating the transport plan along $P$; similarly, for any augmenting path $P \in \mathcal{P}_v$, if the bottleneck capacity of $P$ is determined by its free endpoint $b \in B$ (resp. $r \in A_\delta$), then $b$ (resp. $r$) will be balanced after augmentation. Therefore, $|\mathcal{P}_v| = O(n^2)$ since $|A_\delta \cup B| = O(n^2)$. Next, let $P$ be a path in $\mathcal{P}_e$; in this case, the backward edge $(r, b)$ determining the bottleneck capacity of $P$ will be removed from the transport plan after augmentation.

Consider any triple $(b, r, b')$ that is admissible in $\mathcal{G}'$ but not in $\mathcal{G}$. Recall that by the definition of the admissible triples, $(r, b')$ is a backward edge in $\mathcal{G}'$ and $\mathrm{d}_y(r, b) > \mathrm{d}_y(r, b')$. Since the SEARCHANDCONSOLIDATE procedure does not change the weights $y(\cdot)$, the only case where $(b, r, b')$ is not admissible in $\mathcal{G}$ is when $(r, b')$ is not a backward edge in $\mathcal{G}$, i.e., the pair $(b', r)$ is in $P$ as a forward edge, and updating $\hat{\tau}_\delta$ along $P$ results in transporting mass from $b'$ to $r$. On the other hand, by step 2(b) of the SEARCHANDCONSOLIDATE procedure, a forward edge $(b', r)$ will be added to the search path only if $b'$ is the weighted nearest unvisited neighbor of $r$; in other words, since $\mathrm{d}_y(r, b) > \mathrm{d}_y(r, b')$ and the procedure added $b'$ to the search path (instead of $b$), the point $b$ was marked as visited by the procedure. Therefore, for any newly formed admissible triple $(b, r, b')$, point $b$ (resp. $b'$) is marked as visited (resp. unvisited).

By property (SC2) in Lemma E.4, the point $b'$ cannot form an admissible augmenting path during the same execution of the SEARCHANDCONSOLIDATE procedure. Hence, the edge $(r, b)$ determining the bottleneck capacity of $P$ was a backward edge of the initial transport plan $\hat{\tau}_\delta^0$ and updating the transport plan along each path $P \in \mathcal{P}_e$ removes one of the transporting edges of the transport plan $\hat{\tau}_\delta^0$. Since the transport $\hat{\tau}_\delta^0$ is obtained from the ACYCLIFY procedure, using property (AC2) in Lemma E.8, the transport plan is a forest and the number of backward edges is $O(n^d)$; hence, $|\mathcal{P}_e| = O(n^d)$, as claimed. The total number of augmenting paths found by the procedure, therefore, is $O(n^d)$, and since each augmenting path has a length of at most $2n$, their total length is $O(n^{d+1})$. $\qquad\square$

Using an identical argument, one can show that the total length of all alternating and augmenting paths found by the SEARCHANDAUGMENT procedure is $O(n^{d+1})$. Therefore, each execution of the SEARCHANDCONSOLIDATE and SEARCHANDAUGMENT procedures takes $O(n^{d+1})$ time.

**Efficiency of the REDUCEWEIGHTS and INCREASEWEIGHTS Procedures.** The RE-DUCEWEIGHTS and INCREASEWEIGHTS procedures run a DFS that visits each edge of the residual graph at most once and has a total running time of $O(n^{d+1})$ $d$ dimensions. Furthermore, in the arrangement used to construct partitioning $\mathcal{Y}$, each point $b \in B$ has at most 6 restricted Voronoi cells (three cells that are used in the construction of $\mathcal{X}_\delta$ and three that are used in the construction of $\mathcal{X}_\delta'$). Similar to our proof for the size of the graph, one can show that the total number of vertices in the arrangement used to construct $\mathcal{Y}$ is $O(n^d)$, and the number of regions in $\mathcal{Y}$ is at most $O(n^d)$. The construction of the transport plan $\hat{\tau}$, therefore, can be done in $O(n^d(\Phi + n))$ time since

(1) the mass of all regions in $\mathcal{Y}$ can be determined using the oracle in $O(n^d\Phi)$ time, and

(2) the mass transported on each pair $(\varrho, b) \in \mathcal{Y} \times B$ can be determined in $O(1)$ time.

Converting $\hat{\tau}$ to $\hat{\tau}_\delta$, as is done in the merge step, also takes $O(n^{d+1})$ time, since there are $O(n^{d+1})$ pairs of points in $A_\delta \times B$, and therefore, the total complexity of $\hat{\tau}$ is $O(n^{d+1})$. Finally, storing a sorted list of neighbors for each region $r \in A_\delta$ takes $O(n^{d+1}) \log n$ time in total. Hence, the executions of the REDUCEWEIGHTS and INCREASEWEIGHTS procedures take $O(n^d(\Phi + n \log n))$ time.

**Efficiency of the ACYCLIFY Procedure.** In the first step, the ACYCLIFY procedure uses a dynamic tree structure to remove all cycles of transportation from the transport plan $\hat{\tau}_\delta$. Using the dynamic tree structure by Sleator and Tarjan [45], since the total number of edges of the graph is $O(n^{d+1})$ in $d$ dimensions, the running time of this step would be $O(n^{d+1} \log n)$. In the second step, the procedure runs a partial DFS procedure on the residual graph and cancels the admissible cycles. The partial DFS procedure, upon backtracking from a point $b \in B$ (resp. $r \in A_\delta$), marks the point $b$ (resp. the backward edge $(b', r)$ used to reach $r$) as visited and does not add it again to the search path in the same execution. Furthermore, upon finding an admissible cycle $C$, it cancels the cycle in $O(|C|)$ time. Let $\{C_1, \ldots, C_k\}$ denote the set of all cycles found in the execution of the ACYCLIFY procedure. Given that the size of the residual graph is at most $O(n^{d+1})$, the second step of the ACYCLIFY procedure takes a total of $O(n^{d+1} + \sum_{i=1}^{k} |C_i|)$ time. The following lemma bounds the total length of all cycles found by the ACYCLIFY procedure.

**Lemma E.11.** *The total length of admissible cycles computed during the execution of the second step of the* ACYCLIFY *procedure is $O(n^3)$ in 2 dimensions and $O(n^{d+1})$ time in $d$ dimensions.*

*Proof.* Let $\hat{\tau}_\delta^0$ denote the transport plan maintained by the algorithm after the first step of the ACYCLIFY procedure, i.e., $\hat{\tau}_\delta^0$ is a forest. To prove this lemma, we show that the ACYCLIFY procedure finds $O(n^d)$ admissible cycles, where each cycle has a length of at most $2n$; hence, the total length of all cycles found by the procedure would be $O(n^{d+1})$.

Let $C$ be an admissible cycle found by the procedure; in this case, the backward edge determining the bottleneck capacity of $C$ will be removed from the transport plan after cancellation. For any admissible triple $(b, r, b')$ formed after canceling $C$, using an identical argument as in Lemma E.10, one can show that the edge $(r, b')$ is a backward edge that was on the cycle $C$ as a forward edge, and the point $b$ is marked as visited; hence, by Lemma E.8, the point $b$ does not form an admissible cycle in the same execution of the ACYCLIFY procedure and therefore, the newly formed backward edge

$(r, b')$ cannot be included in any admissible cycles. Therefore, each cycle cancellation removes one of the backward edges of $\hat{\tau}_\delta^0$, where $\hat{\tau}_\delta^0$ is a forest and the number of its transporting edges is $O(n^d)$. Therefore, the total number of cycles found by the ACYCLIFY procedure is $O(n^d)$, and their total length is $O(n^{d+1})$, as claimed. □

From Lemma E.11, the total execution time of the ACYCLIFY procedure is $O(n^{d+1})$ in $d$ dimensions.

# F   Full Algorithm Details from Section A

We first make some necessary definitions and useful notations for the remainder of the section.

Let $\mathcal{B}(c, r)$ denote the Euclidean ball of radius $r$ centered at $c$, and let $\mathbb{G}_\delta = \left\{[0, \delta]^d + \vec{z} : \vec{z} \in \delta\mathbb{Z}^d\right\}$ be the set of axis-aligned grid cells of side length $\delta$. For any hypercube $\square$, let $B_\square := B \cap \square$ denote the set of points of $B$ in $\square$, $c_\square$ denote the center of $\square$, and $\ell_\square$ denote the sidelength of $\square$. Given a hypercube grid cell $\square = \prod_{i=1}^d [a_i, a_i + \delta]$ from $\mathbb{G}_\delta$, we define the set of its child cells as $\mathsf{C}[\square] := \left\{\prod_{i=1}^d [a_i, a_i + \frac{\delta}{2}] + \vec{v} : \vec{v} \in \{0, \frac{\delta}{2}\}^d\right\}$. Additionally, given a tree $T = (V, E)$, we use $\mathrm{pa}(v)$ and $\mathsf{C}[v]$ to denote the parent of $v$ and the set of children of $v$ for each $v \in V$.

We define any pair of sets $P, Q \subseteq \mathbb{R}^d$ as $\varepsilon$-*well separated* if

$$\max\left\{\max_{p_1, p_2 \in P} ||p_1 - p_2||, \max_{q_1, q_2 \in Q} ||q_1 - q_2||\right\} \leq \varepsilon \cdot \min_{p, q \in P \times Q} ||p - q||.$$

Then given a set $S \subseteq \mathbb{R}^d$, a $\varepsilon$-*well separated pair decomposition* of $S$ is a set of pairs of subsets $W = \{(P_1, Q_1), \ldots, (P_k, Q_k)\}$ of $S$ such that (1.) the pair $(P_i, Q_i)$ is $\varepsilon$-well separated for all $1 \leq i \leq k$, and (2.) for every $p, q \in S$, there exists a unique $j$ such that $p \in P_j$ and $q \in Q_j$. It is known that an $\varepsilon$-well separated pair decomposition of a set of $n$ points in $\mathbb{R}^d$ of size $O(n\varepsilon^{-d})$ can be constructed in $O(n(\varepsilon^{-d} + \log n))$ time, e.g. [13].

We divide the remainder of this section to describe each of the components of the algorithm in Section A.

## F.1   Construction of discrete distribution

We first describe the construction of the collection of hypercubes. Compute an $\varepsilon$-well separated pair decomposition $W$ of the discrete point set $B$. For each pair $(B_1, B_2) \in W$, choose arbitrary representative points $b_1 \in B_1$ and $b_2 \in B_2$. Define $\delta_i = 2^i \varepsilon ||b_1 - b_2||$ for all $i \in \mathbb{Z}$. Then let $\mathcal{P}_i(B_1, B_2)$ be the set of axis-aligned grid cells $\square$ from $\mathbb{G}_{\varepsilon\delta_i}$ where $c_\square \in \mathcal{B}(b_1, \delta_i) \cup \mathcal{B}(b_2, \delta_i)$, and initially set $\mathcal{P} = \bigcup_{(B_1, B_2) \in W} \bigcup_{i=0}^{2\log \frac{1}{\varepsilon}} \mathcal{P}_i(B_1, B_2)$. For every $\square \in \mathcal{P}$ such that there exists some $\square' \subset \square$ also in $\mathcal{P}$, we replace $\square$ with its $2^d$ child cells in $\mathcal{P}$. For convenience, we will denote $P = \bigcup_{\square \in \mathcal{P}} \square$ as the set of points covered by $\mathcal{P}$. We conclude with a set of hypercubes $\mathcal{P}$ with the following desired properties.

**Lemma F.1.** $|\mathcal{P}| = O(n\varepsilon^{-2d} \log \varepsilon^{-1})$ *and can be constructed in* $O(n(\varepsilon^{-2d} \log \varepsilon^{-1} + \log n))$ *time. Moreover, every* $\square$ *in* $\mathcal{P}$ *satisfies one of the following two properties: (1.)* $\ell_\square < 2\varepsilon \min_{b \in B} ||b - c_\square||$, *or (2.)* $||b - c_\square|| < \varepsilon \min_{b' \neq b} ||b - b'||$ *and* $\ell_\square \leq \frac{\varepsilon^2}{1-\varepsilon} \min_{b' \neq b} ||b' - c_\square||$ *for some* $b \in B$.

*Proof.* We first prove the bound $|\mathcal{P}| = O(n\varepsilon^{-2d} \log \varepsilon^{-1})$. Note that $|W| = O(n\varepsilon^{-d})$ and can be constructed in $O(n(\varepsilon^{-d} + \log n))$ time. Then by initial construction of $\mathcal{P} = \bigcup_{(B_1, B_2) \in W} \bigcup_{i=0}^{2\log \frac{1}{\varepsilon}} \mathcal{P}_i(B_1, B_2)$ and potential replacement of each hypercube $\square$ in $\mathcal{P}$ with at most its $2^d$ child cells, we observe that

$$|\mathcal{P}| \leq 2^d (n\varepsilon^{-d} \log \varepsilon^{-1}) \cdot \max_{B_1, B_2, i} |\mathcal{P}_i(B_1, B_2)|.$$

Then one can argue that $|\mathcal{P}_i(B_1, B_2)| = O(\varepsilon^{-d})$ for all $i$ and all $(B_1, B_2) \in W$ since at most $O(\varepsilon^{-d})$ interior-disjoint hypercubes of sidelength $\varepsilon \cdot r$ can fit inside the ball of radius $r$. We require $O(|\mathcal{P}_i(B_1, B_2)|)$ time to find the grid cells in each set $\mathcal{P}_i(B_1, B_2)$.

We now prove that one of the two properties holds for each hypercube $\square$ in $\mathcal{P}$. Let $\square \in \mathcal{P}$ be arbitrarily chosen. Suppose $||b - c_\square|| < \varepsilon \min_{b' \neq b} ||b' - b||$ for some $b \in B$. Let $b^*$ be an arbitrary element of $\arg \min_{b' \neq b} ||b' - b||$, and let $(P, Q)$ be the pair in well-separated pair decomposition $W$ such that $b \in P$ and $b^* \in Q$. Then it must be the case that $\square$ is an element of $\mathcal{P}_0(P, Q)$, which implies $\ell_\square \leq \varepsilon^2 ||b^* - b||$. Let $b' \neq b$ be an arbitrary element of $B$. Then observe $||b^* - b|| \leq ||b' - b||$ by definition of $b^*$. But by triangle inequality, $||b' - c_\square|| \geq ||b' - b|| - ||b - c_\square|| \geq (1 - \varepsilon)||b' - b||$. Combining gives

$$\ell_\square \leq \varepsilon^2 ||b' - b|| \leq \frac{\varepsilon^2}{1 - \varepsilon} \cdot ||b' - c_\square||.$$

We note $b' \neq b$ was arbitrarily chosen.

Now suppose there does not exist any $b \in B$ such that $||b - c_\square|| < \varepsilon \min_{b' \neq b} ||b' - b||$. Then for every $b_1, b_2 \in B$, there exists an $i \in \mathbb{Z}$ such that $i > 0$ and $c_\square \in \mathcal{B}(b_1, \delta_i) \setminus \mathcal{B}(b_1, \delta_{i-1})$ where again $\delta_i = 2^i \varepsilon ||b_1 - b_2||$. If there exists a $b_2 \in B$ such that $i \leq 2 \log \varepsilon^{-1}$, then by construction of $\mathcal{P}$ we have that $\ell_\square \leq \varepsilon \delta_i < 2\varepsilon ||b_1 - c_\square||$.

If instead $i > 2 \log \varepsilon^{-1}$ for all $b_2$ then $c_\square \notin \mathcal{B}(b_1, \frac{1}{\varepsilon} \max_{b_2 \neq b_2} ||b_2 - b_1||)$. This implies $||b_1 - c_\square|| > \frac{\Delta}{2\varepsilon}$ since by triangle inequality one can argue that for every $b_1 \in B$ there exists a $b_2 \in B$ with $||b_1 - b_2|| \geq \frac{\Delta}{2}$. But since $\square \in \mathcal{P}$, we note that there must exist a $b_3 \in B$ such that $c_\square \in \mathcal{B}(b_3, \frac{1}{\varepsilon} \max_{b_4 \neq b_3} ||b_4 - b_3||)$ and $\ell_\square \leq \max_{b_4 \neq b_3} ||b_4 - b_3|| \leq \Delta$. Conclude that $\ell_\square \leq \Delta < 2\varepsilon ||b_1 - c_\square||$. $\qquad \square$

Once we have constructed the set of hypercubes $\mathcal{P}$, we create a discrete distribution in the following way. For each $b \in B$, define the $\varepsilon$-*neighborhood of* $b$ as the set

$$\mathcal{N}_\varepsilon(b) = \{\square \in \mathcal{P} : ||b - c_\square|| \leq \varepsilon \min_{b' \neq b} ||b - b'||\}$$

of hypercubes in $\mathcal{P}$ that are much closer to $b$ than any other point in $B$. We will use the notation $N_\varepsilon(b) = \bigcup_{\square \in \mathcal{N}_\varepsilon(b)} \square$ to denote the set of points covered by $\mathcal{N}_\varepsilon(b)$ for convenience. Then define $\hat{\mu}(b) = \int_{N_\varepsilon(b)} \mu(a) da$ for each $b \in B$ and $\hat{\mu}(c_\square) = \int_\square \mu(a) da$ for each hypercube $\square \in \mathcal{P} \setminus \bigcup_{b \in B} \mathcal{N}_\varepsilon(b)$. Finally choose an arbitrary point $r$ from $\mathbb{R}^d \setminus P$ and define $\hat{\mu}(r) := 1 - \int_P \mu(a) da$ as the leftover mass from $\mu$ not covered by $\mathcal{P}$. This completes the construction of the discrete distribution $\hat{\mu}$.

## F.2  Reduction to $\lambda$-capped OT

Mukherjee et al. [34] showed that for the discrete optimal transport problem, computing a $\lambda$-robust OT plan is equivalent to computing a complete transport plan under a capped distance function. We extend the same result to the semi-discrete OT problem.

Given a value $\lambda \geq 0$, define the $\lambda$-*capped distance function* $\hat{d}^\lambda(\cdot, \cdot)$ as $\hat{d}^\lambda(a, b) := \min\{d(a, b), \lambda\}$. For any complete semi-discrete transport plan $\tau^\lambda$ between $\mu$ and $\nu$ with respect to the distance function $\hat{d}^\lambda(\cdot, \cdot)$, we derive a partial semi-discrete transport plan $\tau$, referred to as a $\lambda$-*capped transport plan*, by removing all mass that is transported on edges with a distance at least $\lambda$. Formally, we define $\tau$ for each pair of points $(a, b) \in A \times B$ as $\tau(a, b) = \tau^\lambda(a, b)$ if $d(a, b) \leq \lambda$ and $\tau(a, b) = 0$ otherwise. We refer to the transport plan derived from a complete OT plan under the $\lambda$-capped distances by a $\lambda$-*derived OT plan*. The following lemma relates the $\lambda$-derived OT plans and $\lambda$-robust OT plans.

**Lemma F.2.** *Given a parameter $\lambda > 0$, let $\tau_\lambda$ be an $(\varepsilon, 0)$-approximate $\lambda$-capped transport plan, and let $\tau$ be a partial transport plan derived from $\tau_\lambda$. Then, $\tau$ is a $(\varepsilon, 0)$-approximate $\lambda$-robust OT plan between $\mu$ and $\nu$.*

*Proof.* Let $d_\lambda(a, b) := \min\{d(a, b), \lambda\}$ denote the $\lambda$-capped distance between $a$ and $b$. Since $\tau_\lambda$ is an $(\varepsilon, 0)$-approximate $\lambda$-capped transport plan, we note that for any arbitrary transport plan $\tau_\lambda^*$ where $M(\tau_\lambda^*) = 1$,

$$\sum_{b \in B} \int_A \tau_\lambda(a, b) d_\lambda(a, b) da \leq (1 + \varepsilon) \sum_{b \in B} \int_A \tau_\lambda^*(a, b) d_\lambda(a, b) da.$$

Now let $\tau^*$ denote the $\lambda$-derived transport plan from $\tau_\lambda^*$, i.e.

$$\tau^*(a,b) = \begin{cases} \tau_\lambda^*(a,b), & \mathrm{d}(a,b) \le \lambda, \\ 0, & \text{otherwise.} \end{cases}$$

Then it follows from the definition of $\lambda$-capped distances and derived transport plans that

$$\sum_{b \in B} \int_A \tau_\lambda(a,b) \mathrm{d}_\lambda(a,b) da = \sum_{b \in B} \int_A \tau(a,b) \mathrm{d}(a,b) da + \lambda \left( 1 - \sum_{b \in B} \int_A \tau(a,b) da \right) = w_\lambda(\tau),$$

and

$$\sum_{b \in B} \int_A \tau_\lambda^*(a,b) \mathrm{d}_\lambda(a,b) da = \sum_{b \in B} \int_A \tau^*(a,b) \mathrm{d}(a,b) da + \lambda \left( 1 - \sum_{b \in B} \int_A \tau^*(a,b) da \right) = w_\lambda(\tau^*).$$

Therefore, substituting into the first inequality at the beginning of the proof, we conclude

$$w_\lambda(\tau) \le (1+\varepsilon) w_\lambda(\tau^*).$$

The result follows from the fact that the transport plan $\tau_\lambda^*$ was chosen arbitrarily among transport plans where $\mathrm{M}(\tau_\lambda^*) = 1$. $\qquad \square$

### F.3 Algorithm for $\lambda$-Capped $p$-Wasserstein distance

To compute approximate discrete $\lambda$-capped $p$-Wasserstein distances (we equivalently consider $\mathrm{d}(a,b) = ||a-b||^p$ and do not take the $p^{\text{th}}$ root of the plan cost), we use a compressed directed graph construction of [2], cap distances at $\lambda$, run directed min-cost flow in near-linear time using [16], and shortcut flows to produce a transport plan. We describe the construction of the instance of minimum cost flow for completeness, and defer procedures for minimum cost flow [16] and shortcutting the flow to construct a transport plan [2].

**Sparse directed graph.** Suppose we are given two discrete point sets $\hat{A}, B \subset \mathbb{R}^d$ of at most $m$ points, with respective distributions $\hat{\mu}$ and $\nu$. We construct an $\varepsilon$-well separated pair decomposition $W$ of pairs of quadtree cells from quadtree $T$ on $\hat{A} \cup B$ as in e.g. [25]. The well-separated pair decomposition $W$ guarantees that for every pair of cells $(\Box_1, \Box_2) \in W$, $\frac{1}{2}\ell_{\Box_2} \le \ell_{\Box_1} \le 2\ell_{\Box_2}$. We replace each pair $(\Box_1, \Box_2)$ in $W$ where $\ell_{\Box_1} = 2\ell_{\Box_2}$ with the set $\bigcup_{\Box' \in \mathsf{C}[\Box_1]} (\Box', \Box_2)$ of pairs of $\Box_2$ with each child $\Box'$ of $\Box_1$. This guarantees that each pair of cells in $W$ is at the same level of the quadtree $T$ while only increasing the number of pairs in $W$ by a factor of $2^d$.

We now use the $\varepsilon$-well separated pair decomposition $W$ to construct a sparse graph. Make two copies of the quadtree $T = (V_T, E_T)$, called $T_1 = (V_1, E_1)$ and $T_2 = (V_2, E_2)$, where $E_1$ consists of only directed edges going up the tree and $E_2$ consists of only directed edges going down the tree. We note that $W$ consists of pairs of quadtree cells $(\Box_1, \Box_2)$ where $|\hat{A}_{\Box_1}| > 0$ and $|B_{\Box_2}| > 0$ by construction. For each $(\Box_1, \Box_2) \in W$, let $v_1 = \begin{cases} c_{\Box_1}, & |\hat{A}_{\Box_1}| > 1, \\ a, & \hat{A}_{\Box_1} = \{a\} \end{cases}$ and $v_2 = \begin{cases} c_{\Box_2}, & |B_{\Box_2}| > 1, \\ b, & B_{\Box_2} = \{b\} \end{cases}$ be representative points from $\Box_1$ and $\Box_2$, respectively. Let $E_3 = \{v_1 \to v_2 : (\Box_1, \Box_2) \in W\}$ be the set of directed edges from $v_1$ to $v_2$. Then the sparse graph we construct is $G = (V_G, E_G)$, where $V_G = V_1 \cup V_2$ and $E_G = E_1 \cup E_2 \cup E_3$.

**Lemma F.3.** *The graph $G$ constructed satisfies $|V_G| = O(n\varepsilon^{-2d} \log \Delta \log \varepsilon^{-1})$ and $|E_G| = O(n\varepsilon^{-3d} \log \Delta \log \varepsilon^{-1})$. Additionally, $G$ can be constructed in $O(n\varepsilon^{-3d} \log \Delta \log \varepsilon^{-1})$ time.*

*Proof.* The vertex set $V_G$ of our graph $G$ consists of two copies of centers of all quadtree cells in the quadtree $T$. We note that the quadtree $T$ on $m$ points with spread (or diameter if minimum pairwise distance is assumed to be 1) $\Delta$ has size at most $O(m \log \Delta)$. Then the quadtree built on the support of $\hat{\mu}$ (contains $B$) has size $O(n\varepsilon^{2d} \log \Delta \log \varepsilon^{-1})$ since the size of the support of $\hat{\mu}$ is bounded above by $|\mathcal{P}| + |B| = O(n\varepsilon^{-2d} \log \varepsilon^{-1})$ by Lemma F.1.

The edge set consists of two copies of each tree edge (one up and one down) plus one cross edge for each pair in the well-separated pair decomposition of the support of $\hat{\mu}$. Note that the size of

the support of $\hat{\mu}$ is $O(n\varepsilon^{-2d}\log\varepsilon^{-1})$, and therefore its $\varepsilon$-WSPD has size $O(n\varepsilon^{-3d}\log\varepsilon^{-1})$. By e.g. [26], the $\varepsilon$-WSPD can certainly be constructed in $O(n\varepsilon^{-2d}(\varepsilon^{-d}+\log\Delta)\log\varepsilon^{-1})$ time. Finally, the number of tree edges is bounded by the number of vertices, $O(n\varepsilon^{-2d}\log\varepsilon^{-1})$. Combining bounds on the number of edges of each type gives the desired result. $\qquad\square$

**Construction of minimum-cost flow instance.** As input, the minimum-cost flow problem is given a directed graph $G = (V, E)$ with edge costs $d_G\colon E \to \mathbb{R}$ and demands $\eta\colon V \to \mathbb{R}$ satisfying $\sum_{v\in V}\eta(v) = 0$. The solution to the minimum-cost flow problem is a flow, i.e. a function $f\colon E \to \mathbb{R}_{\geq 0}$ satisfying $\sum_{v:u\to v\in E} f(u \to v) - \sum_{w:w\to u\in E} f(w \to u) = \eta(u)$ for all $u \in V$, minimizing the weighted cost $\sum_{u\to v\in E} f(u \to v) \cdot d_G(u \to v)$. We have constructed our desired sparse graph $G$. to complete the instance of minimum-cost flow for which we call [16], we require edge costs $d_G$ and demands $\eta$.

Now given this sparse graph $G$, we define edge costs so that the cost of the path from $a$ to $b$ is bounded by $(1 + O(\varepsilon)) \cdot \min\{||a - b||^p, \lambda\}$. Note that there does not necessarily exist a path in $G$ between every $u$ and $v$ in $V$. However, by construction of the well-separated pair decomposition $W$ and edge set $E_G$, we guarantee that there exists a unique path from every $a \in \hat{A}$ to every $b \in B$. We define a height parameter $h_\lambda$ in a bottom-up manner. For each point $p \in \hat{A} \cup B$, we set the cost of the edge $p \to \text{pa}(p)$ in $E_1$ and the edge $\text{pa}(p) \to p$ in $E_2$ as

$$d_G(p \to \text{pa}(p)) = d_G(\text{pa}(p) \to p) := 0,$$

and additionally define $h_\lambda(\text{pa}(p)) = h_\lambda(p) := 0$. Then for each other vertex $v \in V_T$ in a bottom-up order, we define $h_\lambda(\text{pa}(v)) = \min\{\frac{\lambda}{2}, h_\lambda(v) + ||v - \text{pa}(v)||^p\}$. Set the cost of the edge $v \to \text{pa}(v)$ in $E_1$ and $\text{pa}(v) \to v$ in $E_2$ as

$$d_G(v \to \text{pa}(v)) = d_G(\text{pa}(v) \to v) := h_\lambda(\text{pa}(v)) - h_\lambda(v).$$

Note this edge cost $d_G$ is always nonnegative since $h_\lambda(\text{pa}(v)) \geq h_\lambda(v)$ for all $v \in V_T$. Then for each edge $u \to v \in E_3$, we define the cost of the edge as

$$d_G(u \to v) := \min\{\lambda - h_\lambda(u) - h_\lambda(v), ||u - v||^p\}.$$

For each $w, z \in V_G$ such that $w \to z$ is not an edge in $E_G$, we let $d_G(w, z)$ denote the shortest path distance between $w$ and $z$ in $G$.

**Lemma F.4.** *The graph $G$ and shortest path distance $d_G$ satisfy*

$$\left(1 - 2^{2p}\varepsilon\right)\min\{\lambda, ||a - b||^p\} \leq d_G(a, b) \leq \left(1 + \left(2^{2p} + 4d^{\frac{p}{2}}\right)\varepsilon\right)\min\{\lambda, ||a - b||^p\}$$

*for any constant $p \geq 1$ and all $a \in \hat{A} \cap V_1$ and $b \in B \cap V_2$.*

*Proof.* The structure of the proof will be as follows. We first prove the upper bound on $\mathbb{E}[d_G(a, b)]$, then prove the lower bound on $\mathbb{E}[d_G(a, b)]$.

1. **Upper bound.** Define the *canonical path* from $a$ to $b$ as the path $\Gamma_{\text{can}}(a \rightsquigarrow b) := a \rightsquigarrow_{T_1} c_{\square_1} \to c_{\square_2} \rightsquigarrow_{T_2} b$, where $u \rightsquigarrow_T v$ denotes the unique shortest path from $u$ to $v$ using only edges from tree $T$ and $(c_{\square_1}, c_{\square_2})$ is the edge added to $E$ for the WSPD pair $(P, Q)$ where $a \in P$ and $b \in Q$. By construction of the set of cross edges, there exists a unique such edge $(\square_1, \square_2)$ for each $(P, Q) \in W$. Moreover, let $\textcent(\Gamma) := \sum_{u\to v\in\Gamma} d_G(u, v)$ denote the cost of the path $\Gamma$ in $G$.

   We note $h_\lambda(\square) \leq \frac{\lambda}{2}$ for all $\square$ in the quadtree $T$ by construction of $h_\lambda$. Moreover, if $\textcent(\Gamma_{\text{can}}(a \rightsquigarrow b)) < \sum_{u\to v\in\Gamma_{\text{can}}(a\rightsquigarrow b)} ||u - v||$, then $\textcent(\Gamma_{\text{can}}(a \rightsquigarrow b)) = \lambda$ by construction of $d_G$ on any path between two leaves with exactly one cross edge. Then

$$d_G(a, b) \leq \textcent(\Gamma_{\text{can}}(a \rightsquigarrow b)) \leq \min\left\{\lambda, \sum_{u\to v\in\Gamma_{\text{can}}(a\rightsquigarrow b)} ||u - v||^p\right\}.$$

   To conclude the upper bound on $d_G(a, b)$, we prove $\sum_{u\to v\in\Gamma_{\text{can}}(a\rightsquigarrow b)} ||u - v|| \leq (1 + \varepsilon)||a - b||$.

   Consider the one cross edge $(\square_1, \square_2)$ of path $\Gamma_{\text{can}}(a \rightsquigarrow b)$. We note $(\square_1, \square_2) \in W$ by construction of the cross edges, and by construction of $W$, one has

$$||c_{\square_1} - c_{\square_2}||^p \leq (1 + 2\varepsilon)^p \cdot ||a - b||^p \leq (1 + 2^{2p}\varepsilon) \cdot ||a - b||^p.$$

Furthermore,

$$\sum_{u \to v \in a \rightsquigarrow_T c_{\square_1}} ||u - v||^p \le 2d^{\frac{p}{2}} \ell_{\square_1}^p \le 2d^{\frac{p}{2}} \varepsilon^p \cdot ||a - b||^p$$

since the length of each vertical edge is exponentially increasing and $W$ is a well-separated pair decomposition. Analogously, $\sum_{w \to z \in c_{\square_2} \rightsquigarrow_T b} ||w - z||^p \le 2d^{\frac{p}{2}} \varepsilon^p \cdot ||a - b||^p$. One can then conclude

$$\sum_{u \to v \in \Gamma_{\mathrm{can}}(a \rightsquigarrow b)} ||u - v|| \le \left(1 + \left(2^{2p} + 4d^{\frac{p}{2}}\right) \varepsilon\right) \cdot ||a - b||.$$

2. **Lower bound.** We prove the canonical path from $a$ to $b$ is the unique path from $a$ to $b$ for any $a \in \hat{A} \cap V_1$ and $b \in B \cap V_2$. Note to get from $a$ to $b$, at least one cross edge is required. Moreover, in $T_1$, the point $a$ can only reach vertices $c_\square$ for cells $\square$ in the quadtree $T$ where $a \in \square$. This is true since $T_1$ only has edges going up the tree. Similarly, in $T_2$, the point $b$ can only be reached by vertices $c_\square$ for cells $\square$ in the quadtree $T$ where $b \in \square$. But by assumption, each cross edge corresponds to a pair of cells in the WSPD $W$. Since there exists a unique pair of cells $(\square_1, \square_2)$ in $W$ where $a \in \square_1$ and $b \in \square_2$, there must exist a unique path in $G$ from $a$ to $b$ (going through the WSPD pair $(\square_1, \square_2)$).

With similar application as above of the fact that $\square_1, \square_2$ is a well-separated pair, we argue that the cost $d_G(a, b)$ is comparable to $||a - b||^p$. Note $d_G(a, b) = \cent(\Gamma_{\mathrm{can}}(a \rightsquigarrow b))$ by the proof above that the path from $a$ to $b$ is unique. Then by definition of $d_G$ on the cross edges, observe

$$d_G(a, b) = \cent(\Gamma_{\mathrm{can}}(a \rightsquigarrow b)) = \min \left\{ \lambda, \sum_{u \to v \in \Gamma_{\mathrm{can}}(a \rightsquigarrow b)} ||u - v||^p \right\}.$$

Consider the one cross edge $(\square_1, \square_2)$ of path $\Gamma_{\mathrm{can}}(a \rightsquigarrow b)$. We note $(\square_1, \square_2) \in W$ by construction of the cross edges, and by construction of $W$, one has

$$||c_{\square_1} - c_{\square_2}||^p \ge (1 - 2\varepsilon)^p \cdot ||a - b||^p \ge (1 - 2^{2p}\varepsilon) \cdot ||a - b||^p.$$

Then the result follows after $\sum_{u \to v \in \Gamma_{\mathrm{can}}(a \rightsquigarrow b)} ||u - v||^p \ge ||c_{\square_1} - c_{\square_2}||^p$ since the cross edge $(\square_1, \square_2)$ is an edge of the path $\Gamma_{\mathrm{can}}(a \rightsquigarrow b)$.

$\square$

We now describe the demands for the min-cost flow instance. For each $a \in \hat{A}$, we take its copy in $V_1$ and assign it a demand of $\eta(a) = \hat{\mu}(a)$. For each $b \in B$, we take its copy in $V_2$ and assign it a demand of $\eta(b) = -\nu(b)$. Then we finally assign a demand of $\eta(v) = 0$ for all other $v \in V_G$. This concludes the construction of the minimum-cost flow instance.

**Shortcutting flows.** Once a minimum cost flow is computed on $G$ using the algorithm of [16], we employ the procedure described in [2] to shortcut the minimum cost flow on $G$ and obtain a feasible transport plan.

### F.4 Algorithm for $\lambda$-Capped 1-Wasserstein distance

The algorithm we use to compute approximate discrete $\lambda$-capped optimal transport under Euclidean metrics follows a similar high-level approach as in Section F.3. We first construct a near-linear sized graph whose shortest path distance approximates Euclidean costs in expectation. We then capped the edge lengths so that shortest path distances between points in $\hat{A}$ and $B$ approximate the $\lambda$-capped Euclidean metric. We solve the minimum-cost flow problem on this graph by designing an efficient approximate primal-dual oracle and running the multiplicative weight method using this oracle as in [47]. Once an approximate solution to minimum-cost flow is computed via multiplicative weights, we shortcut paths to obtain a transport plan.

**Sparse undirected graph and minimum-cost flow instance.** We now construct an undirected sparse graph whose shortest path distance is well-approximated by a tree metric. Our construction is slightly modified from the standard (e.g. [3, 22]) to allow for approximate $\lambda$-capped distances at the expense of a logarithmic increase in the number of edges.

Suppose without loss of generality, $\min_{a,b \in \hat{A} \cup B} ||a - b|| = 1$ and $\hat{A}, B \subseteq [-\Delta', \Delta']^d$. If not, again one can rescale and translate $\hat{A}$ and $B$ so that this condition is satisfied. Let $\square^* = [-2\Delta', 2\Delta']^d + \vec{v}$ be a randomly shifted grid cell containing $\hat{A}, B$, where $\vec{v}$ is chosen uniformly at random from $[-\Delta, \Delta]^d$. We construct a quadtree $T$ recursively with root cell $\square^*$. For a cell $\square \in T$, if $\ell_\square = \frac{\varepsilon}{\log \Delta'}$, then denote $\square$ as a *leaf cell*. Otherwise, we add the child cells $\square' \in \mathsf{C}[\square]$ where $\square' \cap (\hat{A} \cup B) \neq \varnothing$ to the quadtree $T$ and recurse on each $\square'$. This completes the construction of the tree embedding.

Let $V$ be the set of cells in quadtree $T$ union with $\hat{A} \cup B$. For each point $p \in \hat{A} \cup B$, we add an undirected edge $(p, c_\square)$ to $E$, where $\square$ is the unique leaf cell containing $p$. We call such edges *leaf edges*. For every non-leaf cell $\square$ and every child cell $\square' \in \mathsf{C}[\square]$ of $\square$, we add an undirected edge $(c_\square, c_{\square'})$ between the centers of the parent-child pair to $E$. We call these edges *tree edges* or *vertical edges*.

Now construct an $\varepsilon$-well separated pair decomposition $W$ with quadtree cells from $T$ as in Section F.3. For each pair of cells $(\square_1, \square_2)$ in the WSPD $W$, let $\square$ denote their least common ancestor in $T$. Define $\square_1'$ as the minimum sidelength cell of $T$ containing $\square_1$ that satisfies $\ell_{\square_1'} \geq \frac{\varepsilon}{\log \Delta} \ell_\square$. Analogously define $\square_2'$ as the minimum sidelength cell of $T$ containing $\square_2$ that satisfies $\ell_{\square_2'} \geq \frac{\varepsilon}{\log \Delta} \ell_\square$. We add a single edge $(c_{\square_1'}, c_{\square_2'})$ between the centers of these ancestors for each $(\square_1, \square_2) \in W$, and call these edges *cross edges* or *horizontal edges*. Then define the sparse graph as $G = (V, E)$. The following properties on $G$ hold.

**Lemma F.5.** *The sparse graph* $G = (V, E)$ *satisfies* $|V| = O(n\varepsilon^{-2d} \log \Delta \log \varepsilon^{-1})$ *and* $|E| = O(n\varepsilon^{-3d} \log \Delta \log \varepsilon^{-1})$.

*Proof.* The vertex set $V$ of our graph $G$ consists of centers of all quadtree cells in the quadtree $T$. We note that the quadtree $T$ on $m$ points with spread (or diameter if minimum pairwise distance is assumed to be 1) $\Delta$ has size at most $O(m \log \Delta)$. Then the quadtree built on the support of $\hat{\mu}$ (contains $B$) has size $O(n\varepsilon^{2d} \log \Delta \log \varepsilon^{-1})$ since the size of the support of $\hat{\mu}$ is bounded above by $|\mathcal{P}| + |B| = O(n\varepsilon^{-2d} \log \varepsilon^{-1})$ by Lemma F.1.

The edge set consists of each tree edge plus one cross edge for each pair in the well-separated pair decomposition of the support of $\hat{\mu}$. Note that the size of the support of $\hat{\mu}$ is $O(n\varepsilon^{-2d} \log \varepsilon^{-1})$, and therefore its $\varepsilon$-WSPD has size $O(n\varepsilon^{-3d} \log \varepsilon^{-1})$. By e.g. [26], the $\varepsilon$-WSPD can certainly be constructed in $O(n\varepsilon^{-2d}(\varepsilon^{-d} + \log \Delta) \log \varepsilon^{-1})$ time. Finally, the number of tree edges is bounded by the number of vertices, $O(n\varepsilon^{-2d} \log \varepsilon^{-1})$. Combining bounds on the number of edges of each type gives the desired result. $\square$

We define the cost $\mathrm{d}_G$ of each edge as follows. For each leaf edge $(u, v)$, let $\mathrm{d}_G(u, v) = ||u - v||$. We will assume $\lambda \geq 1$; otherwise every transport plan has a cost of $\lambda$ since no pair $a, b \in \hat{A} \times B$ has cost less than $\lambda$. We then construct a height function, as in Section F.3. For each leaf cell $\square$ in the quadtree $T$, define its height as $h_\lambda(\square) = 0$. Then for each vertical edge $(c_\square, c_{\square'})$ where $\square' \in \mathsf{C}[\square]$, define the height of $\square$ as $h_\lambda(\square) = \min\left\{\frac{\lambda}{2}, h_\lambda(\square') + ||c_\square - c_{\square'}||\right\}$ and the length of edge $(c_\square, c_{\square'})$ as $\mathrm{d}_G(c_\square, c_{\square'}) = h_\lambda(\square) - h_\lambda(\square')$. Finally, for each horizontal edge $(c_{\square_1}, c_{\square_2})$, define its cost as $\mathrm{d}_G(c_{\square_1}, c_{\square_2}) = \min\{\lambda - h_\lambda(\square_1) - h_\lambda(\square_2), ||c_{\square_1} - c_{\square_2}||\}$. For any pair of vertices $u, v \in V$, we denote $\mathrm{d}_G(u, v)$ as the shortest path distance between $u$ and $v$. We additionally define $\mathrm{d}_T(u, v)$ as the shortest path distance in $G$ between $u$ and $v$ using only leaf edges and vertical edges. That is, $\mathrm{d}_T(u, v)$ is the shortest path distance on the subgraph $T$ of $G$ with respect to edge costs $\mathrm{d}_G$. Then the following properties of $\mathrm{d}_G$ and $\mathrm{d}_T$ hold.

**Lemma F.6.** *For every* $a, b \in \hat{A} \cup B$,

$$\min\left\{||a - b||, \lambda\right\} \leq \mathbb{E}\left[\mathrm{d}_G(a, b)\right] \leq (1 + 8d\varepsilon) \cdot \min\left\{||a - b||, \lambda\right\},$$

*where expectation is over random choice of* $\vec{v}$ *used to define* $\square^*$.

*Proof.* The structure of the proof will be as follows. We first prove the upper bound on $\mathbb{E}\left[\mathrm{d}_G(a,b)\right]$, then prove the lower bound on $\mathbb{E}\left[\mathrm{d}_G(a,b)\right]$.

1. **Upper bound.** Define the *canonical path* from $a$ to $b$ as the path $\Gamma_{\mathrm{can}}(a \rightsquigarrow b) := a \rightsquigarrow_T c_{\square_1} \to c_{\square_2} \rightsquigarrow_T b$, where $u \rightsquigarrow_T v$ denotes the unique shortest path from $u$ to $v$ using only tree edges and $(\square_1, \square_2)$ is the edge added to $E$ for the WSPD pair $(P, Q)$ where $a \in P$ and $b \in Q$. By construction of the set of cross edges, there exists a unique such edge $(\square_1, \square_2)$ for each $(P, Q) \in W$. Moreover, let $\phi(\Gamma) := \sum_{u \to v \in \Gamma} \mathrm{d}_G(u,v)$ denote the cost of the path $\Gamma$ in $G$.

We note $h_\lambda(\square) \leq \frac{\lambda}{2}$ for all $\square$ in the quadtree $T$ by construction of $h_\lambda$. Moreover, if $\phi(\Gamma_{\mathrm{can}}(a \rightsquigarrow b)) < \sum_{u \to v \in \Gamma_{\mathrm{can}}(a \rightsquigarrow b)} ||u - v||$, then $\phi(\Gamma_{\mathrm{can}}(a \rightsquigarrow b)) = \lambda + \frac{2\sqrt{d}\varepsilon}{\log \Delta} \leq (1 + \frac{2\sqrt{d}\varepsilon}{\log \Delta}) \cdot \lambda$ by construction of $\mathrm{d}_G$ on any path between two leaves with exactly one cross edge (leaf edges have cost at most $\frac{\sqrt{d}\varepsilon}{\log \Delta}$ and it is assumed $\lambda \geq 1$). Then

$$\mathbb{E}\left[\mathrm{d}_G(a,b)\right] \leq \mathbb{E}\left[\phi(\Gamma_{\mathrm{can}}(a \rightsquigarrow b))\right]$$

$$\leq \mathbb{E}\left[\min\left\{\left(1 + \frac{2\sqrt{d}\varepsilon}{\log \Delta}\right) \cdot \lambda, \sum_{u \to v \in \Gamma_{\mathrm{can}}(a \rightsquigarrow b)} ||u - v||\right\}\right]$$

$$\leq \min\left\{\left(1 + \frac{2\sqrt{d}\varepsilon}{\log \Delta}\right) \cdot \lambda, \mathbb{E}\left[\sum_{u \to v \in \Gamma_{\mathrm{can}}(a \rightsquigarrow b)} ||u - v||\right]\right\}$$

by Jensen's Inequality, where expectation is over the random shift $\vec{v}$ in the root of the quadtree $T$. To conclude the upper bound on $\mathbb{E}\left[\mathrm{d}_G(a,b)\right]$, we prove $\mathbb{E}\left[\sum_{u \to v \in \Gamma_{\mathrm{can}}(a \rightsquigarrow b)} ||u - v||\right] \leq (1 + \varepsilon)||a - b||$.

Consider the one cross edge $(\square_1, \square_2)$ of path $\Gamma_{\mathrm{can}}(a \rightsquigarrow b)$. If $(\square_1, \square_2) \in W$, then by construction of $W$ we note $||c_{\square_1} - c_{\square_2}|| \leq (1 + 2\varepsilon) \cdot ||a - b||$. Furthermore,

$$\sum_{u \to v \in a \rightsquigarrow_T c_{\square_1}} ||u - v|| \leq 2\sqrt{d}\ell_{\square_1} \leq 2\sqrt{d}\varepsilon \cdot ||a - b||,$$

and analogously $\sum_{w \to z \in c_{\square_2} \rightsquigarrow_T b} ||w - z|| \leq 2\sqrt{d}\varepsilon \cdot ||a - b||$. One can then conclude

$$\sum_{u \to v \in \Gamma_{\mathrm{can}}(a \rightsquigarrow b)} ||u - v|| \leq \left(1 + \left(2 + 4\sqrt{d}\right)\varepsilon\right) \cdot ||a - b||.$$

Now suppose $(\square_1, \square_2) \notin W$. Then by construction of the cross edges, $\ell_{\square_1} = \ell_{\square_2} \leq \frac{\varepsilon}{\log \Delta} \cdot \ell_{\square_{ab}}$, where $\square_{ab}$ is the smallest cell in the quadtree $T$ containing both $a$ and $b$. In this case, it then follows that

$$\mathbb{E}\left[\sum_{u \to v \in \Gamma_{\mathrm{can}}(a \rightsquigarrow b)} ||u - v||\right] = \mathbb{E}\left[\sum_{u \to v \in a \rightsquigarrow_T c_{\square_1}} ||u - v||\right] + \mathbb{E}\left[||c_{\square_1} - c_{\square_2}||\right]$$

$$+ \mathbb{E}\left[\sum_{w \to z \in c_{\square_2} \rightsquigarrow_T b} ||w - z||\right]$$

$$\leq 2\sqrt{d}\frac{\varepsilon}{\log \Delta} \cdot \mathbb{E}\left[\ell_{\square_{ab}}\right] + \mathbb{E}\left[||c_{\square_1} - c_{\square_2}||\right] + 2\sqrt{d}\frac{\varepsilon}{\log \Delta} \cdot \mathbb{E}\left[\ell_{\square_{ab}}\right]$$

$$= \frac{4\sqrt{d}\varepsilon}{\log \Delta} \cdot \mathbb{E}\left[\ell_{\square_{ab}}\right] + \mathbb{E}\left[||c_{\square_1} - c_{\square_2}||\right].$$

We additionally note that by triangle inequality,

$$||c_{\square_1} - c_{\square_2}|| \leq \left(\sum_{u \to v \in c_{\square_1} \rightsquigarrow_T a} ||u - v||\right) + ||a - b|| + \left(\sum_{w \to z \in b \rightsquigarrow_T c_{\square_2}} ||w - z||\right).$$

Therefore, $\mathbb{E}\left[||c_{\square_1} - c_{\square_2}||\right] \leq ||a - b|| + \frac{4\sqrt{d}\varepsilon}{\log\Delta} \cdot \mathbb{E}\left[\ell_{\square_{ab}}\right]$. Substituting this inequality gives the bound

$$\mathbb{E}\left[\sum_{u\to v\in\Gamma_{\text{can}}(a\rightsquigarrow b)} ||u - v||\right] \leq ||a - b|| + \frac{8\sqrt{d}\varepsilon}{\log\Delta} \cdot \mathbb{E}\left[\ell_{\square_{ab}}\right]. \tag{21}$$

Now to bound $\mathbb{E}\left[\ell_{\square_{ab}}\right]$, we observe that if $\square_{ab}$ is the smallest cell containing both $a$ and $b$, then the child cell $\square_a$ of $\square_{ab}$ containing $a$ must not contain $b$. Therefore, there must be a face on the boundary of $\square_a$ which intersects the line segment from $a$ to $b$. Furthermore, since $\square_{ab}$ is convex and contains both $a$ and $b$, we observe that the entire line segment from $a$ to $b$ must be contained in $\square_{ab}$. We use these two properties to claim $\Pr\left[\ell_{\square_{ab}} = 2^j\right] \leq \frac{\sqrt{d}||a-b||}{2^{j+1}}$. Then summing over all possible lengths of the cell $\square_{ab}$ in the quadtree $T$ gives the bound on the expectation

$$\mathbb{E}\left[\ell_{\square_{ab}}\right] = \sum_{j=\log\frac{\varepsilon}{\log\Delta}}^{\log\Delta} \Pr\left[\ell_{\square_{ab}} = 2^j\right] \cdot 2^j$$

$$\leq \sum_{j=\log\frac{\varepsilon}{\log\Delta}}^{\log\Delta} \frac{\sqrt{d}||a-b||}{2^{j+1}} \cdot 2^j$$

$$= \sum_{j=\log\frac{\varepsilon}{\log\Delta}}^{\log\Delta} \frac{\sqrt{d}}{2} \cdot ||a - b||$$

$$\leq \left(\sqrt{d}\log\Delta\right) \cdot ||a - b||,$$

where the last inequality assumes $\log\frac{\varepsilon}{\log\Delta} \leq \log\Delta$. Substituting this bound above into Equation 21 gives

$$\mathbb{E}\left[\sum_{u\to v\in\Gamma_{\text{can}}(a\rightsquigarrow b)} ||u - v||\right] \leq (1 + 8d\varepsilon) \cdot ||a - b||.$$

This concludes the proof of the upper bound.

2. **Lower bound.** Given a path $\Gamma$, define $\text{Vert}(\Gamma)$ and $\text{Hor}(\Gamma)$ as the set of vertical and cross edges in $\Gamma$, respectively. Again define $\cancel{c}(\Gamma) := \sum_{u\to v\in\Gamma} d_G(u, v)$ to be the cost of the path $\Gamma$ in $G$. For each vertex $v$ of the graph $G$, let $\square_v$ denote the cell of the quadtree corresponding to vertex $v$ and let $\text{lev}(v) := \log\ell_{\square_v}$ denote the level of the vertex $v$ in the quadtree.

Let $\Gamma$ be an arbitrary path from $a$ to $b$ in $G$. If $\cancel{c}(\Gamma) \geq \sum_{u\to v\in\Gamma} ||u - v||$, then since $\Gamma$ is a path from $a$ to $b$ we can use the triangle inequality to conclude

$$\cancel{c}(\Gamma) \geq \sum_{u\to v\in\Gamma} ||u - v||$$

$$\geq ||a - b||$$

$$\geq \min\{||a - b||, \lambda\}.$$

Now suppose $\cancel{c}(\Gamma) < \sum_{u\to v\in\Gamma} ||u - v||$. Then there must be at least one edge $w\to z$ in the path $\Gamma$ with $d_G(w, z) < ||w - z||$.

Suppose $(w, z)$ is a vertical edge. Then since $d_G(w, z) < ||w - z||$ and $(w, z)$ is a vertical edge, $h_\lambda(\square_w) = \frac{\lambda}{2}$ or $h_\lambda(\square_z) = \frac{\lambda}{2}$. Suppose, without loss of generality, $h_\lambda(\square_w) = \frac{\lambda}{2}$. Let $\Gamma_1$ be the subpath of $\Gamma$ starting from $a$ and ending at $w$. Additionally let $\Gamma_2$ be the subpath of $\Gamma$ starting from $w$ and ending at $b$. Then the composition of $\Gamma_1$ and $\Gamma_2$ is equal to $\Gamma$. Therefore,

$$\cancel{c}(\Gamma) = \cancel{c}(\Gamma_1) + \cancel{c}(\Gamma_2).$$

Additionally, note each subpath can be decomposed into vertical and horizontal edges:

$$\cancel{c}(\Gamma_i) = \sum_{u\to v\in\text{Vert}(\Gamma_i)} d_G(u, v) + \sum_{u\to v\in\text{Hor}(\Gamma_i)} d_G(u, v) \quad \text{for } i \in \{1, 2\}.$$

By construction of $W$ and the set of cross edges from $W$, $\mathrm{lev}(u) = \mathrm{lev}(v)$ for every cross edge $(u, v) \in E$. Therefore, to reach $w$ from $a$ in $\Gamma_1$, there must be at least one vertical edge $u_i \to v_i$ in $\mathrm{Vert}(\Gamma_1)$ for each $i < \mathrm{lev}(\square_w)$ such that $\mathrm{lev}(\square_{v_i}) > \mathrm{lev}(\square_{u_i}) = i$. Likewise, to reach $b$ from $w$ in $\Gamma_1$, there must be at least one vertical edge $u_i' \to v_i'$ in $\mathrm{Vert}(\Gamma_2)$ for each $i < \mathrm{lev}(\square_w)$ such that $\mathrm{lev}(\square_{u_i'}) < \mathrm{lev}(\square_{v_i'}) = i$. Then by the fixed depth and definition of child cells of the quadtree $T$, $h_\lambda(\square) = h_\lambda(\square')$ for all pairs of cells where $\mathrm{lev}(\square) = \mathrm{lev}(\square')$. Using the definition of $\mathrm{d}_G(u,v)$ for vertical edges $(u,v)$, we deduce that

$$\sum_{u \to v \in \mathrm{Vert}(\Gamma_1)} \mathrm{d}_G(u,v) \geq \sum_{u_i \to v_i} h_\lambda(\square_{v_i}) - h_\lambda(\square_{u_i}) \geq h_\lambda(\square_w) - 0 = \frac{\lambda}{2}.$$

Note that $h_\lambda(\square_a) = 0$ by definition, where $\square_a$ is the leaf cell in $T$ containing $a$. We similarly observe $\sum_{u \to v \in \mathrm{Vert}(\Gamma_2)} \mathrm{d}_G(u,v) \geq \frac{\lambda}{2}$. Hence, $\cancel{c}(\Gamma) \geq \lambda$.

Now suppose $(w, z)$ is a cross edge. Then by definition of $\mathrm{d}_G(w,z) = \min\{||w - z||, \lambda - h_\lambda(\square_w) - h_\lambda(\square_z)\}$, it must be the case that $\mathrm{d}_G(w,z) = \lambda - h_\lambda(\square_w) - h_\lambda(\square_z)$. Let $\Gamma_1$ denote the subpath of $\Gamma$ from $a$ to $w$ and let $\Gamma_2$ denote the subpath of $\Gamma$ from $z$ to $b$. Then the plan $\Gamma$ is equal to the composition $\Gamma_1 \circ (w \to z) \circ \Gamma_2$. Then in the same manner as the previous case, we argue $\sum_{u \to v \in \mathrm{Vert}(\Gamma_1)} \mathrm{d}_G(u,v) \geq h_\lambda(\square_w)$ and $\sum_{u \to v \in \mathrm{Vert}(\Gamma_2)} \mathrm{d}_G(u,v) \geq h_\lambda(\square_z)$. It immediately follows that

$$\begin{aligned}
\cancel{c}(\Gamma) &\geq \cancel{c}(\Gamma_1) + \mathrm{d}_G(w,z) + \cancel{c}(\Gamma_2) \\
&\geq h_\lambda(\square_w) + [\lambda - h_\lambda(\square_w) - h_\lambda(\square_z)] + h_\lambda(\square_z) \\
&= \lambda.
\end{aligned}$$

To conclude the lower bound, we observe that the path $\Gamma$ from $a$ to $b$ was chosen arbitrarily among all possible paths in $G$.

$\square$

**Lemma F.7.** *For every $u, v \in V$, $\mathrm{d}_T(u,v) \leq O\left(\frac{\log \Delta}{\varepsilon}\right) \cdot \mathrm{d}_G(u,v)$.*

*Proof.* We use the definitions of $\mathrm{Vert}(\Gamma), \mathrm{Hor}(\Gamma), \cancel{c}(\Gamma)$ and $\mathrm{lev}$ as in the proof of Lemma F.6. With some slight abuse of notation, we will use $\mathrm{lev}(\square_v)$ and $\mathrm{lev}(v)$ interchangeably, as well as $h_\lambda(\square_v)$ and $h_\lambda(v)$ for each vertex $v$ of the quadtree $T$ and corresponding cell $\square_v$. We first claim for every edge $(u, v) \in E$, the shortest path from $u$ to $v$ in $G$ is through the edge $(u, v)$ with cost $\mathrm{d}_G(u,v)$.

Suppose $(u, v)$ is a vertical edge. Then $\mathrm{lev}(u) \neq \mathrm{lev}(v)$. Suppose $\Gamma$ is the shortest path from $u$ to $v$ in $G$. Then there exists some edge $w \to z$ in the path $\Gamma$ such that $\mathrm{lev}(w) = \mathrm{lev}(u)$ and $\mathrm{lev}(z) = \mathrm{lev}(v)$. We note, by the definition of the cost $\mathrm{d}_G$ on vertical edges and equal distance between all parent and child cell centers at a fixed level, that $\cancel{c}(\Gamma) \geq \mathrm{d}_G(w,z) = \mathrm{d}_G(u,v)$. Therefore, the edge $u \to v$ is also a shortest path from $u$ to $v$.

Suppose $(u, v)$ is a horizontal edge. Then $\mathrm{d}_G(u,v) = \min\{\lambda - h_\lambda(u) - h_\lambda(v), ||u-v||\}$ by definition. If there exists a path $\Gamma$ from $u$ to $v$ in $G$ such that $\cancel{c}(\Gamma) = \sum_{p \to q \in \Gamma} \mathrm{d}_G(p,q) < ||u - v||$, then by triangle inequality there must exist an edge $w \to z$ on this path $\Gamma$ such that $\mathrm{d}_G(w,z) < ||w - z||$. Suppose this edge $(w, z)$ is a vertical edge. Then without loss of generality assume $w = \mathrm{pa}(z)$. We note that by definition of $\mathrm{d}_G$, if $\mathrm{d}_G(w,z) < ||w - z||$ then $h_\lambda(w) = \frac{\lambda}{2}$. Consider the subpaths $\Gamma_1$ and $\Gamma_2$ of $\Gamma$ from $u$ to $w$ and from $w$ to $v$, respectively. Then as in the proof of Lemma F.6, note

$$\sum_{p \to q \in \mathrm{Vert}(\Gamma_1)} \mathrm{d}_G(p,q) + \sum_{p \to q \in \mathrm{Vert}(\Gamma_2)} \mathrm{d}_G(p,q) \geq \left(\frac{\lambda}{2} - h_\lambda(u)\right) + \left(\frac{\lambda}{2} - h_\lambda(v)\right) = \mathrm{d}_G(u,v).$$

If instead $(w, z)$ is a cross edge, then $\mathrm{d}_G(w,z) = \lambda - h_\lambda(w) - h_\lambda(z)$. Since $(w, z)$ and $(u, v)$ are both cross edges, $\mathrm{lev}(w) = \mathrm{lev}(z)$ and $\mathrm{lev}(u) = \mathrm{lev}(v)$. If $\mathrm{lev}(w) \leq \mathrm{lev}(u)$, then

$$d_G(w,z) = \lambda - h_\lambda(w) - h_\lambda(z) \geq \lambda - h_\lambda(u) - h_\lambda(v) \geq \mathrm{d}_G(u,v).$$

Else suppose $\mathrm{lev}(w) > \mathrm{lev}(u)$. Let $\Gamma_1$ be the subpath of $\Gamma$ from $u$ to $w$ and $\Gamma_2$ be the subpath of $\Gamma$ from $z$ to $v$. Then for every $j \in [\mathrm{lev}(u), \mathrm{lev}(w))$, there exist vertical edges $u_j \to v_j \in \Gamma_1$ and $v'_j \to u'_j \in \Gamma_2$ where $\mathrm{lev}(u_j) = \mathrm{lev}(u'_j) = j < \mathrm{lev}(v_j) = \mathrm{lev}(v'_j)$. Then

$$
\begin{aligned}
\mathcal{c}(\Gamma) &= \mathcal{c}(\Gamma_1) + \mathrm{d}_G(w,z) + \mathcal{c}(\Gamma_2) \\
&\geq \sum_{p \to q \in \mathrm{Vert}(\Gamma_1)} \mathrm{d}_G(p,q) + [\lambda - h_\lambda(w) - h_\lambda(z)] + \sum_{p \to q \in \mathrm{Vert}(\Gamma_2)} \mathrm{d}_G(p,q) \\
&\geq \sum_{j=\mathrm{lev}(u)}^{\mathrm{lev}(w)} \mathrm{d}_G(u_j, v_j) + [\lambda - h_\lambda(w) - h_\lambda(z)] + \sum_{j=\mathrm{lev}(u)}^{\mathrm{lev}(w)} \mathrm{d}_G(u'_j, v'_j) \\
&= \sum_{j=\mathrm{lev}(u)}^{\mathrm{lev}(w)} (h_\lambda(v_j) - h_\lambda(u_j)) + [\lambda - h_\lambda(w) - h_\lambda(z)] + \sum_{j=\mathrm{lev}(u)}^{\mathrm{lev}(w)} \left( h_\lambda(v'_j) - h_\lambda(u'_j) \right) \\
&= (h_\lambda(w) - h_\lambda(u)) + [\lambda - h_\lambda(w) - h_\lambda(z)] + (h_\lambda(z) - h_\lambda(v)) \\
&= [\lambda - h_\lambda(u) - h_\lambda(v)] \geq \mathrm{d}_G(u,v).
\end{aligned}
$$

This concludes the claim that for every edge $(u,v) \in E$, the shortest path from $u$ to $v$ in $G$ is through the edge $(u,v)$ with cost $\mathrm{d}_G(u,v)$.

After the above claim, it suffices to prove $\mathrm{d}_G(u,v) \leq O(\frac{\log \Delta}{\varepsilon}) \cdot \mathrm{d}_G(u,v)$ for every cross edge $(u,v)$. Let $(u,v)$ be an arbitrary cross edge. Suppose $\mathrm{d}_G(u,v) = \lambda - h_\lambda(u) - h_\lambda(v)$. Then observe

$$
\begin{aligned}
\mathrm{d}_T(u,v) &= \sum_{w \to z \in u \leadsto_T \mathrm{lca}(u,v)} (h_\lambda(z) - h_\lambda(w)) + \sum_{w \to z \in u \leadsto_T \mathrm{lca}(u,v)} (h_\lambda(w) - h_\lambda(z)) \\
&= (h_\lambda(\mathrm{lca}(u,v)) - h_\lambda(u)) + (h_\lambda(\mathrm{lca}(u,v)) - h_\lambda(v)) \\
&\leq 2 \cdot \frac{\lambda}{2} - h_\lambda(u) - h_\lambda(v) = \mathrm{d}_G(u,v).
\end{aligned}
$$

If instead $\mathrm{d}_G(u,v) = ||u - v||$, then observe

$$
\begin{aligned}
\mathrm{d}_T(u,v) &\leq \sum_{w \to z \in u \leadsto_T \mathrm{lca}(u,v)} ||w - z|| + \sum_{w \to z \in u \leadsto_T \mathrm{lca}(u,v)} ||w - z|| \\
&\leq \frac{2\sqrt{d} \log \Delta}{\varepsilon} \cdot ||u - v|| = \frac{2\sqrt{d} \log \Delta}{\varepsilon} \cdot \mathrm{d}_G(u,v)
\end{aligned}
$$

by construction that for each cross edge $u,v$, with corresponding cells $\square_u$ and $\square_v$ in the quadtree, $\ell_{\square_u} = \ell_{\square_v} \geq \frac{\varepsilon}{\log \Delta} \cdot \ell_{\square_{\mathrm{lca}(u,v)}}$. This concludes the desired statement of the Lemma. $\qquad \square$

We then define the demands $\eta$ on $V$ by $\eta(a) = \hat{\mu}(a)$ for all $a \in \hat{A}$, $\eta(b) = \nu(b)$ for all $b \in B$, and $\eta(v) = 0$ for all $v \in V \setminus (\hat{A} \cup B)$. This completes the construction of the minimum-cost flow instance we solve.

**Approximate minimum-cost flow oracle.** We describe a simple $O(\frac{\log \Delta}{\varepsilon})$-approximate primal-dual oracle for minimum-cost flow on $G$. The oracle, given an arbitrary demand function $\eta$, will compute a primal-dual pair $(f, y)$ satisfying the following properties:

1. $f$ is a feasible primal solution to the min-cost flow problem, i.e. $f \geq 0$ and $\sum_{v:(u,v) \in E} f(u \to v) - f(v \to u) = \eta(u)$ for all $u \in V$,

2. $y$ is $O\left(\frac{\log \Delta}{\varepsilon}\right)$-approximately feasible, i.e. $y(u) - y(v) \leq O\left(\frac{\log \Delta}{\varepsilon}\right) \cdot \mathrm{d}_G(u,v)$ for all $u, v \in E$, and

3. the dual objective value dominates the primal objective, i.e. $\sum_{v \in V} y(v)\eta(v) \geq \sum_{(u,v) \in E} f(u \to v)\mathrm{d}_G(u,v)$.

Then it is known from existing works, e.g. [7, 44, 47], that given an oracle satisfying the properties above, one can apply the multiplicative weight update method to obtain a $(1 + \varepsilon)$-approximate minimum-cost flow $f$ routing the demand $\eta$.

**Lemma F.8.** *[[47] Lemma 11] Given an oracle which returns a primal-dual pair satisfying conditions (1-3) above upon receiving a demand function $\eta\colon V \to \mathbb{R}$, there exists an algorithm which computes a $(1 + \varepsilon)$-approximate minimum-cost flow after $O(\varepsilon^{-4} \log n \log^2 \Delta)$ calls to the oracle plus $O(|E|)$ operations per oracle call.*

The oracle builds a primal-dual solution on the quadtree $T$ in a greedy manner. For a vertex $v \in V$, let $T(v)$ denote the subtree of $T$ rooted at $v$ and define $\eta(T(v)) = \sum_{u \in T(v)} \eta(u)$. Then for every vertical edge $(u, v) \in E$ where $v = \mathrm{pa}(u)$, set $f(u \to v) = \eta(T(u))$ if $\eta(T(u)) \geq 0$ and $f(v \to u) = -\eta(T(u))$ if $\eta(T(u)) < 0$.

To construct a dual solution, let $r$ denote the root vertex of the tree $T$ and set $y(r) = 0$. Then in a top-down manner, for every vertical edge $(u, v) \in E$ where $v = \mathrm{pa}(u)$, we set $y(u) = y(v) + \mathrm{d}_G(u, v) \cdot \mathrm{sgn}(\eta(T(u)))$ where $\mathrm{sgn}(x)$ denotes the sign function.

**Lemma F.9.** *Given an arbitrary demand function $\eta$, the oracle described above computes a primal-dual pair $(f, y)$ in $O(n\varepsilon^{-3d} \log \Delta \log \varepsilon^{-1})$ time satisfying*

1. *$\sum_{v:(u,v)\in E} f(u \to v) - f(v \to u) = \eta(u)$ for all $u \in V$,*

2. *$y(u) - y(v) \leq O\left(\frac{\log \Delta}{\varepsilon}\right) \cdot \mathrm{d}_G(u, v)$, and*

3. *$\sum_{v \in V} y(v)\eta(v) \geq \sum_{(u,v)\in E} (f(u \to v) + f(v \to u)) \mathrm{d}_G(u, v)$.*

*Proof.* 1. By definition of $f$, we have that for every leaf vertex $u$ of the tree $T$, $f(u \to \mathrm{pa}(u)) - f(\mathrm{pa}(u) \to u) = \eta(T(u)) = \eta(u)$.

Now suppose $v$ is a non-leaf vertex of $T$ (vertices of $T$ and $G$ are the same), and $\sum_{v:(u,v)\in E} f(u \to v) - f(v \to u) = \eta(u)$ for all $u \in T(w)$ and all $w \in \mathsf{C}[v]$. Then, note that $f(w \to v) - f(v \to w) = \eta(T(w))$ for all $w \in \mathsf{C}[v]$ by construction of $f$. Therefore, to guarantee the desired property, we must have $f(v \to \mathrm{pa}(v)) - f(\mathrm{pa}(v) \to v) = \eta(v) + \sum_{w\in\mathsf{C}[v]} \eta(T(w))$. Observe that $\eta(v) + \sum_{w\in\mathsf{C}[v]} \eta(T(w)) = \eta(T(v))$ by definition of $\eta(T(v))$. Then by definition of $f$, it is indeed true that $f(v \to \mathrm{pa}(v)) - f(\mathrm{pa}(v) \to v) = \eta(T(v))$.

2. By construction of $y$, we observe that $|y(u) - y(v)| \leq \mathrm{d}_G(u, v) = \mathrm{d}_T(u, v)$ for every vertical edge $(u, v) \in E$ in the graph $G$. Then for every cross edge $(w, z) \in E$, it follows that

$$y(w) - y(z) = \sum_{u\to v \in w \rightsquigarrow_T z} y(u) - y(v) \leq \sum_{u\to v \in w \rightsquigarrow_T z} \mathrm{d}_T(u, v) = \mathrm{d}_T(w, z),$$

where $w \rightsquigarrow_T z$ is the path in the tree $T$ from $w$ to $z$. The result follows from Lemma F.7, where it is shown that $\mathrm{d}_T(w, z) \leq O\left(\frac{\log \Delta}{\varepsilon}\right) \cdot \mathrm{d}_G(w, z)$.

3. We use complementary slackness and strong duality to prove the solution is an optimal primal-dual pair on the tree $T$. By definition of $f$ and $y$, we note that for every vertical edge $(u, v)$ where $u$ is the child of $v$, $y(u) - y(v) = \mathrm{d}_G(u, v)$ if $f(u \to v) > 0$ (i.e. if $\eta(T(u)) > 0$) and $y(v) - y(u) = \mathrm{d}_G(u, v)$ if $f(v \to u) > 0$ (i.e. if $\eta(T(u)) < 0$). Additionally if $f(u \to v) > 0$ then $f(v \to u) = 0$. Moreover, $f(u \to v)$ is only strictly positive on vertical edges by construction of $f$. Therefore,

$$
\begin{aligned}
\sum_{(u,v)\in E} (f(u \to v) + f(v \to u)) \mathrm{d}_G(u, v) &= \sum_{(u,v)\in E: f(u\to v)>0} f(u \to v)\mathrm{d}_G(u, v) \\
&= \sum_{(u,v)\in E: f(u\to v)>0} f(u \to v) \cdot (y(u) - y(v)) \\
&= \sum_{u\in V} \sum_{v:(u,v)\in E} (f(u \to v) - f(v \to u)) \cdot y(u) \\
&= \sum_{u\in V} y(u)\eta(u).
\end{aligned}
$$

The third equality above follows from rearranging terms and the last equality above follows from the first condition of the Lemma.

$\square$

**Shortcutting flows.** Once a minimum cost flow is computed on the geometric graph $G$, shortcutting the resulting flow on $G$ to obtain an approximate transport plan in roughly linear time in $|E|$ is standard. See e.g. [22, 28].

### F.5 Semi-discrete plan from discrete plan.

Now suppose one has computed an $(\varepsilon_r, 0)$-approximate $\lambda$-capped discrete transport plan $\hat{\tau}_\lambda$ between $\hat{\mu}$ and $\nu$. We show how to transform $\hat{\tau}_\lambda$ into an $(\varepsilon_r, 0)$-approximate $\lambda$-capped semi-discrete transport plan $\tau_\lambda$ between $\mu$ and $\nu$. For all $\square \in \mathcal{P} \setminus \bigcup_{b \in B} \mathcal{N}_\varepsilon(b)$, define $\tau_\lambda(a, b) = \mu(a) \cdot \frac{\hat{\tau}_\lambda(c_\square, b)}{\hat{\mu}(c_\square)}$ over all $a \in \square$ and $b \in B$. Likewise, let $\tau_\lambda(a, b) = \mu(a) \cdot \frac{\hat{\tau}_\lambda(r, b)}{\hat{\mu}(r)}$ for all $a \in \mathbb{R}^d \setminus P$ and $b \in B$. What remains is a procedure to route the mass within each cell $\square$ of the neighborhoods $\mathcal{N}_\varepsilon(b)$.

We repeat the following local routing scheme for each $b \in B$. If $\hat{\tau}_\lambda(b, b) < \frac{1}{2}\hat{\mu}(b)$, then let $\tau_\lambda(a, b') = \mu(a) \cdot \frac{\hat{\tau}_\lambda(b, b')}{\hat{\mu}(b)}$ for all $a \in N_\varepsilon(b)$ and $b' \in B$. Otherwise, define

$$\mathcal{P}_i(b) = \{\square \in \mathbb{G}_{\varepsilon\delta_i} : ||c_\square - b|| < \delta_i\}$$

and $P_i(b) = \bigcup_{\square \in \mathcal{P}_i(b)} \square$ for all $i \in \mathbb{Z}$. Let $i^*$ be such that $\int_{P_{i^*}(b)} \mu(a)da \leq \hat{\tau}_\lambda(b, b) < \int_{P_{i^*+1}(b)} \mu(a)da$. Since $\hat{\tau}_\lambda(b, b) \leq \hat{\mu}(b)$, we note $P_{i^*}(b) \subseteq N_\varepsilon(b)$. Let $\tau_\lambda(a, b) = \mu(a)$ for all $a \in P_{i^*}(b)$. Then for each $\square \in \mathcal{P}_{i^*+1}(b)$ such that there exists a $\square' \in \mathcal{P}_{i^*}(b)$ where $\square' \subset \square$, we replace $\square$ with its $2^d$ child cells in $\mathcal{P}_{i^*+1}(b)$. Initialize an undispersed mass counter $\hat{\tau}_\lambda^{\mathrm{res}}(b, b) = \hat{\tau}_\lambda(b, b) - \int_{P_{i^*}(b)} \mu(a)da$. For each $\square \in \mathcal{P}_{i^*+1}(b) \setminus \mathcal{P}_{i^*}(b)$ in increasing order of $||c_\square - b||$, set $\tau_\lambda(a, b) = \mu(a) \cdot \frac{\min\{\hat{\tau}_\lambda^{\mathrm{res}}(b,b), \int_\square \mu(a)da\}}{\int_\square \mu(a)da}$ on all $a \in \square$ and subtract $\min\{\hat{\tau}_\lambda^{\mathrm{res}}(b, b), \int_\square \mu(a)da\}$ from $\hat{\tau}_\lambda^{\mathrm{res}}(b, b)$. After assigning $\tau_\lambda(a, b)$ for all $b \in B$ and $a \in N_\varepsilon(b)$, we then set $\tau_\lambda(a, b') = (\mu(a) - \tau_\lambda(a, b)) \cdot \frac{\hat{\tau}_\lambda(b, b')}{\hat{\mu}(b) - \hat{\tau}_\lambda(b, b)}$ for all remaining $a \in N_\varepsilon(b)$ and $b' \neq b$.

**Lemma F.10.** *Suppose $\mu$ is a continuous distribution with compact support $A \subset \mathbb{R}^d$ and $\nu$ is a discrete distribution with support $B \subset \mathbb{R}^d$ for some constant $d \geq 1$. Let $\hat{\mu}$ be the discrete distribution formed from $\mu$ as in Section F.1, and let $\mathrm{d}(a, b) = ||a - b||^p$ for fixed constant $p \geq 1$. If $\hat{\tau}_\lambda$ is an $(\varepsilon, 0)$-approximate discrete $\lambda$-capped transport plan between $\hat{\mu}$ and $\nu$, then $\tau_\lambda$ is an $(O(\varepsilon), 0)$-approximate semi-discrete $\lambda$-capped transport plan between $\mu$ and $\nu$.*

*Proof.* Let $\tau^*$ be an arbitrary transport plan between $\mu$ and $\nu$, where $\mathrm{M}(\tau^*) = 1$, and let $\hat{\tau}^*$ be the transport plan from $\hat{\mu}$ to $\nu$ defined by $\hat{\tau}^*(c_\square, b) = \int_\square \tau^*(a, b)da$ for all $b \in B$ and $\square \in \mathcal{P} \setminus \mathcal{N}_\varepsilon(b)$ and $\hat{\tau}^*(b, b) = \int_{N_\varepsilon(b)} \tau^*(a, b)da$ for all $b \in B$. By Lemma F.1, we have that all cells $\square \in \mathcal{P}$ which are not contained in the neighborhood $\mathcal{N}_\varepsilon(b)$ of a point $b$ are $2\varepsilon$-well separated from $b$. We use this property to find

$$\sum_{b \in B} \int_A \tau_\lambda(a, b)\mathrm{d}(a, b)da \leq (1 + 2\sqrt{d}\varepsilon)^p \sum_{b \in B} \sum_{a \in \hat{A}} \hat{\tau}_\lambda(a, b)\mathrm{d}(a, b) + \sum_{b \in B} \int_{N_\varepsilon(b)} \tau_\lambda(a, b)\mathrm{d}(a, b)da.$$

Now since $\hat{\tau}_\lambda$ is a $(\varepsilon_r, 0)$-approximate discrete $\lambda$-capped OT plan between $\hat{\mu}$ and $\nu$, we find

$$\sum_{b \in B} \sum_{a \in \hat{A}} \hat{\tau}_\lambda(a, b)\mathrm{d}(a, b) \leq (1 + \varepsilon) \sum_{b \in B} \sum_{a \in \hat{A}} \hat{\tau}_\lambda^*(a, b)\mathrm{d}(a, b).$$

Finally, we again use the property that all cells $\square \in \mathcal{P}$ which are not contained in the neighborhood $\mathcal{N}_\varepsilon(b)$ of a point $b$ are $\varepsilon$-well separated from $b$ to compare $\hat{\tau}^*$ with $\tau^*$. This property along with the fact that $\mathrm{d}(b, b) = 0$ implies

$$\sum_{b \in B} \sum_{a \in \hat{A}} \hat{\tau}^*(a, b)\mathrm{d}(a, b) \leq (1 + 2\sqrt{d}\varepsilon)^p \sum_{b \in B} \int_A \tau^*(a, b)\mathrm{d}(a, b)da$$

$$- (1 + 2\sqrt{d}\varepsilon)^p \sum_{b \in B} \int_{N_\varepsilon(b)} \tau^*(a, b)\mathrm{d}(a, b)da.$$

We combine the three above inequalities to deduce

$$
\begin{aligned}
\sum_{b \in B} \int_A \tau_\lambda(a,b)da \le{}& \left(1+2\sqrt{d}\varepsilon\right)^{2p}(1+\varepsilon)\sum_{b \in B}\int_A \tau^*(a,b)\mathrm{d}(a,b)da \\
&+\sum_{b \in B}\int_{N_\varepsilon(b)}\tau_\lambda(a,b)\mathrm{d}(a,b)da \\
&-\left(1+2\sqrt{d}\varepsilon\right)^{2p}\sum_{b \in B}\int_{N_\varepsilon(b)}\tau^*(a,b)\mathrm{d}(a,b)da \\
\le{}& \left(1+\left(2^{4p+1}d^p+1\right)\varepsilon\right)\cdot\sum_{b \in B}\int_A \tau^*(a,b)\mathrm{d}(a,b)da \\
&+\sum_{b \in B}\int_{N_\varepsilon(b)}\tau_\lambda(a,b)\mathrm{d}(a,b)da \\
&-\left(1+2^{4p}d^p\varepsilon\right)\cdot\sum_{b \in B}\int_{N_\varepsilon(b)}\tau^*(a,b)\mathrm{d}(a,b)da.
\end{aligned}
$$

It now suffices to show that

$$
\sum_{b \in B}\int_{N_\varepsilon(b)}\tau_\lambda(a,b)\mathrm{d}(a,b)da \le \left(1+2^{2p}d^p\varepsilon\right)\cdot\sum_{b \in B}\int_{N_\varepsilon(b)}\tau^*(a,b)\mathrm{d}(a,b)da.
$$

If this is true, then we substitute into the above inequality and conclude with the desired result.

By construction of $\tau_\lambda$ within $N_\varepsilon(b)$ for each $b \in B$, we observe that if $\tau_\lambda(a,b)>0$, then the cell $\square \in \mathcal{P}_{i*+1}(b)$ where $a \in \square$ satisfies $||c_\square-b|| \le ||c_{\square'}-b||$ for all $\square' \in \mathcal{P}_{i*+1}(b)$ where there exists some $a' \in \square'$ such that $\tau_\lambda(a',b)<\mu(a')$. Therefore, for any $a,a' \in N_\varepsilon(b)$ where $\tau_\lambda(a,b)>0$ and $\tau_\lambda(a',b)<\mu(a)$,

$$
||a-b|| \le \frac{1+\frac{\sqrt{d}\varepsilon}{2}}{1-\frac{\sqrt{d}\varepsilon}{2}}\cdot||a'-b|| \le \left(1+2\sqrt{d}\varepsilon\right)\cdot||a'-b||
$$

if $\varepsilon \le \frac{1}{\sqrt{d}}$. For $\mathrm{d}(a,b)=||a-b||^p$, it then immediately follows that

$$
\mathrm{d}(a,b) \le \left(1+2\sqrt{d}\varepsilon\right)^p\cdot\mathrm{d}(a',b) \le \left(1+2^{2p}d^{\frac{p}{2}}\varepsilon\right)\cdot\mathrm{d}(a',b)
$$

for all such $a$ and $a'$, and therefore one can argue

$$
\sum_{b \in B}\int_{N_\varepsilon(b)}\tau_\lambda(a,b)\mathrm{d}(a,b)da \le \left(1+2^{2p}d^{\frac{p}{2}}\varepsilon\right)\cdot\sum_{b \in B}\int_{N_\varepsilon(b)}\tau^*(a,b)\mathrm{d}(a,b)da.
$$

This concludes the statement of the Lemma, since we assume $d$ and $p$ are constants. $\qquad\square$

### F.6 Putting Everything Together

We finally prove Theorem A.1 by simply combining all above lemmas in Appendix F.

**Theorem A.1.** *Let $\mu$ be a continuous distribution with compact support $A \subset \mathbb{R}^d$, $\nu$ be a discrete distribution with support $B \subseteq \mathbb{R}^d$, $\lambda>0$ be a parameter and $d,p \ge 1$ be constants. Suppose $Q_1$ is the time complexity to compute $\int_\square \mu(a)da$ for any hypercube $\square \subseteq A$ and $Q_2$ is the time complexity to compute, given a point $b \in \mathbb{R}^d$ and constant $c \ge 0$, the radius $r \ge 0$ for which the Euclidean ball $\mathcal{B}(b,r)$ of radius $r$ centered at $b$ satisfies $\int_{\mathcal{B}(b,r)}\mu(a)da=c$. Then a $(\varepsilon,0)$-approximate $\lambda$-robust semi-discrete transport plan can be computed in $n\varepsilon^{-2d}\log\varepsilon^{-1}\left(n^{o(1)}\varepsilon^{-2d}\log\Delta+Q_1+Q_2\right)$ time if $\mathrm{d}(a,b)=||a-b||^p$ and in $O\left(n\varepsilon^{-2d}\log\varepsilon^{-1}\left(\varepsilon^{-2d-4}\log^3\Delta\log n+Q_1+Q_2\right)\right)$ time with probability at least $\frac{1}{2}$ if $\mathrm{d}(a,b)=||a-b||$.*

*Proof.* By Lemma F.1, we have that the set of hypercubes $\mathcal{P}$ satisfies $|\mathcal{P}|=O(n\varepsilon^{-2d}\log\varepsilon^{-1})$ and can be constructed in $O(n(\varepsilon^{-2d}\log\varepsilon^{-1}+\log n))$ time. Using $\mathcal{P}$, we construct the discrete distribution

$\hat{\mu}$ in $O(|\mathcal{P}| \cdot Q_1)$ time as in Appendix F.1. Then by Lemmas F.2 and F.10, we know that if one computes an $(O(\varepsilon), 0)$-approximate $\lambda$-capped transport plan $\hat{\tau}_\lambda$ between $\hat{\mu}$ and $\nu$, an $(O(\varepsilon), 0)$-approximate transport plan $\tau_\lambda$ between $\mu$ and $\nu$ can then be computed in $O\left(|\mathcal{P}| + n\left(\varepsilon^{-d} \log \Delta + Q_2\right)\right)$ time (we assume $i^*$ can be computed in $Q_2$ time for each $b \in B$). It therefore suffices to bound the time to construct the $(O(\varepsilon), 0)$-approximate $\lambda$-capped transport plan $\hat{\tau}_\lambda$.

For $p = 1$, we note that the graph $G$ in Appendix F.4 can be computed in $O(|E|) = O(n\varepsilon^{-3d} \log \Delta \log \varepsilon^{-1})$ time by Lemma F.5. Moreover, by Lemma F.6, we note that the minimum cost flow on $G$ has expected cost within a $(1 + O(\varepsilon))$ factor of the cost of an optimal $\lambda$-capped transport plan between $\hat{\mu}$ and $\nu$. Then the tree oracle described in Section F.4 takes $O(|V|) = O(n\varepsilon^{-2d} \log \Delta \log \varepsilon^{-1})$ time. By Lemmas F.8 and F.9, the tree oracle can be boosted into an $(O(\varepsilon), 0)$-approximate $\lambda$-capped transport plan in $O((|V| + |E|)\varepsilon^{-4} \log n \log^2 \Delta) = O(n\varepsilon^{-3d-4} \log^3 \Delta \log n \log \varepsilon^{-1})$ time.

For $p \geq 1$, we note that the graph $G$ in Appendix F.3 can be computed in $O(|E|) = O(n\varepsilon^{-3d} \log \Delta \log \varepsilon^{-1})$ time by Lemma F.3. Then the minimum-cost flow on $G$ can be computed in $O(|E|^{1+o(1)}) = O(n^{1+o(1)}\varepsilon^{-3d-o(1)} \log \Delta)$ time by the algorithm of [16]. One can then shortcut the minimum-cost flow on $G$ to form an $(O(\varepsilon), 0)$-approximate $\lambda$-capped transport plan between $\hat{\mu}$ and $\nu$ as in [2]. The resulting transport plan is an $(O(\varepsilon), 0)$-approximate $\lambda$-capped transport plan between $\hat{\mu}$ and $\nu$ by Lemma F.4.

We conclude that an $(O(\varepsilon), 0)$-approximate $\lambda$-robust transport plan between $\mu$ and $\nu$ can be computed in $O(n^{1+o(1)}\varepsilon^{-3d-o(1)} \log \Delta)$ time for $p \geq 1$ and in $O(n\varepsilon^{-3d-4} \log^3 \Delta \log n \log \varepsilon^{-1})$ time for $p = 1$ with probability at least $1/2$. To get an $(\varepsilon, 0)$-approximate $\lambda$-robust transport plan, one can simply choose an appropriately smaller value of $\varepsilon$. $\qquad\square$

