# OpenReview forum: "Efficient Algorithms for Robust and Partial Semi-Discrete Optimal Transport"
_NeurIPS.cc/2025/Conference — NeurIPS 2025 poster_

### Official Review · Reviewer_NHUU · 2025-07-01

**Clarity:** 3
**Significance:** 2
**Originality:** 2
**Rating:** 5
**Confidence:** 4

**Summary:**

In this paper, the authors extend the deterministic semi-discrete optimal transport problem, which involves transporting a continuous distribution to a discrete one, to a stochastic setting. Noise and outliers are considered via two variants: $\alpha$-partial optimal transport, where only an $\alpha$-fraction of the probability mass is transported, and $\lambda$-robust optimal transport, where mass can be discarded at a cost of $\lambda$ per unit.

The authors propose a novel theoretical characterization of exact solutions using restricted Voronoi cells and weight functions on the discrete distribution domain, under specific conditions. For inexact solutions, they rigorously derive a connection between the two problem variants and introduce a reduction scheme along with corresponding error bounds. To solve the $\lambda$-robust OT problem with high accuracy, the authors generalize the combinatorial cost-scaling paradigm, incorporating all necessary modifications. The resulting algorithm achieves a complexity bound of $n^{O(d)} \log(\Delta/\varepsilon)$. All results hold for arbitrary cost functions, with additional acceleration techniques developed specifically for Wasserstein distances.

**Questions:**

How long does it take to compute the $\Phi$ factor, both for known distributions (e.g., normal, exponential) and unknown distributions? What methods are available for this?


 Can your methods handle distributions for which only sample data is available?


Are the obtained convergence rates optimal with respect to their dependencies on parameters such as n (sample size), d (dimensionality) and others?

**Ethical Concerns:**

["NO or VERY MINOR ethics concerns only"]

**Final Justification:**

I'm satisfied authors' answers.

**Limitations:**

The paper lacks practical validation of the proposed methods, particularly comparisons with deterministic counterparts in scenarios where robustness is essential. Even basic 2D experiments on synthetically corrupted data could have provided a proof of concept while demonstrating the methods' robustness and quality.

While the theoretical contributions are significant, their novelty appears somewhat incremental, as they primarily combine (in a non-trivial way) existing ideas, problem formulations, and methodologies.

Minor weakness: the notation $\delta$ is ambiguous, it is used as a parameter in Section 3, and as units in Introduction.

**Paper Formatting Concerns:**

No problem

**Quality:**

3

**Strengths And Weaknesses:**

Optimal Transport is an important topic, and new robust algorithms represent valuable contributions to both theory and practice.

The strongest side of the paper is its complex rigorous theoretical analysis of the proposed methods and properties of the solutions. These findings are interesting and cover all aspects of the considered setups and connections between them.

Despite the amount of information, the paper’s writing is clear and well-structured making it easy to follow.

For weakness, see the Limitations section.

---

> ### Author Rebuttal · Authors · 2025-07-31
>
> We appreciate your thoughtful review and comments. We answer your main questions and concerns below.
>
>
> > How long does it take to compute the $\Phi$ factor, both for known distributions (e.g. normal, exponential) and unknown distributions?
>
> The factor of $\Phi$ depends on how the distribution is represented. If the distribution is a parametric distribution with a constant number of parameters, e.g. normal or exponential (mixture), then a constant degree polygonal query region can be integrated over in constant time using numerical methods. On the other hand if the continuous distribution is represented as a histogram or piecewise-linear function, then the time depends on the complexity of the distribution inside the query region. In low dimensions, one could use some geometric data structures to expedite this computation, but in high dimensions, a simple method would be more practical. Finally, if the distribution is represented as a large representative sample, then the $\Phi$ factor simply corresponds to assigning each sample point to the region of the partition it is contained in. In turn, our algorithm successfully computes a transport plan between a discrete distribution and a large sample of an unknown continuous distribution.
>
> ---
> ---
> > Can your methods handle distributions for which only sample data is available?
>
> Our algorithm can handle sample data from a distribution, as long as one global sample is used to approximate the continuous mass inside each region of the arrangement (rather than sampling from the distribution for each mass query).
>
> ---
> ---
> > Are the obtained convergence rates optimal with respect to their dependencies on parameters such as $n, d$?
>
> A: As noted in Section 2, as well as references [9] and Bourne et al., the optimal semi-discrete transport plan requires computing a (restricted) weighted Voronoi diagram. There are known lower bounds of $n^{\Omega(d)}$ for the complexity of Voronoi diagrams in the worst case in $d$ dimensions (see e.g. Har-Peled "Geometric Approximation Algorithms"), and therefore our high-precision algorithm is near-optimal. While we do not have a rigorous lower bound for the optimal runtime of a low-precision algorithm in constant dimension, we suspect that optimal transport is at least as hard as nearest neighbor search (a naive plan would assign each point to its nearest neighbor). The runtime of our low-precision algorithm is near linear in $n$ and incurs comparable dependence in $\varepsilon$ and $d$ to the best known algorithms for approximate nearest neighbor search.
>
> ---
> ---
>
> >The paper lacks practical validation of the proposed methods, particularly comparisons with deterministic counterparts in scenarios where robustness is essential. Even basic 2D experiments on synthetically corrupted data could have provided a proof of concept while demonstrating the methods' robustness and quality.
>
> While the original submission emphasized our theoretical contributions, we appreciate your interest in practical implementation and experimental evaluation. In response, we include below preliminary experimental results. A more extensive evaluation will be added in the next version.
>
> **Implementation Details.**
> We provide a Python implementation of our algorithm. However, it relies on two core procedures that lack efficient implementations in Python:
> 1. Estimating continuous mass within regions defined by the arrangement of the restricted Voronoi cells and their expansions.
> 2. Maintaining a dynamic tree data structure as per Sleator and Tarjan (reference [44] in the appendix).
>
> To address these challenges in our prototype:
> - For (1), we approximate the mass within each region using a fine grid, summing the masses of grid points lying inside the region.
> - For (2), we simulate the dynamic tree by naïvely iterating over all edges.
>
> An efficient implementation will require optimized versions of these components, which remains a technical challenge in Python.
>
> ---
> **Preliminary Results on Algorithm Performance.**
> We evaluated our implementation using a continuous distribution sampled from a Gaussian distribution inside the unit square. The discrete support consists of $n$ samples drawn from the same distribution, each assigned a mass of $1/n$. We set $\lambda = 0.2$ and $\varepsilon = 0.02$. Results are averaged over 10 runs for each $n$.
>
> **Key Findings:**
> - The additive error stays within the target $\varepsilon = 0.02$.
> - The number of iterations in the two steps of our algorithm (Step 1 and 2) remains well below our upper bounds proved in the paper ($6n$ and $12n$ as described in Lemma 4.3 and D.9).
> - The total path and cycle lengths computed in the two steps grow subquadratically in $n$, aligning with our $O(n^2)$ complexity analysis (Section 4.3).
>
> The details of our experiments are presented in the following table.
>
> | n | Error | Step 1 iterations | Step 2 iterations | Step 1 paths and cycles length | Step 2 paths and cycles length | Regions |
> | :--: | :--: | :--: | :--: | :--: | :--: | :--: |
> | 10 | 0.009 | 1 | 4 | 41 | 361 | 198  |
> | 20 | 0.018 | 2 | 4 | 320 | 991  | 529  |
> | 30 | 0.021 | 3 | 5 | 688 | 1719  | 862  |
> | 40 | 0.015 | 3 | 5 | 979 | 2385  | 1212  |
> | 50 |  0.018 | 4 | 5 | 1603 | 3061  | 1544  |
> | 60 | 0.015 | 5 | 6 | 1973 | 3673  | 1855  |
> | 70 | 0.017 | 5 | 6 | 2512 | 4129  | 2130  |
> | 80 | 0.016 | 6 | 7 | 2845 | 4785  | 2421  |
>
> ---
> **Robustness of $\alpha$-OPT.**
>
> We examined the effect of 10% noise in the continuous distribution (added in the top-right corner) and measured how much noise was transported under various $\alpha$ values. Results confirm the robustness of $\alpha$-OPT: for $\alpha$ < 0.9, almost no noise mass is transported.
>
>
> |Alpha| Cost| \% inliers transported| \% noise transported|
> | :--: | :--: | :--: | :--: |
>  |0.700| 0.010| 0.777| 0.000|
>  |0.725| 0.011| 0.804| 0.000|
>  |0.750| 0.012| 0.830| 0.015|
>  |0.775| 0.013| 0.859| 0.005|
>  |0.800| 0.014| 0.887| 0.005|
>  |0.825| 0.016| 0.914| 0.009|
>  |0.850| 0.017| 0.942| 0.014|
>  |0.875| 0.019| 0.970| 0.005|
>  |0.900| 0.022| 0.996| 0.020|
>  |0.925| 0.031| 0.999| 0.244|
>  |0.950| 0.045| 0.999| 0.496|
>  |0.975| 0.060| 0.999| 0.748|
>
> **Robustness of $\lambda$-ROT.**
> We conducted similar experiments for $\lambda$-ROT. As $\lambda$ increases, total mass transported grows, but noise transport remains limited when the transported mass is under 0.9.
>
> |Lambda| Cost| Cost ROT| \% inliers transported| \% noise transported| Mass transported|
>    | :--: | :--: | :--: | :--: | :--: | :--: |
> | 0.05| 0.014| 0.024| 0.896| 0.004| 0.807|
> | 0.10| 0.020| 0.031| 0.980| 0.009| 0.884|
>    | 0.15| 0.021| 0.037| 0.991| 0.012| 0.894|
> | 0.20| 0.023| 0.042| 0.996| 0.046| 0.902|
> | 0.25| 0.024| 0.047| 0.999| 0.073| 0.907|
> | 0.30| 0.023| 0.051| 0.999| 0.064| 0.907|
> | 0.35| 0.025| 0.055| 0.999| 0.141| 0.914|
> | 0.40| 0.027| 0.061| 1.0| 0.147| 0.915|
> | 0.45| 0.028| 0.063| 1.0| 0.212| 0.921|
> | 0.50| 0.034| 0.066| 1.0| 0.349| 0.935|
> | 0.55| 0.041| 0.068| 1.0| 0.506| 0.951|
>
> Please let us know if any further clarification or experiments would strengthen the paper. We are happy to incorporate additional feedback into the next version of our paper.

---

> > ### Comment · Reviewer_NHUU · 2025-08-05
> >
> > I would like to thank the authors for their detailed responses which have fully addressed my questions. If time allows, it would be valuable to examine a broader variety of more complex distributions and noise outliers within the same experimental framework (e.g., different continuous and discrete distributions, including heavier-tailed ones).

---

> > > ### Author Response · Authors · 2025-08-06
> > >
> > > Thank you for the helpful suggestion. In the next version of our paper, we will expand our experiments to include more complex distributions. Below, we present one such additional experiment.
> > >
> > >
> > >
> > > In this experiment, we approximate heavy-tailed behavior within a bounded region (the unit square) using mixtures of Beta distributions that induce skewed and dispersed mass. Specifically, the discrete distribution consists of $n=20$ samples drawn from a Beta($\alpha=2,\beta$) distribution, with $\beta$ varying in the range [3,10]. As $\beta$ increases, the distribution becomes increasingly concentrated near [0,0]. The continuous distribution is constructed by contaminating the same base distribution with 15\% noise drawn from a Beta($\alpha$=5,$\beta$=2) distribution (a distribution concentrated close to [0.9,0.9]). For smaller values of $\beta$, the two components overlap significantly; for larger values, they become well-separated. Each experiment was repeated 10 times, and we report the averaged results.
> > >
> > >
> > > These experiments confirm that our algorithm remains reliable even when inliers and outliers overlap. In particular, the computed partial and robust transport plans consistently focus on the inlier mass while effectively disregarding much of the outlier mass—especially as the main and noise distributions become more dissimilar. Performance metrics (e.g., computational complexity, convergence rate, and error) remain consistent with those observed in the Gaussian case.
> > >
> > >
> > > ---
> > > ---
> > >
> > > **Robustness of $0.85$-OPT:**
> > >
> > > |$\beta$|Cost|\% Inliers transported|\% Outliers transported|
> > > | :--: | :--: | :--: | :--: |
> > > |3|0.016|0.926|0.419|
> > > |4|0.016|0.957|0.245|
> > > |5|0.014|0.977|0.132|
> > > |6|0.013|0.978|0.125|
> > > |7|0.013|0.990|0.056|
> > > |8|0.012|0.991|0.053|
> > > |9|0.011|0.994|0.035|
> > > |10|0.011|0.996|0.021|
> > >
> > > ---
> > > ---
> > >
> > > **Robustness of $0.15$-ROT:**
> > >
> > > |$\beta$|Cost|Robust cost|\% Inliers transported|\% Outliers transported|Total mass transported|
> > > | :--: | :--: | :--: | :--: | :--: | :--: |
> > > |3|0.022|0.041|0.980|0.493|0.907|
> > > |4|0.020|0.041|0.992|0.350|0.896|
> > > |5|0.019|0.042|0.995|0.245|0.883|
> > > |6|0.017|0.041|0.998|0.191|0.877|
> > > |7|0.015|0.042|0.998|0.091|0.862|
> > > |8|0.013|0.041|0.999|0.085|0.862|
> > > |9|0.012|0.040|0.999|0.053|0.857|
> > > |10|0.011|0.040|1.000|0.048|0.857|
> > >
> > > ---
> > > ---
> > >
> > > **Performance of our Algorithm:**
> > >
> > > |$\beta$|Error|Step 1 iterations|Step 2 iterations|Step 1 paths and cycles length|Step 2 paths and cycles length|Regions|
> > > | :--: | :--: | :--: | :--: | :--: | :--: | :--: |
> > > |3|0.012|4|6|420|993|495|
> > > |4|0.016|7|8|332|1026|495|
> > > |5|0.005|6|7|190|986|462|
> > > |6|0.007|6|8|131|971|435|
> > > |7|0.010|9|8|229|886|421|
> > > |8|0.008|3|6|161|810|382|
> > > |9|0.007|6|8|89|849|388|
> > > |10|0.007|3|5|55|766|351|

---

### Official Review · Reviewer_CmYB · 2025-07-02

**Clarity:** 2
**Significance:** 2
**Originality:** 2
**Rating:** 4
**Confidence:** 2

**Summary:**

This paper studies two formulations of semi-discrete optimal transport (OT) problem: $\alpha$-optimal partial transport and $\lambda$-robust OT. The authors provide theoretical investigations on the optimal solutions in these formulations, connection between the algorithms for each of the settings and present their own algorithm for each of the formulations.

**Questions:**

- Could you conduct some experiments justifying your theoretical conclusions?
- Could you please provide a comment regarding the originality of your approach which follows the ideas of the one suggested in (Agarwal et al., 2024)?

**Ethical Concerns:**

["NO or VERY MINOR ethics concerns only"]

**Final Justification:**

During the rebuttal, the authors have addressed my main concerns. That is, they conducted the experiments which I have suggested, and promised to improve the clarity of the paper, e.g., improve the paper text, add missing sections, etc. Thus, I increase my score to slightly positive (4). I do not assign higher score since *all* of the experiments were conducted during the rebuttal. I think that the question if this represent a major revision or not could be discussed among the Reviewers and Area Chairs.

**Limitations:**

The limitations of the approach are stated only in the NeurIPS paper checklist. I kindly suggest the authors create a separate ‘Limitations’ section in the main text or Appendix of their paper and list the limitations there. One limitation is missing here - the paper establishes results only for the $p$-Wasserstein distance as the OT cost.

**Paper Formatting Concerns:**

Paper misses the Conclusion/Discussion section with summarization of the achieved results in a general form. There is no separate 'Limitations' section also, the limitations are included only in the NeurIPS checklist.

**Quality:**

2

**Strengths And Weaknesses:**

**Strengths.** The paper provides a lot of theoretical results on the connection and properties of partial and robust OT formulations, their solutions and corresponding algorithms.

**Weaknesses.** My main concern corresponds to the **lack of experiments** in the paper. The authors present a lot of theoretical results related to partial/robust OT formulations, propose new algorithms to solve these problems, establish their convergence properties, but **do not test them in any experiments**. I am confused by this fact, since while the paper proposes the algorithms, it should provide at least some experimental validation of their performance. I am also interested in real-world applications of the proposed algorithms, i.e., experiments showing that semi-discrete partial/robust OT formulations and corresponding algorithms are useful in practice. However, at this step, even toy experiments are missing.

The related concern corresponds to the paper **clarity**. The paper is really hard to follow, most of the text is written in a technical, mathematical style starting from the ‘Introduction’ section. While I understand that it mainly focuses on theoretical investigations, the text should be written in an accessible and understandable form:
-  I kindly suggest the authors exclude complex explanations of their results from the ‘Introduction’ section (lines 60-94) and rather include the paper contributions in a general form trying to avoid specific mathematical notations and definitions which are then introduced one more time in main sections.
- I also have concerns regarding the clarity of OT formulations. In the classic OT problem, the marginals of the plan should coincide with input and target measures. However, in lines 98-104, you give the definition of the OT plan replacing the equality constraint on the marginals with the inequalities. While I understand that the purpose was to give a general definition applicable for the partial/robust OT cases, the explanation might be ambiguous for the reader and should be clarified.
- I am also wondering why the paper lacks a ‘Conclusion’/’Discussion’ section with summarization of the achieved results in a general form.
- Another minor concern corresponds to the lines 117-118 which operate with the notions of LP formulation/duality which were not given in the text before that.

Another restriction of the established theory corresponds to the usage of the $p$-Wasserstein distance in the partial/robust OT formulations. This point should be stated as a limitation of this work since general OT formulations might cover many other costs.

**In summary,** I have doubts about the relevance of this paper for the current conference. The paper focuses on the theoretical results related to the  partial/robust OT formulations and does not include proper experimental validation of the suggested algorithms  (no experiments at all). Theoretical results might represent an interest for the ML community, but their motivation, ideas and originality should be clearly explained in the paper’s main text while now the exposition is very technical and hard to follow.  However, currently the exposition of the paper seems to be more suitable for a mathematical venue where the significance of the established theory (nearly 30 pages in Appendix) could be carefully checked.

---

> ### Author Rebuttal · Authors · 2025-07-31
>
> We appreciate the reviewer’s comment regarding the density of technical details in the introduction. In view of this comment, we will fix the introduction to present the motivation, high-level ideas, and contributions in a clearer and more accessible manner, deferring technical details to later sections. We will also do our best to make the later sections more accessible.
>
>
> ---
> ---
> > Could you conduct some experiments justifying your theoretical conclusions?
>
> **Summary of Contributions.**
>  Our paper presents the following key contributions:
>  - A characterization of partial and robust variants of the optimal transport (OT) problem in the semi-discrete setting,
>  - Theoretical guarantees on the approximation error when using an approximate $\lambda$-ROT solver to compute an approximate $\alpha$-OPT plan, and,
>  - Two algorithmic solutions for the $\lambda$-ROT problem.
>
> While the original submission emphasized our theoretical contributions, we appreciate your interest in practical implementation and experimental evaluation. In response, we include below preliminary experimental results. A more extensive evaluation will be added in the next version.
>
> **Implementation Details.**
> We provide a Python implementation of our algorithm. However, it relies on two core procedures that lack efficient implementations in Python:
> 1. Estimating continuous mass within regions defined by the arrangement of the restricted Voronoi cells and their expansions.
> 2. Maintaining a dynamic tree data structure as per Sleator and Tarjan (reference [44] in the appendix).
>
> To address these challenges in our prototype:
> - For (1), we approximate the mass within each region using a fine grid, summing the masses of grid points lying inside the region.
> - For (2), we simulate the dynamic tree by naïvely iterating over all edges.
>
> An efficient implementation will require optimized versions of these components, which remains a technical challenge in Python.
>
> ---
> **Preliminary Results on Algorithm Performance.**
> We evaluated our implementation using a continuous distribution sampled from a Gaussian distribution inside the unit square. The discrete support consists of $n$ samples drawn from the same distribution, each assigned a mass of $1/n$. We set $\lambda = 0.2$ and $\varepsilon = 0.02$. Results are averaged over 10 runs for each $n$.
>
> **Key Findings:**
> - The additive error stays within the target $\varepsilon = 0.02$.
> - The number of iterations in the two steps of our algorithm (Step 1 and 2) remains well below our upper bounds proved in the paper ($6n$ and $12n$ as described in Lemma 4.3 and D.9).
> - The total path and cycle lengths computed in the two steps grow subquadratically in $n$, aligning with our $O(n^2)$ complexity analysis (Section 4.3).
>
> The details of our experiments are presented in the following table.
>
> | n | Error | Step 1 iterations | Step 2 iterations | Step 1 paths and cycles length | Step 2 paths and cycles length | Regions |
> | :--: | :--: | :--: | :--: | :--: | :--: | :--: |
> | 10 | 0.009 | 1 | 4 | 41 | 361 | 198  |
> | 20 | 0.018 | 2 | 4 | 320 | 991  | 529  |
> | 30 | 0.021 | 3 | 5 | 688 | 1719  | 862  |
> | 40 | 0.015 | 3 | 5 | 979 | 2385  | 1212  |
> | 50 |  0.018 | 4 | 5 | 1603 | 3061  | 1544  |
> | 60 | 0.015 | 5 | 6 | 1973 | 3673  | 1855  |
> | 70 | 0.017 | 5 | 6 | 2512 | 4129  | 2130  |
> | 80 | 0.016 | 6 | 7 | 2845 | 4785  | 2421  |
>
>
> ---
> **Reduction from $\alpha$-OPT to $\lambda$-ROT.**
> We evaluated our reduction from $\alpha$-OPT to $\lambda$-ROT by using the Sinkhorn algorithm to approximate $\lambda$-ROT plans, followed by binary search over $\lambda$ to compute an $\alpha$-OPT plan. The continuous distribution in our experiments is an exponential distribution over the unit square, centered at the bottom-left corner, with 10% noise added to the top-right corner. The discrete set $B$ consists of $n=400$ samples from the same exponential distribution, each assigned mass $1/n$. We target a transported mass of $\alpha$ = 0.9.
>
> *Additive Error:* For any fixed regularization parameter, we evaluate the additive error (with respect to the optimal $\lambda$-ROT solution) incurred by Sinkhorn, and compare it to the additive error of our computed $\alpha$-OPT solution (with respect to the optimal $\alpha$-OPT). We observe that the error in $\alpha$-OPT closely mirrors that of $\lambda$-ROT across different choices of regularization parameter. Moreover, as the regularization parameter decreases, the additive errors in $\alpha$-OPT also diminish. This behavior is consistent with the theoretical guarantee established in Theorem 3.5.
>
> |Regularization | Robust Cost | Partial Cost | Additive error of ROT | Additive error of OPT |
> | :--: | :--: | :--: | :--: | :--: |
> |0.01|0.176|0.084|0.000|0.000|
> |0.02|0.186|0.095|0.009|0.012|
> |0.03|0.182|0.100|0.012|0.017|
> |0.04|0.193|0.112|0.023|0.029|
> |0.05|0.205|0.125|0.034|0.041|
> |0.06|0.216|0.137|0.046|0.053|
> |0.07|0.227|0.148|0.057|0.065|
> |0.08|0.238|0.159|0.068|0.075|
>
> *Approximation factor:* Similarly, we evaluate the approximation factor for each method using the same setup. Once again, we find that the approximation factor in $\alpha$-OPT closely mirrors that of $\lambda$-ROT over a range of regularization parameters. As the regularization parameter decreases, the approximation factor in $\alpha$-OPT also approaches one, further corroborating the theoretical prediction in Theorem 3.5.
>
> |Regularization | Robust Cost | Partial Cost | Relative error of ROT | Relative error of OPT |
> | :--: | :--: | :--: | :--: | :--: |
> |0.01|0.176|0.084|0.998|1.003|
> |0.02|0.186|0.095|1.050|1.142|
> |0.03|0.182|0.100|1.071|1.200|
> |0.04|0.193|0.112|1.136|1.346|
> |0.05|0.205|0.125|1.203|1.493|
> |0.06|0.216|0.137|1.269|1.638|
> |0.07|0.227|0.148|1.334|1.776|
> |0.08|0.238|0.159|1.397|1.904|
>
> Please let us know if any further clarification or experiments would strengthen the paper. We are happy to incorporate additional feedback into the next version of our paper.
>
> ---
> ---
> > Could you please provide a comment regarding the originality of your approach which follows the ideas of the one suggested in Agarwal et al., 2024?
>
> **Challenges in Extending Cost-Scaling to Partial Transport:**
>  Cost-scaling algorithms have proven effective for solving the complete optimal transport (OT) problem. However, extending these techniques to partial optimal transport introduces substantial challenges—even in fully discrete settings (see, e.g., Ramshaw and Tarjan, FOCS 2012 [37])—and these difficulties are further amplified in the semi-discrete setting.
> A central obstacle lies in the behavior of the dual weights: in complete OT, the optimal solution is invariant under translations of the dual weight vector for the discrete set $B$. In contrast, in partial OT, the absolute magnitudes of the dual weights are crucial, as they control the truncation of Voronoi cells and thus directly determine which regions of the continuous distribution are assigned to each discrete point.
>
> **Limitations of Existing Scaling Algorithms.**
>  Recent cost-scaling algorithms (e.g., Agarwal et al., SODA 2024; NeurIPS 2024) increase the magnitudes of the dual weights for the discrete set $B$, and implicitly for the continuous domain $A$, over successive scales. However, these methods lack a mechanism to reduce or correct the dual weights once they are overestimated. This renders existing scaling algorithms unsuitable for partial OT, where accurate dual weight magnitudes are critical.
>
> **Our Solution.**
>  To overcome this limitation, we introduce a novel step that uses consolidating paths and augmenting paths to reroute mass in the residual graph and correct overestimated dual weight magnitudes. The addition of this step makes it non-trivial to establish both the correctness of the modified algorithm and its efficiency matching that of Agarwal et al., NeurIPS 2024. These challenges necessitate several new ideas, as detailed in Lemmas 4.2 and 4.3 of the main text and Lemma D.9 in the Appendix.
>
> **Geometric Interpretation of this limitation and consolidating paths.**
>  At the start of a scale $\delta$, some mass from the continuous distribution may lie well inside the restricted Voronoi cells of the discrete points $B$, yet remain untransported—these are what we call violating deficit regions (see Figure 3). In the complete OT setting, such violations are naturally resolved as the plan evolves to transport all mass. In the partial OT setting, however, this issue is non-trivial and must be explicitly resolved to ensure correctness. Our consolidating paths are specifically designed to eliminate all violating deficit regions by restructuring the transport plan to ensure that the total transported mass lies strictly within the restricted Voronoi regions, thereby guaranteeing correctness of the partial OT plan.

---

> > ### Comment · Reviewer_CmYB · 2025-08-06
> > **Thank you & Additional questions**
> >
> > I thank the authors for the provided answers and conducted experiments. However, there are several aspects which still raise my concerns. First, some of the weaknesses which I mentioned in my initial review left undiscussed, e.g., lack of the discussion section, absence of limitation stating that the algorithm deals only with a quadratic cost. I kindly ask the authors to take these points into consideration during the preparation of the revised version of the paper. Second, I have several questions about preliminary experiments conducted during the rebuttal:
> > - As far as I understand, you work with 2-dimensional data in these experiments? I am wondering whether the proposed algorithm is scalable to higher dimensions.
> > - How exactly do you calculate the additive and relative errors?
> > - Is it possible to compare your approach with alternatives in the described experimental setup? For example, comparison with (Agarwal et al., 2024) seems to be valuable. It is interesting to see how this approach behaves in case of the outliers in the data.
> >
> > I will appreciate if the authors could provide clarifications on these points during the remaining days of rebuttal.

---

> > > ### Author Response · Authors · 2025-08-07
> > >
> > > We sincerely thank the reviewer for their follow-up questions.
> > >
> > > ---
> > > ---
> > >
> > > **Applicability to Other Distance Functions:**
> > > Our algorithm and its analysis extend to distance functions where both the weighted bisectors and the balls are semi-algebraic sets of constant degree. This includes all $\ell_p^q$ norms for any constant $p, q\ge 1$. We would like to note that for arbitrary distance functions, the weighted Voronoi diagram might have an unbounded complexity. In such cases, as opposed to a discrete transport plan, a semi-discrete transport plan may not have a finite representation. We will clarify this in the next version of the paper and also add a **Conclusion** section summarizing our contributions and discussing the range of admissible cost functions.
> > >
> > > ---
> > > ---
> > > **Scalability to Higher Dimensions:**
> > > As noted in Section 2 and discussed in reference [9], computing the optimal semi-discrete transport plan requires constructing a (restricted) weighted Voronoi diagram. It is known that the combinatorial complexity of such diagrams in $d$ dimensions can be as high as $n^{\Omega(d)}$ in the worst case. Therefore, the execution time of our exact algorithm is $n^{O(d)}$, which is near-optimal given these lower bounds.  However, the complexity of the (restricted) Voronoi diagram is small in many instances, e.g., when points are chosen randomly from a distribution. Assuming there is an efficient algorithm for computing the Voronoi diagram in such cases, our algorithm will also be efficient and not incur the worst-case cost.
> > >
> > > We also evaluated the implementation of our algorithm on a 3D Gaussian distribution with 10% additive noise. The results below demonstrate that our algorithm continues to behave predictably with respect to error and iteration counts, even as the dimension increases:
> > >
> > > |n|Error|Step 1 iterations|Step 2 iterations|Step 1 paths and cycles length|Step 2 paths and cycles length|Regions|
> > > | :--: | :--: | :--: | :--: | :--: | :--: | :--: |
> > > |10|0.002|2|5|25|454|389|
> > > |20|0.016|5|6|303|2514|1776|
> > > |30|0.013|5|6|500|4664|3292|
> > > |40|0.016|11|8|1587|7057|4807|
> > > |50|0.028|12|11|2820|10954|6681|
> > > |60|0.024|16|12|3644|13141|8072|
> > > |70|0.020|15|10|4735|15310|9008|
> > >
> > > ---
> > > ---
> > >
> > > **Computation of Additive and Relative Errors:**
> > > To compute the additive error for Sinkhorn, we take the absolute difference in cost of the plan produced by Sinkhorn and the optimal transport plan. The approximation factor for the ROT computed by the Sinkhorn algorithm is the ratio of the cost of Sinkhorn’s plan to the optimal transport plan.
> > >
> > > ---
> > > ---
> > > **Comparison with Agarwal et al. (2024):**
> > > The algorithm of Agarwal et al. (2024) addresses the complete semi-discrete OT problem, and ours generalizes it to compute partial transport plans too. Notably, while their paper does not provide an implementation, by setting appropriate parameters, our implementation serves as an implementation of their algorithm.
> > >
> > > In the revised version of our paper, we will include images illustrating how the partial semi-discrete transport plan selectively transports inlier mass in contrast to complete transport plans (as in Agarwal et al. (2024)), which are distorted by the presence of outliers.
> > >
> > > ---
> > > ---
> > > **Summary:**
> > > Finally, we would like to reiterate that the **primary contributions of our work** are new characterizations of partial semi-discrete OT, duality-based reductions between partial and robust semi-discrete OT, and two efficient algorithms for partial semi-discrete OT with provable guarantees on their performance. To complement this, we have implemented a prototype of our algorithm as a proof of concept. An optimized implementation that addresses all algorithmic engineering issues is beyond the scope of this paper. If the paper is accepted, we are committed to releasing the code publicly for the benefit of ML community and also to facilitate future research.

---

> > > > ### Comment · Reviewer_CmYB · 2025-08-08
> > > > **Thank you**
> > > >
> > > > The authors have addressed my major concerns, thus, I will increase my score.

---

### Official Review · Reviewer_umB2 · 2025-07-03

**Clarity:** 4
**Significance:** 3
**Originality:** 3
**Rating:** 4
**Confidence:** 3

**Summary:**

Paper studies two robust formulations of optimal transport (OT) : $\alpha$-optimal partial transport ($\alpha$-OPT), which transports only a fraction of the mass of the two input distributions, and $\lambda$-robust optimal transport ($\lambda$-ROT), which penalizes untransported mass via a total variation regularization. While both variants have been well-studied in fully discrete settings, e.g. by Caffarelli and MacCann (ref [9] of the paper), this work proposes to extend them to semi-discrete OT. The authors introduce novel characterizations of optimal solutions using restricted weighted Voronoi diagrams, which captures the points of the Voronoi cells that are within a given distance.

Authors show that any solver for $\lambda$-ROT can be leveraged to solve $\alpha$-OPT, thus unifying the algorithmic treatment of these problems. They propose two algorithmic contributions: (1) a high-precision combinatorial algorithm for semi-discrete $\lambda$-ROT using a refined cost-scaling approach (2) a near-linear-time approximation algorithm. Theoretical analysis includes correctness proofs, runtime bounds, and approximation guarantees even when only a fraction of the mass is transported.

The paper does not include empirical experiments as it is primarily theoretical in nature. Authors clearly articulate limitations, such as the exponential dependence on dimension.

**Questions:**

- Could you provide numerical experiments (even in simple cases) that could validate experimentally your algorithm and show its practical utility?
- how does this work relates to Bourne et al., 2018?
- the connection between $\alpha$-OPT and $\lambda$-ROT has been thoroughly examined in the work of Caffarelli and MacCann (ref [9] of the paper). Could you clarify how your results relate to their duality findings?

**Ethical Concerns:**

["NO or VERY MINOR ethics concerns only"]

**Final Justification:**

I keep a positive evaluation of the paper, leaning toward acceptance. I found the paper really clear and it provides new results about robust semi-discrete OT problems. My main concern is about the lack of experimental evaluation, beyond toy experimentation.

**Limitations:**

Yes

**Paper Formatting Concerns:**

yes

**Quality:**

3

**Strengths And Weaknesses:**

Strengths:
- The paper addresses an important problem: robust and partial OT in semi-discrete settings, for which no/few prior algorithms exists.
- The characterizations via restricted Voronoi diagrams are very elegant and provide useful geometric insight.
- The paper is exceptionally well-crafted, presenting the ideas clearly and accessibly despite the complexity of the subject matter.
- Although I didn't check all the proofs, I think the algorithm and results are reasonable.

Weaknesses:
- While the contribution is theoretical in nature, numerical evaluations are not provided.
- the connection between this work and Bourne et al., 2018, are not discussed

Bourne, D. P., Schmitzer, B., & Wirth, B. (2018). Semi-discrete unbalanced optimal transport and quantization. arXiv preprint arXiv:1808.01962.

---

> ### Author Rebuttal · Authors · 2025-07-31
>
> We appreciate your thoughtful review and comments. Below, we answer your main questions and concerns.
>
> ---
> ---
> >Could you provide numerical experiments (even in simple cases) that could validate experimentally your algorithm and show its practical utility?
>
> **Summary of Contributions.**
>  Our paper presents the following key contributions:
>  - A characterization of partial and robust variants of the optimal transport (OT) problem in the semi-discrete setting,
>  - Theoretical guarantees on the approximation error when using an approximate $\lambda$-ROT solver to compute an approximate $\alpha$-OPT plan, and,
>  - Two algorithmic solutions for the $\lambda$-ROT problem.
>
> While the original submission emphasized our theoretical contributions, we appreciate your interest in practical implementation and experimental evaluation. In response, we include below preliminary experimental results. A more extensive evaluation will be added in the next version.
>
> **Implementation Details.**
> We provide a Python implementation of our algorithm. However, it relies on two core procedures that lack efficient implementations in Python:
> 1. Estimating continuous mass within regions defined by the arrangement of the restricted Voronoi cells and their expansions.
> 2. Maintaining a dynamic tree data structure as per Sleator and Tarjan (reference [44] in the appendix).
>
> To address these challenges in our prototype:
> - For (1), we approximate the mass within each region using a fine grid, summing the masses of grid points lying inside the region.
> - For (2), we simulate the dynamic tree by naïvely iterating over all edges.
>
> An efficient implementation will require optimized versions of these components, which remains a technical challenge in Python.
>
> ---
> **Preliminary Results on Algorithm Performance.**
> We evaluated our implementation using a continuous distribution sampled from a Gaussian distribution inside the unit square. The discrete support consists of $n$ samples drawn from the same distribution, each assigned a mass of $1/n$. We set $\lambda = 0.2$ and $\varepsilon = 0.02$. Results are averaged over 10 runs for each $n$.
>
> **Key Findings:**
> - The additive error stays within the target $\varepsilon = 0.02$.
> - The number of iterations in the two steps of our algorithm (Step 1 and 2) remains well below our upper bounds proved in the paper ($6n$ and $12n$ as described in Lemma 4.3 and D.9).
> - The total path and cycle lengths computed in the two steps grow subquadratically in $n$, aligning with our $O(n^2)$ complexity analysis (Section 4.3).
>
> The details of our experiments are presented in the following table.
>
> | n | Error | Step 1 iterations | Step 2 iterations | Step 1 paths and cycles length | Step 2 paths and cycles length | Regions |
> | :--: | :--: | :--: | :--: | :--: | :--: | :--: |
> | 10 | 0.009 | 1 | 4 | 41 | 361 | 198  |
> | 20 | 0.018 | 2 | 4 | 320 | 991  | 529  |
> | 30 | 0.021 | 3 | 5 | 688 | 1719  | 862  |
> | 40 | 0.015 | 3 | 5 | 979 | 2385  | 1212  |
> | 50 |  0.018 | 4 | 5 | 1603 | 3061  | 1544  |
> | 60 | 0.015 | 5 | 6 | 1973 | 3673  | 1855  |
> | 70 | 0.017 | 5 | 6 | 2512 | 4129  | 2130  |
> | 80 | 0.016 | 6 | 7 | 2845 | 4785  | 2421  |
>
> Please let us know if any further clarification or experiments would strengthen the paper. We are happy to incorporate additional feedback into the next version of our paper.
>
> ---
> ---
> > How does this work relate to Bourne et al., 2018? Could you clarify how your results relate to the duality findings of Caffarelli and McCann [9]?
>
> A: First, we appreciate the reviewer for bringing the work of Bourne et al. to our attention. We will cite their work in the final version. These papers also form connections among $\alpha$-OPT, $\lambda$-ROT, and weighted Voronoi diagrams. But we respectfully submit that we give a richer characterization that provides deeper insights into these problems. For example, [9] works with the weighted Voronoi diagram under the capped distance function and argues that the mass transported to $b$ in a $\lambda$-ROT plan is contained within the weighted Voronoi cell centered at $b$. We note that the Voronoi diagram under the capped distance is counter-intuitive, e.g., the boundary of a Voronoi cell may not be connected, and thus hard to interpret.  Similarly, Bourne et al. prove an optimal $\alpha$-OPT plan routes the mass of $b$ to a subset of the mass within its weighted Voronoi cell, and only find experimentally that the mass transported to each discrete point is contained within a ball (they do not prove the second claim). In contrast, we work with the weighted Voronoi diagram under the Euclidean (or squared Euclidean) distance, where the Voronoi diagram is more standard, and prove that the $\alpha$-OPT plan routes $b$ to all mass of its restricted Voronoi cell, i.e. its weighted Voronoi cell intersected with the ball of radius equal to the dual weight. With our method, we additionally prove that every point whose mass is not fully routed in the optimal $\alpha$-partial transport plan has a maximal dual weight, which is an even stronger guarantee. Thus we provide a more detailed characterization. For the sake of space, we did not go into as much detail on the differences between Section 2 and prior work. We appreciate the question and will make this difference more clear in a final version.
>
> Furthermore, [9] only studies the duality between exact solutions of $\lambda$-ROT and $\alpha$-OPT. In contrast, our main contribution  (see Section 3) is to show that an approximate solver for one could be used to compute an approximate solution for the other. This result is nontrivial, as shown by the pathological example in Section 3.
>
> We finally note that our work provides two new combinatorial algorithms to approximate the $\lambda$-ROT problem to varying degrees of precision.

---

> > ### Author Response · Authors · 2025-08-07
> >
> > We greatly appreciate your time and feedback, and we would be happy to clarify any remaining questions or concerns you might have.
> >
> > As the discussion deadline approaches, we’d be grateful if you have a chance to take a final look and let us know if there’s anything else we can address.

---

> > > ### Comment · Reviewer_umB2 · 2025-08-09
> > >
> > > Thank you very much for your detailed answer. I keep my positive evaluation of the paper.

---

### Official Review · Reviewer_3gQ4 · 2025-07-03

**Clarity:** 3
**Significance:** 2
**Originality:** 2
**Rating:** 4
**Confidence:** 4

**Summary:**

This paper presents two categories of semi-discrete OT which are the α-partial transport problem and TV-regularized optimal transport formulation with regularization parameter λ. The two categories can be demonstrated in one common formulation. The theoretical analysis reveals a fundamental connection between these problems, demonstrating that their optimal solutions can be characterized as restricted Laguerre diagrams - a geometric insight that enables us to establish their algorithmic equivalence. Building on this theoretical foundation, novel computational algorithm are developed. A rigorous analysis of the algorithm's precision and computational complexity is presented.

**Questions:**

1. Could numerical experiments be incorporated to validate the algorithm's efficacy?
2. These two robust formulations of semi-discrete optimal transport provide mathematically grounded mechanisms. It will enhance noise resistance through distinct regularization paradigms. Could this theoretical claim be empirically validated through some experiments?

**Ethical Concerns:**

["NO or VERY MINOR ethics concerns only"]

**Limitations:**

yes

**Paper Formatting Concerns:**

The format is good enough.

**Quality:**

3

**Strengths And Weaknesses:**

Strengths:
1. Provides a novel characterization of optimal solutions as restricted Laguerre diagrams, deepening the geometric understanding of partial semi optimal transportation with TV regularization. The theory is mathematically rigorous.
2. Establishes a strong equivalence between α-partial and λ-robust OT, enabling solvers for one problem to be adapted to the other.
3. The proposed method achieves exact solutions with guaranteed precision, while also providing an efficient approximation scheme, with their respective computational complexities rigorously analyzed.
Weaknesses:
The theoretical findings lack empirical validation and practical substantiation.

---

> ### Author Rebuttal · Authors · 2025-07-31
>
> We appreciate your thoughtful review and comments. We address your concerns below.
>
> **Summary of Contributions.**
>  Our paper presents the following key contributions:
>  - A characterization of partial and robust variants of the optimal transport (OT) problem in the semi-discrete setting,
>  - Theoretical guarantees on the approximation error when using an approximate $\lambda$-ROT solver to compute an approximate $\alpha$-OPT plan, and,
>  - Two algorithmic solutions for the $\lambda$-ROT problem.
>
> While the original submission emphasized our theoretical contributions, we appreciate your interest in practical implementation and experimental evaluation. In response, we include below preliminary experimental results. A more extensive evaluation will be added in the next version.
>
> **Implementation Details.**
> We provide a Python implementation of our algorithm. However, it relies on two core procedures that lack efficient implementations in Python:
> 1. Estimating continuous mass within regions defined by the arrangement of the restricted Voronoi cells and their expansions.
> 2. Maintaining a dynamic tree data structure as per Sleator and Tarjan (reference [44] in the appendix).
>
> To address these challenges in our prototype:
> - For (1), we approximate the mass within each region using a fine grid, summing the masses of grid points lying inside the region.
> - For (2), we simulate the dynamic tree by naïvely iterating over all edges.
>
> An efficient implementation will require optimized versions of these components, which remains a technical challenge in Python.
>
> ---
> ---
> **Preliminary Results on Algorithm Performance.**
> We evaluated our implementation using a continuous distribution sampled from a Gaussian distribution inside the unit square. The discrete support consists of $n$ samples drawn from the same distribution, each assigned a mass of $1/n$. We set $\lambda = 0.2$ and $\varepsilon = 0.02$. Results are averaged over 10 runs for each $n$.
>
> **Key Findings:**
> - The additive error stays within the target $\varepsilon = 0.02$.
> - The number of iterations in the two steps of our algorithm (Step 1 and 2) remains well below our upper bounds proved in the paper ($6n$ and $12n$ as described in Lemma 4.3 and D.9).
> - The total path and cycle lengths computed in the two steps grow subquadratically in $n$, aligning with our $O(n^2)$ complexity analysis (Section 4.3).
>
> The details of our experiments are presented in the following table.
>
> | n | Error | Step 1 iterations | Step 2 iterations | Step 1 paths and cycles length | Step 2 paths and cycles length | Regions |
> | :--: | :--: | :--: | :--: | :--: | :--: | :--: |
> | 10 | 0.009 | 1 | 4 | 41 | 361 | 198  |
> | 20 | 0.018 | 2 | 4 | 320 | 991  | 529  |
> | 30 | 0.021 | 3 | 5 | 688 | 1719  | 862  |
> | 40 | 0.015 | 3 | 5 | 979 | 2385  | 1212  |
> | 50 |  0.018 | 4 | 5 | 1603 | 3061  | 1544  |
> | 60 | 0.015 | 5 | 6 | 1973 | 3673  | 1855  |
> | 70 | 0.017 | 5 | 6 | 2512 | 4129  | 2130  |
> | 80 | 0.016 | 6 | 7 | 2845 | 4785  | 2421  |
>
>
> ---
> ---
> **Robustness of $\alpha$-OPT.**
>
> We examined the effect of 10% noise in the continuous distribution (added in the top-right corner) and measured how much noise was transported under various $\alpha$ values. Results confirm the robustness of $\alpha$-OPT: for $\alpha$ < 0.9, almost no noise mass is transported.
>
>
> |Alpha| Cost| \% inliers transported| \% noise transported|
> | :--: | :--: | :--: | :--: |
>  |0.700| 0.010| 0.777| 0.000|
>  |0.725| 0.011| 0.804| 0.000|
>  |0.750| 0.012| 0.830| 0.015|
>  |0.775| 0.013| 0.859| 0.005|
>  |0.800| 0.014| 0.887| 0.005|
>  |0.825| 0.016| 0.914| 0.009|
>  |0.850| 0.017| 0.942| 0.014|
>  |0.875| 0.019| 0.970| 0.005|
>  |0.900| 0.022| 0.996| 0.020|
>  |0.925| 0.031| 0.999| 0.244|
>  |0.950| 0.045| 0.999| 0.496|
>  |0.975| 0.060| 0.999| 0.748|
>
> **Robustness of $\lambda$-ROT.**
> We conducted similar experiments for $\lambda$-ROT. As $\lambda$ increases, total mass transported grows, but noise transport remains limited when the transported mass is under 0.9.
>
> |Lambda| Cost| Cost ROT| \% inliers transported| \% noise transported| Mass transported|
>    | :--: | :--: | :--: | :--: | :--: | :--: |
> | 0.05| 0.014| 0.024| 0.896| 0.004| 0.807|
> | 0.10| 0.020| 0.031| 0.980| 0.009| 0.884|
>    | 0.15| 0.021| 0.037| 0.991| 0.012| 0.894|
> | 0.20| 0.023| 0.042| 0.996| 0.046| 0.902|
> | 0.25| 0.024| 0.047| 0.999| 0.073| 0.907|
> | 0.30| 0.023| 0.051| 0.999| 0.064| 0.907|
> | 0.35| 0.025| 0.055| 0.999| 0.141| 0.914|
> | 0.40| 0.027| 0.061| 1.0| 0.147| 0.915|
> | 0.45| 0.028| 0.063| 1.0| 0.212| 0.921|
> | 0.50| 0.034| 0.066| 1.0| 0.349| 0.935|
> | 0.55| 0.041| 0.068| 1.0| 0.506| 0.951|
>
> Please let us know if any further clarification or experiments would strengthen the paper. We are happy to incorporate additional feedback into the next version of our paper.

---

> > ### Author Response · Authors · 2025-08-07
> >
> > We greatly appreciate your time and feedback, and we would be happy to clarify any remaining questions or concerns you might have.
> >
> > As the discussion deadline approaches, we’d be grateful if you have a chance to take a final look and let us know if there’s anything else we can address.

---

### Decision · Program_Chairs · 2025-09-17

**Decision:**

Accept (poster)

**Comment:**

The paper investigates the partial and robust variants of semi-discrete OT. All the reviewers agree that this is an important problem and the paper offers novel and interesting insights to this problem via the restricted Laguerre diagrams. The original submission did not contain any numerical verification results of the algorithm, however during the discussion period, such numerical results were provided to the satisfactory of the reviewers. For this reason, I think the paper has been improved and does meet the bar of acceptance. I strongly recommend that the authors incorporate these important information (experimental result, conclusion section) in the final paper if it is accepted.